# Non-normal flow rules affect fracture angles in sea ice viscous-plastic rheologies

Damien Ringeisen[1], L. Bruno Tremblay[2], and Martin Losch[1]

[1]Alfred-Wegener-Institut, Helmholtz-Zentrum für Polar- und Meeresforschung, Bremerhaven, Germany
[2]Department of Atmospheric and Oceanic Sciences, McGill University, Montréal, Canada

**Correspondence:** Damien Ringeisen (damien.ringeisen@awi.de)

**Abstract.** The standard viscous-plastic (VP) sea ice model with an elliptical yield curve and a normal flow rule has at least two issues. First, it does not simulate fracture angles below 30° in uni-axial compression, in contrast with observations of Linear Kinematic Features (LKFs) in the Arctic Ocean. Second, there is a tight, but unphysical coupling between the fracture angle, post-fracture deformation, and the shape of the yield curve. This tight coupling was identified as the reason for the overestimation of fracture angles. In this paper, these issues are addressed by removing the normality constraint on the flow rule in the standard VP model. The new rheology is tested in numerical uni-axial loading tests. To this end, an elliptical plastic potential — which defines the post-fracture deformations, or flow rule — is introduced independently of the elliptical yield curve. As a consequence, the post-fracture deformation is decoupled from the mechanical strength properties of the ice. We adapt the Roscoe's angle theory, which is based on observations of granular materials, to the context of sea ice modeling. In this framework, the fracture angles depend on both yield curve and plastic potential parameters. This new formulation predicts accurately the results of the numerical experiments with a root-mean-square error below 1.3°. The new rheology allows for angles of fracture smaller than 30° in uni-axial compression. For instance, a plastic potential with an ellipse aspect ratio smaller than two (i.e., the default value in the standard viscous-plastic model) can lead to fracture angles as low as 22°. Implementing an elliptical plastic potential in the standard VP sea ice model requires only small modifications to the standard VP rheology. The momentum equations with the modified rheology, however, are more difficult to solve numerically. The independent plastic potential solves the two issues with VP rheology addressed in this paper: in uni-axial loading experiments, it allows for smaller fracture angles, which fall within the range of satellite observations, and it decouples the angle of fracture and the post-fracture deformation from the shape of the yield curve. The orientation of the post-fracture deformation along the fracture lines (convergence and divergence), however, is still controlled by the shape of the plastic potential and the location of the stress state on the yield curve. A non-elliptical plastic potential would be required to change the orientation of deformation and to match deformation statistics derived from satellite measurements.

## 1 Introduction

Sea ice plays a significant role in the energy budget of the climate system and therefore has a strong influence on future climate projections. Sea ice dynamics are located primarily along narrow lines of deformation, called Linear Kinematic Features

(LKFs), where floes slide along and grind against each other. LKFs can form in divergence, creating stretches of open water or leads, or in convergence, creating piles of ice or ridges. LKFs in the Arctic sea ice cover influence the Earth system in many ways: heat and moisture exchange take place primarily over open water (Badgley, 1965), and salt rejection during ice formation in leads creates dense water and influences the thermohaline circulation (Nguyen et al., 2011, 2012; Itkin et al., 2015). Locally, the ice strength depends on the sea ice state (e.g., thickness, concentration, and damage), which in turn is affected by

sea ice fracture with thermodynamic growth in opening leads and with local dynamical growth during ridge formation. One observable and quantifiable feature of LKFs in Arctic sea ice is the intersection angles between individual LKFs. The LKFs have an influence on the local ice strength, emergent anisotropy and future deformation in the pack ice, and therefore sea ice mass balance (Aksenov and Hibler, 2001). Reproducing the LKFs patterns, density, and orientation is important for accurate sea ice and climate projections at high-resolution.

LKFs are ubiquitous features of granular media, and sea ice is often described as such a granular material (Overland et al., 1998; Erlingsson, 1988; Anderson, 1942; Schall and van Hecke, 2010). Similar to the crumbling of rocks, sea ice also exhibits brittle fracture, as floes break into smaller pieces. Brittle behavior adds a level of complexity because it implies that models must represent both the dynamics of intact ice (brittle — fracture or elastic regime) and the dynamics of a fractured system (granular — friction or plastic regime) (Handin, 1969). The dominant deformation process along LKFs is shear. Sometimes

this shear is associated with non-zero divergence, and this divergence along shear bands is referred to as *dilatancy* (Stern et al., 1995). Granular matter theory can explain the dilatancy along LKFs. In this work, we consider sea ice as a granular material and focus on the dynamics of the fractured system. We use the term *fracture* as the failure of a compact assemblage of floes and define the *fracture angle* as half of the angle between intersecting LKFs.

Different *rheological models* assume different material behavior before and after fracture. Common sea ice rheological

models are, for example, Viscous-Plastic (VP, Hibler, 1977), Elastic-Plastic (EP, Coon et al., 1974), Elastic-Anisotropic-Plastic (EAP, Tsamados et al., 2013), or Maxwell-Elasto-Brittle (MEB, Dansereau et al., 2016). In these different rheological models, various stress–strain(-rate) relationships, or *constitutive equations*, can be defined. In the following, we refer to models with different constitutive equations as different *rheologies*. We focus on the VP rheological model. A specific VP rheology is defined by a yield curve and plastic potential. The yield curve defines the stress criteria for the transition from small viscous

deformations (creep) to the large plastic deformations (friction). The plastic potential determines the ensuing post-fracture deformation, called the flow rule. The flow rule is normal to the plastic potential (Drucker and Prager, 1952). The plastic potential can be independent of, or equal to the yield curve. In the latter case, the flow rule is also normal to the yield curve and is called a normal-flow rule or associated flow rule. Several yield curves have been used in sea ice VP models, some with a normal flow rule (Hibler, 1979; Zhang and Rothrock, 2005) and some with a non-normal flow rule (Ip et al., 1991; Tremblay

and Mysak, 1997; Hibler and Schulson, 2000; Wang, 2007). We reiterate that two plastic models with the same yield curve but with different flow rules are referred to as two different rheologies, as they behave differently during the creation of LKFs.

The Viscous-Plastic rheology is an appropriate continuum rheology for modelling sea ice as a granular material because it includes (1) a yield condition for plastic deformation, and (2) a flow rule that allows to represent the divergent and convergent motion along shear lines, that is, the dilatancy observed in granular media. Continuum plastic flow models with normal or

non-normal flow rules are often used in other scientific fields to model granular geo-materials (Vermeer and De Borst, 1984; Mánica et al., 2018).

LKFs have been studied in satellite observations (Stern et al., 1995; Kwok, 2001; Schulson and Hibler, 2004; Weiss et al., 2007) and numerical models (Spreen et al., 2017; Hutter et al., 2018). In VP sea ice models, LKFs are represented as narrow zones of plastic deformation in a background field of nearly un-deformed ice (viscous creep) (Hutchings et al., 2005). This

behavior has been argued to be the reason for low temporal intermittency and spatial localization in VP models, leading to a spatial and temporal scaling of LKFs that is different from observations (Rampal et al., 2016). LKFs emerge clearly in plastic flow models at high resolution (Hutchings et al., 2005; Hutter et al., 2018; Koldunov et al., 2019). VP models reproduce observed intermittency and spatial localization even without brittle fracture dynamics (Bouchat and Tremblay, 2017; Hutter et al., 2018), albeit at higher resolution than Maxwell-Elasto-Brittle models (e.g., Rampal et al., 2019).

New models have been designed to represent sea ice fracture, for example, brittle models with a damage parameter that keeps the memory of previous fracture (Dansereau et al., 2016; Girard et al., 2011), or anisotropic viscous-plastic rheologies models (Tsamados et al., 2013; Heorton et al., 2018). Still, as of today, the viscous-plastic rheology with elliptical yield curve and normal flow rule (Hibler, 1979) is the *de facto standard* rheology in global climate models. For example, of the 33 global climate models of the Climate Model Inter-comparison Project 5 (CMIP5), 30 use the VP rheology with an elliptical yield

curve and normal flow rule (Stroeve et al., 2014). Below, we refer to this rheology as the *standard VP rheology*.

The orientation of LKFs is a well studied subject in the field of engineering and granular materials (LKFs are called shear bands in this field). Two classical solutions coexist and set two limit angles for the orientation of fractures: the Coulomb angle (static behavior) and the Roscoe angle (dynamic behavior). The Coulomb angle of fracture $\theta_C$ between the fracture line and the first principal stress is determined by the Mohr–Coulomb criterion. It is a function only of the internal angle of friction $\phi$

(Coulomb, 1773; Mohr, 1900):

$$\theta_C = \frac{\pi}{4} - \frac{\phi}{2}. \tag{1}$$

Roscoe (1970) challenged this concept by considering the case of dilatant material and found from experiments with sand that the dilatancy angle $\delta$ is the main parameter determining the orientation of shear bands (see Fig. 6 in Tremblay and Mysak, 1997, for a definition of the dilantancy angle $\delta$ in the context of sea ice modeling.). Dilatancy refers to divergence along shear

bands or LKFs. This divergence is a function of the distribution of contact points between individual floes at the sub-grid scales. A positive angle of dilatancy is associated with contact points that (on average) oppose the macroscopic shear motion and create divergence along the shear band; while negative dilatancy is associated with a closing of the shear line (ridging in the case of sea ice). The Roscoe angle of fracture is defined as:

$$\theta_R = \frac{\pi}{4} - \frac{\delta}{2}. \tag{2}$$

A general theory derived from experiments with sand that takes into account both the angle of friction and the angle of dilation combines the Coulomb and Roscoe angles as (Arthur et al., 1977; Vardoulakis, 1980):

$$\theta_A = \frac{\pi}{4} - \frac{1}{4}(\phi + \delta), \tag{3}$$

where $\theta_A$ is called the Arthur angle. Tremblay and Mysak (1997) used this general theory to design their sea ice rheology. Vermeer (1990) proposed a theoretical framework based on the grain size and showed that the angle of fracture in most experiments falls between the two extremes: $\theta_C \leq \theta \leq \theta_R$, with $\delta < \phi$ in sands. If $\phi = \delta$ then $\theta_R = \theta_C = \theta_A$, and the flow rule is normal to the yield curve. In other words, for $\phi = \delta$, the principal axes of stress and the principal axes of strain are coaxial. This condition, however, is not generally satisfied for granular materials: experiments with sand have shown differences between $\phi$ and $\delta$ of the order of $30°$ (Balendran and Nemat-Nasser, 1993; Vardoulakis and Graf, 1985; Bolton, 1986). Note that both mechanisms, friction and dilatancy, are not radically different: a larger dilatancy angle implies a larger grain size, more contact normals opposing the flow, hence more friction (Vermeer, 1990). The concept of the internal angle of friction can be used to link the orientation of LKFs to plastic rheologies with a normal flow rule (Ringeisen et al., 2019, Appendix B). So far, only the yield curve has been considered when investigating the orientation of LKFs in the viscous-plastic model (as in e.g., Hibler and Schulson, 2000; Hutchings et al., 2005; Wang, 2006). Therefore, it is unknown which of the three angles (Coulomb, Roscoe, Arthur) provide the most accurate prediction for this case.

The fracture angles with the standard VP rheology cannot be smaller than $30°$ in uni-axial compression, even by changing the ellipse aspect ratio $e$ (Ringeisen et al., 2019). In contrast, observations show fracture angles generally below $30°$ (e.g., $14°$ (Marko and Thomson, 1977), $15 \pm 1.5°$ (Erlingsson, 1988), $17°$ to $18°$ (Cunningham et al., 1994)) and a clear peak in the distribution of angles between $20°$ to $25°$ (Hutter and Losch, 2020). In addition, uni-axial loading compression experiments with lateral confinement (achieved via the addition of thinner ice surrounding the ice slab, Ringeisen et al., 2019) showed that: (1) the angle of fracture is a function of the slope of the yield curve in stress invariant space, (2) the ellipse aspect ratio determines the divergence along the LKFs, and (3) the fracture angle is a function of the confining pressure. These three properties of the standard VP rheology do not agree with the theory and observations of granular media behavior, namely that shear band orientations and divergent or convergent motion at the slip lines are a function mainly of the shear strength of the material and orientation of the contact normals (or dilatancy angle), and that the confining pressure has only a limited effect (Balendran and Nemat-Nasser, 1993; Alshibli and Sture, 2000; Han and Drescher, 1993; Desrues and Hammad, 1989). Note that some of these last experiments are tri-axial tests, and that bi-axial tests of 2D granular sea ice might yield different results as sea ice can "escape" in the vertical direction. Biaxial tests on sea ice samples show that small confinements lead to coulombic shear faults fractures with a similar internal friction coefficient and similar fracture angle. However, larger confinements lead to a spalling raft-like behavior with a broader range of fracture angles (Schulson et al., 2006a). The fracture angles are similar in different regions of the Arctic with different background stress conditions (Erlingsson, 1988; Marko and Thomson, 1977; Cunningham et al., 1994). This observation supports the hypothesis that the angle of fracture in sea ice is independent of the confining pressure. Finally, the distribution of intersection angles simulated with the standard VP rheology at high-resolution has a peak of the distribution at larger angles than the Radarsat Geophysical Processor System (RGPS) dataset (Hutter et al., 2019; Hutter and Losch, 2020). The unphysical behavior of the standard VP rheology (1, 2, 3, and the distribution of intersection angles) is connected to the shape of the yield curve in conjunction with a normal flow rule.

The flow rule has the advantage that it can be observed with remote sensing methods, in contrast to observing stress which requires in-situ measurements. The ratio of shear to divergence along the shear bands or LKFs allows to infer the dilatancy

angle of granular material. Observations of sea ice drift in the Arctic show that most of the deformation takes place in shear with some divergence (Stern et al., 1995). The distribution of the ratio of divergence and convergence can be reproduced by modifying the ellipse aspect ration $e$ of the standard VP rheology (Bouchat and Tremblay, 2017). Separating the link between the fracture angle and the flow rule from the yield curve is necessary to design VP rheologies that are consistent with observed sea ice deformation.

In this paper, we investigate the effects of a non-normal flow rule on fracture angles. We use the non-normal flow rule as a means of separating the state of stress (at failure) and the post-fracture deformation. To this end, we study the non-normal flow rule in the context of the standard VP rheological model using a similar shape for the plastic potential (i.e., an ellipse) because (1) the ellipse is widely used in the community, and (2) its behavior is well documented (compared to other models), providing a solid basis for comparison. For these two reasons, we use the elliptical yield curve despite the fact that it is not the most appropriate yield curve to model sea ice as a granular material. This paper provides a new generalized theoretical framework for any viscous-plastic material with normal or non-normal flow rules. Following Ringeisen et al. (2019), we test the new model in simple uni-axial loading experiments where the relationship between fracture angle and flow-rule can be easily identified.

The paper is structured as follows. Section 2 describes the model (2.1), the new rheology (2.2), and a general theory linking the fracture angles and a general flow rule (2.3). The sections 3 and 4 describe the idealized experimental setup and the results. Section 5 discusses these results and their implication on current and future rheologies. Conclusions follow in section 6.

## 2   Sea ice Model and rheology

### 2.1   Building the sea ice VP constitutive equations

We consider sea ice as a 2D viscous-plastic material. The ice velocities are calculated from the sea ice momentum equations:

$$\rho h \frac{\partial \boldsymbol{u}}{\partial t} = -\rho h f \, \boldsymbol{k} \times \boldsymbol{u} + \boldsymbol{\tau}_a + \boldsymbol{\tau}_o - \rho h \nabla \phi_s + \nabla \cdot \boldsymbol{\sigma}, \tag{4}$$

where $\rho$ is the ice density, $h$ is the grid cell averaged sea ice thickness, $\boldsymbol{u}$ is the ice drift velocity field, $f$ is the Coriolis parameter, $\boldsymbol{k}$ is the vertical unit vector, $\boldsymbol{\tau}_a$ is the surface air stress, $\boldsymbol{\tau}_o$ is the ocean drag, $\nabla \phi_s$ is acceleration from the gradient of sea surface height, and $\boldsymbol{\sigma}$ is the vertically integrated internal ice stress tensor defined by the sea ice VP constitutive equations. The constitutive equations define the vertically integrated stress tensor $\sigma$ as function of the strain rate tensor $\dot{\epsilon}$ and the state variables $\chi$ (e.g., ice thickness, ice strength, ice concentration). The components of the strain rate tensor are computed from the velocities as $\dot{\epsilon}_{ij} = \frac{\partial u_i}{\partial x_j}$. The constitutive equations then have the form:

$$\boldsymbol{\sigma} = f(\dot{\boldsymbol{\epsilon}}, \chi). \tag{5}$$

In the sea ice VP model the stresses are independent of the strain rates for large deformation events (the plastic states with stresses on the yield curve) and they depend on the strain rates for small deformations (the viscous states with stresses inside

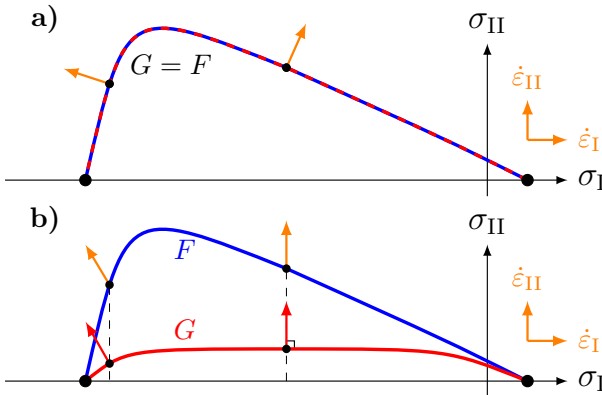

**Figure 1.** Schematic yield curve F (blue) and plastic potential G (red) for a normal (**a**) and non-normal (**b**) flow rule. The flow rule (orange) for a given stress on the yield curve is normal to the plastic potential (red) for the same $\sigma_{\mathrm{I}}$. Note that the stress and strain invariant axes are assumed to coincide.

the yield curve). It is this set of equations that defines the rheology of sea ice and determines the fracture pattern and the opening or closing along the fractures.

One of the state variables in the model is the maximum compressive strength $P$. This variable represents the maximum compressive stress that sea ice can bear in uniform compression before ridging. We use the simple standard relationship (Hibler, 1979):

$$P = P^{\star} h e^{-C^{*}(1-A)}, \tag{6}$$

where $C^{\star}$ is a free parameter (typically $C^{\star} = 20$), $h$ is the mean ice thickness, $A$ is the fractional sea ice area cover in a grid
cell, and $P^{\star}$ is the ice strength of 1 m ice at 100% concentration ($A = 1$).

The yield curve represents the stress states for which sea ice deforms plastically while enclosing the stress states for which sea ice slowly deforms viscously. We express the yield curve as a function of the stresses $\sigma_{ij}$ and the state variables $\chi$:

$$F(\sigma_{ij}, \chi) = 0. \tag{7}$$

The yield curve can be represented in principal stress ($\sigma_1$ and $\sigma_2$) or stress invariants space ($\sigma_{\mathrm{I}}$ and $\sigma_{\mathrm{II}}$). Figure 1 shows an
arbitrary yield curve in stress invariants space. Although equation (7) determines if the deformation is plastic or viscous, it does not determine how the ice will deform after fracture. In order to obtain a closed system of equations, we define a plastic potential that defines the flow rule.

The plastic potential determines the direction of deformation for stress states on the yield curve. The flow rule represents the direction of deformation in the grid cell. The orientation of the flow rule in the coordinate system ($\dot{\epsilon}_{\mathrm{I}}, \dot{\epsilon}_{\mathrm{II}}$), as shown in orange in
Fig. 1, indicates if the grid cell deforms in convergence ($\dot{\epsilon}_{\mathrm{I}} < 0$) or divergence ($\dot{\epsilon}_{\mathrm{I}} > 0$), and shear ($\dot{\epsilon}_{\mathrm{II}}$). Just as the yield curve,

the plastic potential can be written as:

$$G(\sigma_{ij}, \chi) = 0. \tag{8}$$

The direction of the deformation, called the flow rule, is perpendicular to the plastic potential. This is shown in red on Fig. 1b and mathematically expressed by

$$\frac{\partial G}{\partial \sigma_{ij}}(\sigma_{ij}, \chi) = \lambda \dot{\epsilon}_{ij}, \tag{9}$$

where $\lambda > 0$ is the unknown flow rate. The flow rule is applied for stress states on the yield curve at the same compressive stresses (orange arrows in Fig. 1b). If the plastic potential and the yield curve are the same ($G = F$), the flow rule is called an *associated* or *normal* flow rule, as the flow rule is also perpendicular to the yield curve (see Fig. 1a).

Using Eq. (7) and Eq. (9), we can write a system of 5 equations (four from Eq. 9 and one from Eq. 7) for 5 unknowns ($\sigma_{11}$, $\sigma_{22}$, $\sigma_{12}$, $\sigma_{12}$, $\lambda$). Solving this system of equations allows us to write the constitutive equations for the sea ice model as function of the components of the strain-rate ($\dot{\epsilon}_{11}$, $\dot{\epsilon}_{22}$, $\dot{\epsilon}_{12}$, $\dot{\epsilon}_{21}$) and the state variables $\chi$.

After deriving these constitutive equations, we assume that the stress and strain rate tensors are symmetric, that is, $\sigma_{12} = \sigma_{21}$ and $\dot{\epsilon}_{ij} = \frac{1}{2}\left(\frac{\partial u_i}{\partial x_j} + \frac{\partial u_j}{\partial x_i}\right)$. The symmetry follows from ignoring the rotation in an isotropic medium. Note that we first need to solve this system of equations without using the symmetry condition: the symmetry condition is only invoked at the end. Applying the symmetry before solving the system of equation changes the nature of the initial tensor, and the resulting constitutive equations would be different.

An ideal plastic model, with the stresses independent of the strain rates, has a singularity because the non-linear viscosities tend to infinity as the strain rates tend to zero. Hibler (1977) solved this issue with a regularization that limits the value of the bulk and shear viscosities $\zeta$ and $\eta$ to a maximum value. When the viscosities are capped to their maximum values, the stresses are linearly related to the strain rates and the material behaves as a viscous material. VP sea-ice models typically cap the viscosity at

$$\zeta_{\max} = \frac{1}{2\Delta_{\min}} \cdot P = \left(2.5 \times 10^8 \, \text{s}\right) \cdot P \tag{10}$$

and $\eta_{\max} = \frac{\zeta_{\max}}{e_G^2}$ to regularize the momentum equations. When this regularization is in effect, $\zeta$ and $\eta$ are independent of the deformation field ($\Delta$) and the stress divergence reduces to harmonic viscosity with constant coefficients. $\Delta_{\min} = 2 \times 10^{-9} \, \text{s}^{-1}$ (Hibler, 1979, 1977) translates to a deformation time scale of almost 16 years. Therefore, viscous deformations are slow and negligible with respect to the plastic deformations that operate on synoptic time scale, and sea ice VP rheologies can be considered as ideal plastic. The viscous behavior can be seen as a consequence of regularizing the viscosities rather than an implementation of a physical behavior.

## 2.2 Elliptical yield curve with non-normal flow rule

We now build a rheology with an elliptical yield curve and a non-normal flow rule, that is, we use a plastic potential $G$ that is different from the yield curve $F$. By doing this, we will change the orientation of the flow rule, without changing the yield stress

state (see Fig. 2 and Fig. 4 for some examples). We use a different, but still elliptical plastic potential for simplicity: this choice requires only minor modifications to a typical VP sea ice model. We define the yield condition $F$ and the plastic potential $G$ as a function of the state variables $\chi$: the ice compression strength $P$, the ice tensile strength $T = k_t P$ (König Beatty and Holland, 2010), the yield curve's ellipse ratio $e_F$, and the plastic potential's ellipse ratio $e_G$ by

$$X(\sigma_\mathrm{I}, P, e_X, k_t) = \left( \frac{\sigma_\mathrm{I} + \frac{P(1-k_t)}{2}}{\frac{P(1+k_t)}{2}} \right)^2 + \left( \frac{\sigma_\mathrm{II}}{\frac{P(1+k_t)}{2e_X}} \right)^2 - 1 = 0, \tag{11}$$

for $X = F, G$ for the yield curve or the plastic potential. Using Eq. (11), we write $\sigma_\mathrm{II}$ as a function of $\sigma_\mathrm{I}$ as:

$$\sigma_{\mathrm{II},X} = \frac{1}{e_X} \sqrt{P^2 k_t - \sigma_\mathrm{I}^2 - \sigma_\mathrm{I} P (1 - k_t)}. \tag{12}$$

Following Hibler (1977, 1979), we derive the constitutive equations $\sigma_{ij}$:

$$\sigma_{ij} = 2\eta \dot{\epsilon}_{ij} + (\zeta - \eta) \dot{\epsilon}_{kk} \delta_{ij} - \frac{P(1-k_t)}{2} \delta_{ij}, \tag{13}$$

where the shear and bulk viscosities $\eta$ and $\zeta$ are defined by:

$$\zeta = \frac{P(1+k_t)}{2\Delta} \text{ and } \eta = \frac{\zeta}{e_G^2} = \frac{P(1+k_t)}{2e_G^2 \Delta} \tag{14}$$

with

$$\Delta = \sqrt{(\dot{\epsilon}_{11} - \dot{\epsilon}_{22})^2 + \frac{e_F^2}{e_G^4} ((\dot{\epsilon}_{11} - \dot{\epsilon}_{22})^2 + 4\dot{\epsilon}_{12}^2)}. \tag{15}$$

Figure 2 shows an example of yield curve and plastic potential, with the resulting flow rule. For $e_G > e_F$, the absolute value of the divergence is smaller and the shear strain rate is larger compared to a normal flow rule ($e_G = e_F$) and vice versa for $e_G < e_F$.

## 2.3 Linking fracture and flow rule

In this section, we generalize the theory linking the rheological model and the fracture angles in simple uni-axial compressive test (Ringeisen et al., 2019) to materials with a non-associated flow rule. To this end, we follow the theory of Roscoe (1970) where the angle of fracture depends uniquely on the angle of dilatancy of a granular material. Based on laboratory experiments, Roscoe (1970) states that the *velocity characteristics* (the post-failure deformation) seem to be a better predictor than the *stress characteristics* (the stress at failure) for the orientation of shear bands in granular materials. The Roscoe angles can then be compared to the Coulomb angles, as defined in Ringeisen et al. (2019), and to the results from the idealized experiments in Section 4.

Figure 3 illustrates the case of an arbitrary yield curve with an arbitrary plastic potential. To adapt the Roscoe angles to sea ice modeling, we proceed as follows: (1) the stress state on the yield curve (point **p** on Fig. 3a) defines the position and size the Mohr's circle at fracture (blue circle on Fig. 3b), (2) the slope of the plastic potential determines the point on the Mohr's circle

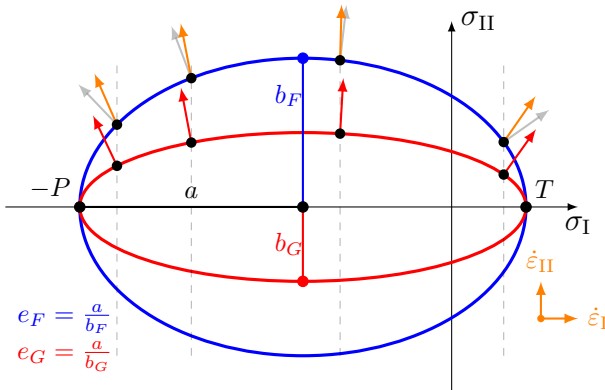

**Figure 2.** Elliptical yield curve with a non-normal flow rule, a yield curve ellipse aspect ratio $e_F = 2$ (blue) and a plastic potential ellipse aspect ratio $e_G = 4$ (red). The gray and orange arrows show the normal and non-normal flow rules, respectively.

where deformation takes place, that is, the slope directly predicts the fracture angle $\theta$ as a function of the dilatancy angle $\delta$ (per Roscoe theory, Fig. 3b). For the special case of uni-axial compression, we (A) determine the stress state on the yield curve for uni-axial compression as a function of the yield curve ellipse ratio $e_F$, and (B) compute the slope of the plastic potential at that stress state as a function of the plastic potential ellipse ratio $e_G$. Finally, we combine (2) and (B) to compute the theoretical prediction for the fracture angle as a function of ellipse ratios $e_G$ and $e_F$. Figure 3 shows the geometrical construction that links the angle of dilatancy $\delta$ to the slope of the plastic potential $\tan(\gamma_G)$:

$$\sin(\delta) = \tan(\gamma_G) = -\frac{\partial \sigma_{II,G}}{\partial \sigma_I}. \tag{16}$$

Note that the minus sign above was included in the derivative of the yield curve function in Eq. (B1) and (B2) of Ringeisen et al. (2019). This equation agrees with the definition of Roscoe (1970) $\sin(\delta) = \frac{\dot{\epsilon}_I}{\dot{\epsilon}_{II}}$, because the ratio of $\dot{\epsilon}_I$ to $\dot{\epsilon}_{II}$ is equal to the slope of the plastic potential $-\frac{\partial \sigma_{II,G}}{\partial \sigma_I}$, as the flow rule is perpendicular to the plastic potential. Figure 3 also shows the normal flow rule, which, in agreement with the coulombic theory, would lead to different fracture angles (light blue lines). From Fig. 3, the fracture angle can be written as:

$$\theta_R = \frac{\pi}{4} - \frac{\delta}{2}. \tag{17}$$

Substituting Eq. (16) in the equation above, the relationship between the fracture angle and the plastic potential becomes

$$\theta_R(\sigma_I) = \frac{1}{2}\left[\frac{\pi}{2} - \arcsin\left(-\frac{\partial \sigma_{II,G}}{\partial \sigma_I}(\sigma_I)\right)\right] \tag{18}$$

$$= \frac{1}{2}\arccos\left(-\frac{\partial \sigma_{II,G}}{\partial \sigma_I}(\sigma_I)\right). \tag{19}$$

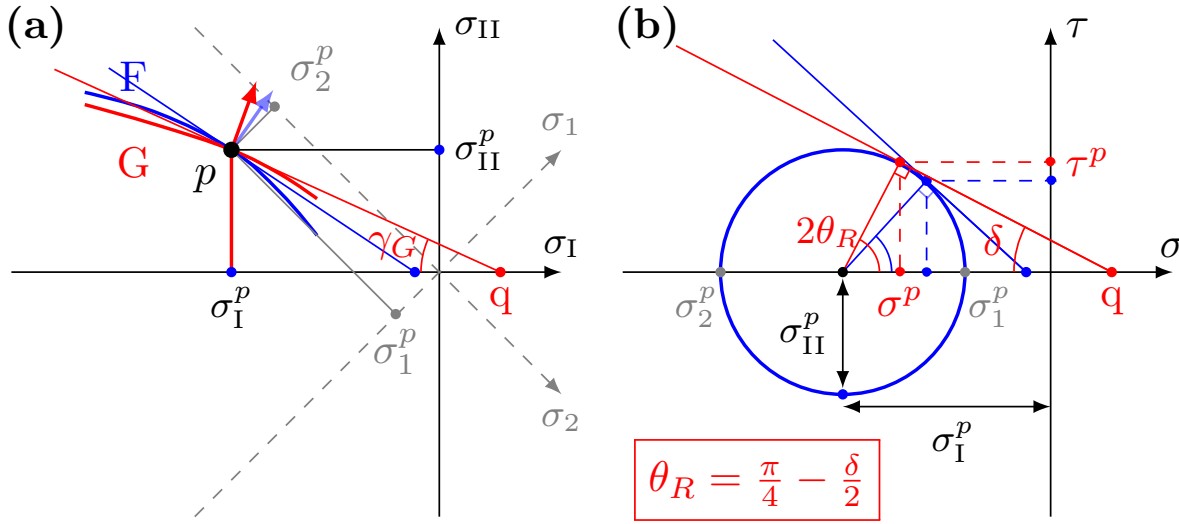

**Figure 3.** Link between fracture angle and yield curve: **(a)** Arbitrary yield curve F (blue) and plastic potential G (red) in stress invariants space. The plastic potential and yield curve intersect at a stress state $p$ for illustration purposes only. The red arrow is perpendicular to $G$, but non-normal to the yield curve $F$. The tangent to the plastic potential G at point $p$ has a slope $\mu_G = \tan(\gamma_G)$ and intersects the $\sigma_\mathrm{I}$-axis at point $q$ (thin red line). For reference, the normal and tangent to the yield curve F are shown as a thin blue arrow and line. Gray dashed lines show the principal stress axes. **(b)** Mohr's circle for the fracture state p in (a) (for normal in blue and for non-normal flow rule in red) in the fracture plane of reference $(\sigma, \tau)$ of center $\sigma_\mathrm{I}^p$ and radius $\sigma_\mathrm{II}^p$. The thin red line is the tangent to the Mohr's circle passing through the point $q$ on the $\sigma$ axis. By this geometrical construction, $\sin(\delta) = \tan(\gamma_G) = \mu_G$ (Only valid for $|\mu_G| \leq 1$). $\delta$ is called the dilatancy angle. Again for comparison with panel (a), the blue lines and dots depicts the case of the normal flow rule (i.e., $G = F$). When considering the plastic potential, the angle of fracture is written as of the dilatancy angle $\delta$ as $\theta = \frac{\pi}{4} - \frac{\delta}{2}$. By comparison again, $\delta = \phi$ in the case of a normal flow rule and the Roscoe theory ($\theta = \frac{\pi}{4} - \frac{\delta}{2}$) reduces to the Mohr–Coulomb theory ($\theta = \frac{\pi}{4} - \frac{\phi}{2}$).

We calculate the fracture angles for the elliptical yield curve with non-normal flow rule in uni-axial compression along the $y$ axis. In this case, $\sigma_{11} = \sigma_{12} = 0$, $\sigma_{22} < 0$, and the principal stresses and stress invariants can be written as:

$$\sigma_1 = \frac{1}{2}\left(\sigma_{11} + \sigma_{22} + \sqrt{(\sigma_{11} - \sigma_{22})^2 + 4\sigma_{12}^2}\right) = 0, \tag{20}$$

$$\sigma_2 = \frac{1}{2}\left(\sigma_{11} + \sigma_{22} - \sqrt{(\sigma_{11} - \sigma_{22})^2 + 4\sigma_{12}^2}\right) = \sigma_{22}. \tag{21}$$

$$\sigma_\mathrm{I} = \frac{\sigma_1 + \sigma_2}{2} = \frac{\sigma_{22}}{2} \tag{22}$$

$$\sigma_\mathrm{II} = \frac{\sigma_1 - \sigma_2}{2} = -\frac{\sigma_{22}}{2} = -\sigma_\mathrm{I}. \tag{23}$$

From Eq. 23, the maximum shear stress $\sigma_{\mathrm{II},F}^p$ in the fracture plane in uni-axial compression can be expressed as

$$\sigma_{\mathrm{II},F}^p(\sigma_\mathrm{I}^p) = -\sigma_\mathrm{I}^p, \tag{24}$$

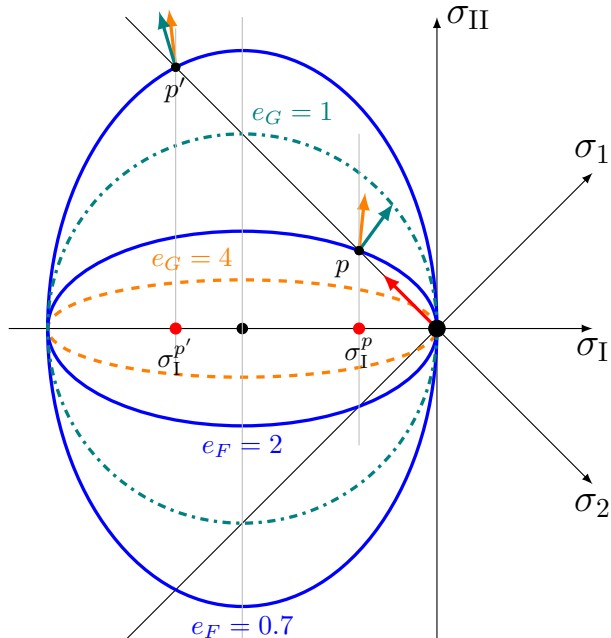

**Figure 4.** Trajectory of maximum normal stress (red arrow) in a uni-axial loading test experiment in a material with two different elliptical yield curves (blue) and plastic potentials (dashed orange and dash-dotted teal). The orange and teal arrows show the flow rule normal to the plastic potential of the same colour for the same stress state. For $e_G < e_F$, the ratio of divergence to shear increases. The opposite is true for $e_G > e_F$. A similar figure in principal stress space is presented in Ringeisen et al. (2019)

where $p$ indicates the stress state at the fracture. Figure 4 shows the stress trajectory in principal stress space for uni-axial compression. It also shows how the flow rule changes for the same stress state when using two different elliptical plastic potentials.

In the following, we use the normalized stress invariants $\sigma_I' = \frac{\sigma_I}{P}$ and $\sigma_{II}' = \frac{\sigma_{II}}{P}$ to simplify the notation. The slope of the yield curve or the plastic potential depends only on $e_F$ and $e_G$, but not on $P$. Substituting Eq. (12), $\sigma_I'$, and $\sigma_{II}'$ in Eq. (24), we obtain,

$$\sigma_{II}'^p = -\sigma_I'^p = \frac{1}{e_F}\sqrt{k_t - \sigma_I'^p(\sigma_I'^p + 1 - k_t)}, \tag{25}$$

and solve the first stress invariant $\sigma_I^p$ on the fracture plane in uni-axial compression

$$\sigma_I'^p = \frac{(k_t - 1) - \sqrt{(1 - k_t)^2 + 4k_t(1 + e_F^2)}}{2(1 + e_F^2)}. \tag{26}$$

The slope of the tangent at $\sigma_I^p$ to the plastic potential is given by the derivative of Eq. (12):

$$\frac{\partial \sigma_{II,G}'}{\partial \sigma_I'}(\sigma_I'^p) = \frac{1}{2e_G}\frac{-2\sigma_I'^p - 1 + k_t}{\sqrt{k_t - \sigma_I'^p(\sigma_I'^p + 1 - k_t)}}. \tag{27}$$

Substituting Eq. (26) into Eq. (27), yields

$$\left.\frac{\partial \sigma'_{\text{II},G}}{\partial \sigma'_{\text{I}}}\right|_{\sigma'^{P}_{\text{I}}} = \frac{1}{e_G e_F}\left(1 - \frac{(1+e_F^2)}{1+\sqrt{1+4\frac{k_t}{(1-k_t)^2}(1+e_F^2)}}\right). \tag{28}$$

or for zero tensile strength ($k_t = 0$),

$$\left.\frac{\partial \sigma'_{\text{II},G}}{\partial \sigma'_{\text{I}}}\right|_{\sigma'^{P}_{\text{I}},k_t=0} = \frac{1}{2e_F e_G}(1 - e_F^2). \tag{29}$$

The fracture angle can finally be written as a function of $e_G$ and $e_F$ from Eq. (18):

$$\theta_{e,nn}(e_F, e_G) = \frac{1}{2}\arccos\left(\frac{1}{2e_F e_G}(e_F^2 - 1)\right). \tag{30}$$

As expected, for $e_F = e_G = e$ we recover the fracture angle derived in Ringeisen et al. (2019):

$$\theta_{e,n}(e) = \frac{1}{2}\arccos\left[\frac{1}{2}\left(1 - \frac{1}{e^2}\right)\right]. \tag{31}$$

## 3   Experimental setup and numerical scheme

Following Ringeisen et al. (2019), we load a rectangular ice floe of 8 km by 25 km with a uniform thickness of $h = 1\,\text{m}$ and a uniform sea ice concentration of $A = 100\%$ (see Fig. 5). The numerical domain has the dimensions $L_x = 10\,\text{km}$ and $L_y = 25\,\text{km}$. At $y = 0$, we use a closed, solid boundary with a no slip condition (i.e., $u = v = 0$). At $x = 0$ and $L_x$, we use Neumann boundary conditions:

$$\left.\frac{\partial A}{\partial x}\right|_{x=0,L_x} = \left.\frac{\partial h}{\partial x}\right|_{x=0,L_x} = \left.\frac{\partial u}{\partial x}\right|_{x=0,L_x} = \left.\frac{\partial v}{\partial x}\right|_{x=0,L_x} = 0. \tag{32}$$

On the left and right sides of the domain ($x < 1\,\text{km}$ and $x > 9\,\text{km}$), we have open water between the ice floe and the boundary to ensure that the boundaries have no effect on the simulation. At ($y = L_y$), we use a Dirichlet boundary condition for ice velocity ($v$ the velocity in $y$-direction increasing linearly in time simulating an axial loading test) and a Neumann boundary condition for ice thickness and concentration :

$$v(t)|_{y=L_y} = a_v \cdot t\,,\ u(t)|_{y=L_y} = 0 \tag{33}$$

$$\left.\frac{\partial A}{\partial y}\right|_{y=L_y} = \left.\frac{\partial h}{\partial y}\right|_{y=L_y} = 0 \tag{34}$$

with $a_v = -5 \cdot 10^{-4}\,\text{m}\,\text{s}^{-2}$. The grid spacing of the domain is 25 m, and the timestep is 0.1 s. For simplicity, the Coriolis parameter $f = 0$.

The non-linear momentum equations (4) are integrated using a Picard solver with 15 000 non-linear (or outer-loop) iterations (Losch et al., 2010). For the linearized problem within each non-linear iteration, we use a line successive (over-)relaxation (LSR) method (Zhang and Hibler, 1997), with a tolerance criterion of $|u_k - u_{k-1}|_{\max} < 10^{-11}\,\text{m}\,\text{s}^{-1}$, where $k$ is the linear

iteration index. We use an inexact approach with only a maximum of 200 linear iterations for the linearized equations; the linearized system does not reach the tolerance criterion for the first non-linear iterations, but does so as the non-linear system approaches a converged solution. We chose a very small tolerance and residual norm for the solution of the linear and non-linear problem in order to simulate a clean fracture with a well defined fracture angle — for comparison with theory and observations. These criteria are much stricter than common recommendations for Arctic sea ice simulations (e.g. Lemieux and Tremblay, 2009). We expect that numerical sea ice models are computationally more challenging with a non-normal flow rule than with a normal flow rule. The non-normality of the flow rule relative to the yield curve introduces more complexity because Drucker's stability postulate is not satisfied (Vermeer and De Borst, 1984; Balendran and Nemat-Nasser, 1993). This particular uni-axial loading experiment is also complex to solve numerically because the forcing is localized on the boundary, in contrast to real geophysical system integrations where wind and ocean currents are acting over the entire surface of the ice.

The intersection angles between the LKFs are measured manually with the *Measure Tool* from the GNU Image Manipulation Program (GIMP, version 2.8.16, gimp.org). We estimated the accuracy as $\pm 1°$ (Ringeisen et al., 2019). The first 5 seconds of simulations are used to define the sea ice fracture and calculate the fracture angle. Although the forced deformation is very slow, the stresses reach the yield curve already in the first timestep (0.1 s). The fracture is created immediately, but because of the large viscosity of the viscous states with a deformation timescale of approximately 35 years the fracture progression is not visible immediately (Ringeisen et al., 2019). Therefore we show the deformation after 5 s. During these 5 s there is no fundamental change other than the initial deformation becoming clearer. The angle of each fracture line is measured and used to compute the average fracture angle and the standard deviation. Note that the fracture angles do not depend on resolution, scale, geometry, or boundary conditions (see Ringeisen et al., 2019, their Sec. 3.2 ). We do not use a replacement pressure scheme (Ip et al., 1991; Ip, 1993), because it has no influence with the angle of fracture (not shown).

# 4 Results

We study the evolution of the fracture angle $\theta$ when the plastic potential changes while the yield curve stays the same (see Fig. 4 for details). In this manner, the ice breaks for the exact same stress state but with a different flow rule. For simplicity, we test here the elliptical yield curve without tensile strength ($k_t = 0$).

Figure 6 shows the fracture pattern for the standard yield curve ellipse ratio $e_F = 2.0$ and three values of the plastic potential ellipse ratio $e_G = 1.4$, $2.0$, and $4.0$. The fractures form a diamond shape, similar to the shapes observed at large scales (Erlingsson, 1988), in laboratory experiments (Schulson, 2001), and modeled with DEM models (Wilchinsky et al., 2010) or other continuous sea ice models (Ringeisen et al., 2019; Heorton et al., 2018). With a normal flow rule ($e_G = 2.0$), single pairs of fracture lines with one unique fracture angle, large deformation along the LKFs, and smaller deformations (by several orders of magnitude) within diamond-shape floes are simulated. With the non-normal flow rule ($e_G = 1.4$ and $e_G = 4$) we make three observations:

1. Asymmetric secondary fracture lines appear, in contrast to the normal flow rule simulation. We attribute the asymmetry and presence of secondary fractures to the lack of full numerical convergence associated with the violation of Drucker's

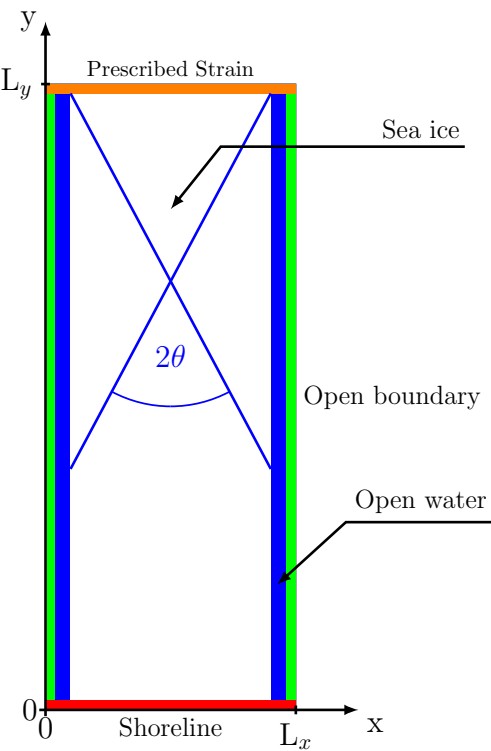

**Figure 5.** Model domain with a solid wall at $y = 0$ (red), Dirichlet boundary conditions with $\boldsymbol{u} = 0$ at $y = 0$ and prescribed velocities at $y = L_y$. Open boundaries at $x = 0, L_x$ (green) with Neumann boundary conditions. For the conservation of mass, ice thickness and concentration equations ($h$, $A$) Neumann boundary conditions are used on all boundaries. $\theta$ is the measured fracture angle with respect to the vertical; the blue line represents an LKF.

principle, or the non-normality of the flow rule (the ratio of divergence to shear strain rate differs from that of the shear to normal stress). For instance, the $L_2$ norm of the residual (R) in the non-linear equations decreases by four orders of

magnitude for the normal flow rule compared with two orders of magnitude for the non-normal flow rule for the same number of non-linear iterations (15 000); specifically to $R = 8 \times 10^{-4}$ for $e_G = e_F = 2$ to $R = 6 \times 10^{-2}$ for $e_G = 1.4$ and $e_F = 2$. Note that a Jacobian-free Newton-Krylov (JFNK) solver with a quadratic local numerical convergence does not perform better because the global convergence is poor with a combination of localized forcing and high grid resolution (Losch et al., 2014; Williams et al., 2017).

2. The width and activity of the LKFs is also affected by the flow rule. With $e_G = 1.4$, the lines are thinner, the shear along the LKFs is smaller and there is little shear between the fracture lines. With $e_G = 4.0$, the fracture lines are broader, the shear strain rate along the LKFs is higher and there is more shear between the fracture lines. With $e_G = 1.4$ the

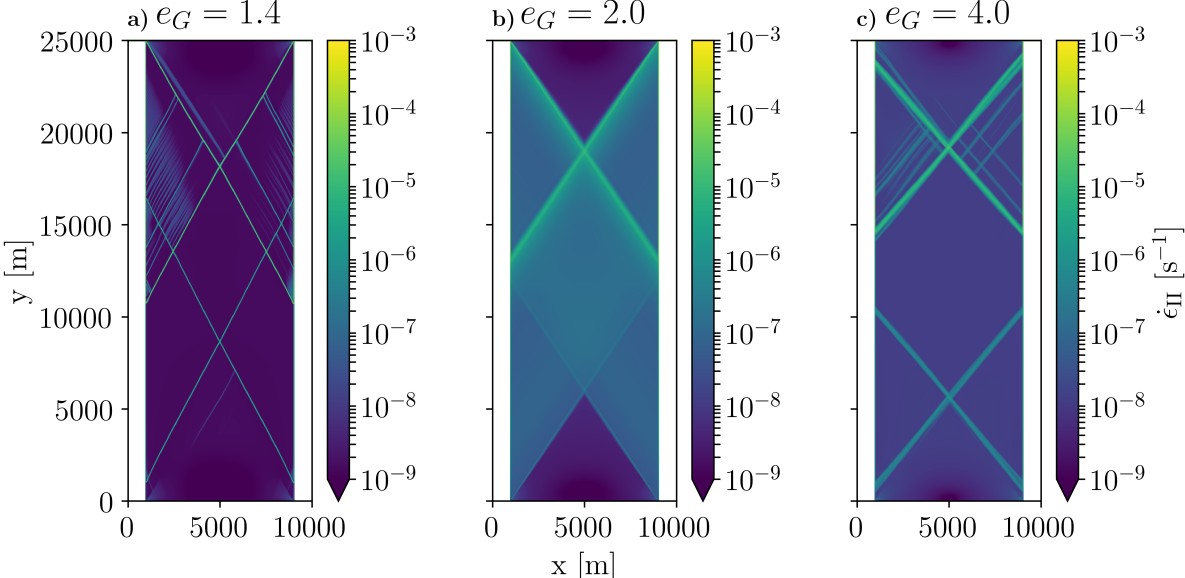

**Figure 6.** Diamond-shaped fracture pattern in the shear deformation field $\dot{\epsilon}_{II}$ for $e_F = 2.0$ and three different values of $e_G$ after five seconds of simulation. For the non-normal flow rule (panels a and c), there are primary and secondary fracture lines, in contrast to the normal flow rule (panel b) where single pair of fracture lines are simulated. The fracture angles are $29.92 \pm 1.28°$ for $e_G = 1.4$, $34.3 \pm 0.25°$ for $e_G = 2.0$, and $40.7 \pm 0.94°$ for $e_G = 4.0$. The error corresponds to two standard deviation ($2\sigma$) of the measured fracture angles.

deformation at the fracture is mainly in divergence, while for $e_G = 4.0$, the deformation is mainly in shear and there is more stress transmitted to the ice in between the fracture lines.

340 3. The fracture angle changes as the plastic potential changes. The angles are wider with $e_G = 4$ than $e_G = 1.4$. The effect of flow rule orientation on the fracture angles is discussed below.

We now present results from four sets of simulations with fixed yield curve ellipse ratios at $e_F = 0.7, 1.0, 2.0, 4.0$. For each of these, we test the sensitivity of the results to changes in the plastic potential ellipse ratio $e_G$. The choice of yield curve ellipse ratios $e_F$ are: the standard value of (Hibler, 1979), values suggested by Bouchat and Tremblay (2017) and Dumont et al.
345 (2009), and an extreme value resulting in a very small shear strength and smaller fracture angles.

Figure 7a shows how the fracture angles evolve as the plastic potential ellipse ratio $e_G$ changes for each of the four values of $e_F$. There is a clear dependence of the fracture angles on the relative eccentricity of the plastic potential and yield curve. For $e_G > e_F$, the shear strain rate increases along the LKFs (see Fig. 6c) and the fracture angles tend toward $45°$ as $e_G$ increases, in agreement with the theory (Eq. 18). For $e_G < e_F$, the flow rule implies more divergence (for $e_F > 1$, or convergence for
350 $e_F < 1$) and less shear along the LKFs (see Fig. 6a), and the fracture angles move away from $45°$ as $e_G$ decreases. More generally, for $e_F < 1$, the fracture angle increases with increasing convergence along the LKFs as $e_G$ decreases. For $e_F > 1$, the fracture angle decreases with increasing divergence as $e_G$ decreases. For $e_F = 1$ (a circular yield curve), the fracture angles

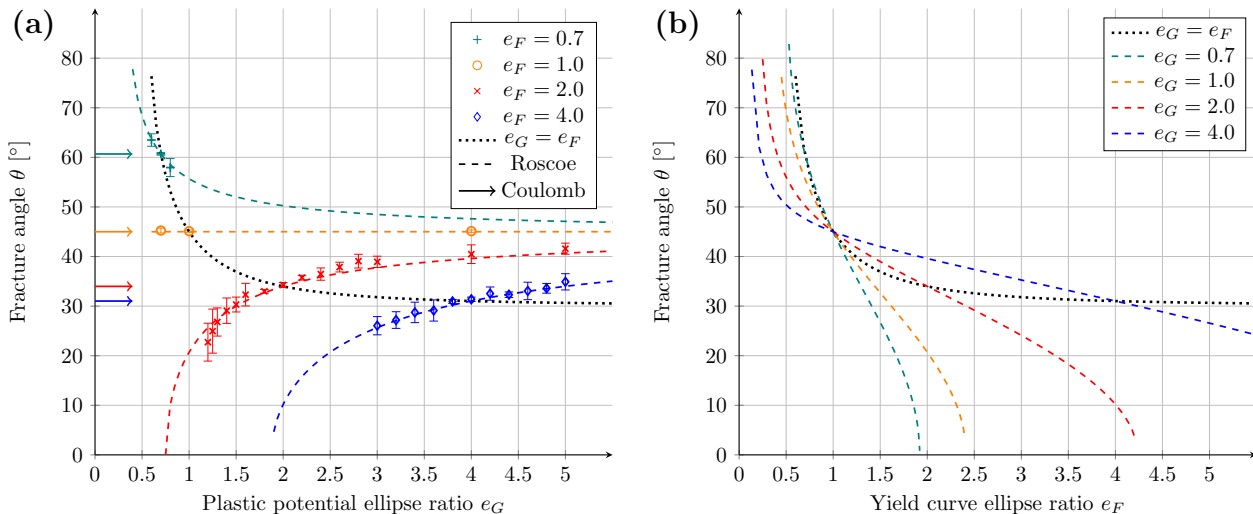

**Figure 7. (a)** Fracture angles as a function of the plastic potential ellipse ratio $e_G$ for different yield curve ellipse ratios ($e_F = 0.7, 1.0, 2.0$, and 4.0). The markers with ranges are the mean and two standard deviations of the fracture angles. The dashed lines show the Roscoe angle (Eq. 30). The arrows mark the Coulomb angles as a function of $e_F$, which are constant with respect to $e_G$. Colors indicate the value of $e_F$ for lines and markers. The $r^2$ between theory and modeled angles for $e_F = 0.7, 2.0$, and 4.0 are 0.97, 0.95, and 0.97. **(b)** Roscoe fracture angle computed from Eq. (30) as a function of $e_F$ with a constant $e_G$, for illustration. As $e_F$ changes, both the stress state and the flow rule change (see Fig. 4), resulting in a more complex behavior. The black dotted line for the normal flow rule ($e_F = e_G$) is drawn for reference.

are independent of $e_G$ because the fracture takes place at the peak of the yield curve and the flow rule is not affected by changes of the plastic potential ellipse ratio ($e_G$).

The coloured dashed lines in Fig. 7a show the fracture angles $\theta_{e,nn}(e_F, e_G)$ predicted by Eq. (30). The coefficient of determination $r^2$ and the root-mean-square error (RMSE) between the simulated angle of fracture and theoretical predictions are 0.97 and 0.37° for $e_F = 0.7$, 0.95 and 1.22° for $e_F = 2.0$, and 0.97 and 0.47° for $e_F = 4.0$. The RMSE is 0.37° for $e_F = 1.0$, $r^2$ being inapplicable. That is, the theory predicts the fracture angles accurately. This result shows that the flow rule plays a major role in the simulated fracture angle for a given rheology. The black dashed line show the change of the fracture angle 360    with a normal flow rule ($e_G = e_F$, Eq. 31).

     For completeness, Fig. 7b also show the theoretical predictions for a constant plastic potential ellipse ratio $e_G$ for varying yield curve ellipse ratios $e_F$. The fracture angles become smaller as $e_F$ increases. Yield curve ellipse ratio smaller than $e_F = 1$ do not create fracture angles below 45°.

## 5   Discussion

The idealized experiments using the elliptical yield curve with a non-normal flow rule confirm that the type of deformation and the fracture angle are intimately linked with the shape of the plastic potential. We observe that, irrespective of the plastic

potential elliptical aspect ratio, a yield curve ellipse ratio $e_F < 1$ does not allow fracture angles smaller than $45°$ in uni-axial compression. To reduce the fracture angles with yield curve ellipse ratios $e_F > 1$, plastic potential ellipse ratios $e_G$ smaller than the yield curve ellipse ratio are required, that is, $e_G < e_F$. The idealized experiments show that with a plastic potential in a viscous-plastic model we can separate the yield criterion from the resulting deformation (flow rule). This allows decoupling the mechanical strength properties of the material (ice) from its post-fracture behavior. The results illustrate clearly how the yield curve defines the stress for which the ice will deform, that is, the transition between viscous and plastic deformation, and how the relative shape of the plastic potential with respect to the yield curve defines both the fracture angle and the type of deformation (convergence or shear) along the fracture line. The resulting fracture angles are in excellent agreement with the Roscoe angle predictions (Roscoe, 1970).

Understanding the link between rheology and fracture angle is necessary for choosing or designing a rheology that is capable of reproducing the observed intersection angles between pairs of LKFs and consequently the emerging anisotropy. A independent plastic potential may resolve several inconsistencies of the standard elliptical yield curve with a normal flow rule (discussed in Ringeisen et al., 2019), namely:

1. In the standard VP model with an elliptical yield curve and normal flow rule, adding shear strength increases the fracture angle, in contradiction to granular matter theory (Coulomb, 1773). This behavior is linked to the specific shape of the elliptical yield curve with a maximum shear stress at $P/2$ and an ascending and a descending part. In principle, we can decrease the fracture angle with increasing shear strength ($e_F$ decreasing) by decreasing $e_G$, but only if $e_F > 1$, but then the flow rule is far from "normal", making the numerical convergence difficult.

2. Because of the elliptical shape of the yield curve, the angle of fracture in the standard VP model changes with confining pressure (Ringeisen et al., 2019, Sec. 3.2.2, Fig. 8) unlike laboratory experiments with granular materials (e.g., sand) where the fracture angle is only weakly sensitive to the confining pressure (Han and Drescher, 1993; Desrues and Hammad, 1989; Alshibli and Sture, 2000). This behavior cannot be eliminated with an elliptical plastic potential, as the normal stress along the LKFs increases with confining pressure and the flow rule changes from divergence to convergence as one passes the maximum shear stress at $P/2$ (Ringeisen et al., 2019). A different plastic potential function would change this behavior. However, this would make the model implementation and numerical convergence even more difficult. We note that a 3D granular material like sand cannot release stress by ridging as sea ice does. A 2D material, such as sea ice, can ridge and "escape to the 3rd dimension" after fracture. Therefore, we expect a change in the fracture angles at large confinement. Laboratory experiments show this behavior and yield stresses in sea ice change above a critical confinement ratio (Golding et al., 2010; Schulson, 2002). It is still not clear whether these results can be extrapolated to the modeling sea ice as a 2D medium at the geophysical scale, although several common features can be found (Schulson, 2002).

3. In the standard VP model with a normal flow rule, the divergence and convergence are set by the ellipse ratio of the yield curve, and thus by the relative amounts of compressive and shear stress. The plastic potential ellipse ratio $e_G$ changes the flow rule but does not change the sign of the divergence along the LKFs which is solely determined by the yield curve

ellipse ratio $e_F$. With the elliptical plastic potential, convergent motion remains convergent and only the ratio of shear to convergence changes. To change this behavior, a different shape of plastic potential is required, for example a teardrop plastic potential.

4. The fracture angles in the standard VP models are larger than observed. Using a non-normal flow rule allows us to change the fracture angle in uni-axial compression to values below $30°$. This is not possible with a normal flow rule (Ringeisen et al., 2019).

We discuss the elliptical yield curve here because it the most commonly used one and its behavior is better documented than any other model in use in the community. This provides a known reference for studying the use of non-associated flow rules. Our goal is to provide a reference for the future development of viscous-plastic rheologies with non-normal flow rules rather than suggest a new VP rheology. Alternatives to the elliptic yield curve have been used before; for instance, the Mohr–Coulomb, the Coulombic yield curve, or the teardrop yield curves (Figure 8). The concept of a plastic potential in conjunction with these yield curves may also prove useful in solving the issues described above. A detailed analysis of the simulations using the family of Mohr–Coulomb and Teardrop yield curves is beyond the scope of this work and will be presented in a subsequent study. Below, we use the experience from our simulations to infer how alternative yield curves may address deficiencies in the standard VP rheology.

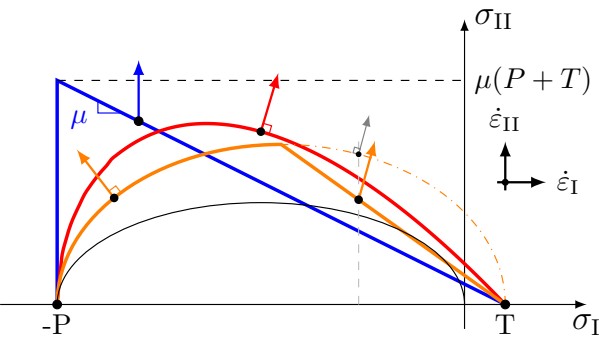

**Figure 8.** Alternative yield curves and flow rules: The Mohr–Coulomb yield curve with shear (a non-normal) flow rule (blue, Ip et al., 1991), the modified Coulombic yield curve with normal (elliptic part) and non-normal (linear part) flow rule (orange, Hibler and Schulson, 2000), and the teardrop yield curve with a normal flow rule (red, Zhang and Rothrock, 2005). The elliptical yield curve with $e_F = 2.0$ is shown for reference (black thin line). $P$ is the compressive ice strength, and $T$ the tensile ice strength.

Non-normal flow rules can be combined with the Mohr–Coulomb family of yield curve. For a Mohr—Coulomb yield curve with a double sliding law (i.e., pure shear deformation, Ip et al., 1991), the Roscoe theory predicts a fracture angle of approximately $45°$ that is independent of the slope of the yield curve. This behavior can be mimicked using an elliptical yield curve and plastic potential by setting $e_G \gg e_F$, hence $\delta \simeq 0$ and $\theta = 45°$ (Fig. 7a). This contradicts the Coulomb theory, which predicts an angle of fracture that depends exclusively on the internal angle of friction (Eq. 1). Combining an angle of dilatancy with

a Mohr–Coulomb yield curve (Tremblay and Mysak, 1997) would allow an angle of fracture depending on $\delta$ that is different with shear and divergence ($\delta > 0$) or with convergence ($\delta < 0$) along the LKFs. Such a fracture angle and divergence would be independent of the shear strength and the confining pressure in agreement with Roscoe's angle of fracture so that such a rheology could potentially solve all four issues on page 17. It is also important to note that the Mohr–Coulomb yield curves do not satisfy the convexity requirements of Drucker's stability postulate. Mohr–Coulomb yield curves in plastic Earth mantle models lead to a variety of fracture angles corresponding to the Coulomb angle, Roscoe angle, and the intermediate Arthur angles (Buiter et al., 2006; Kaus, 2010; Mancktelow, 2006). However, such geological models are designed for incompressible medium, and making inferences for the compressible formulation of sea ice models is difficult.

The Coulombic yield curve uses the two straight limbs from the Mohr–Coulomb yield curve and an elliptical cap of the standard VP rheology for large compressive stresses (Hibler and Schulson, 2000). In this rheology, the flow rule over the two straight limbs is defined by the elliptical yield curve; that is, the ellipse serves as a plastic potential for the Mohr–Coulomb yield curve. The Coulombic yield curve leads to unrealistic and asymmetrical fracture lines *(i)* when the stress states fall onto the non-differentiable intersection between the straight limbs and the elliptical cap (Ringeisen et al., 2019), and *(ii)* when the stress states fall onto the two straight limbs with the non-normal flow rule. Note that straight and symmetric fracture lines in this rheology are only possible when all the stress states are on the Mohr–Coulomb limbs and the flow rule at the fracture line is near-normal, that is, at the location where the normal to the elliptic plastic potential is nearly perpendicular to the limbs of the Mohr–Coulomb yield curve (Ringeisen et al., 2019). Hibler and Schulson (2000) already inferred that the flow rule may have an effect on the angle of fracture, but the authors limited their case to the framework of flawed ice and did not consider Roscoe's theory of dilatancy. The rheology of Hibler and Schulson (2000) was tested in an idealized experiment more complex than ours (Hutchings et al., 2005), but the effect of using a non-normal flow rule was not explored. The complexity of their setup may explain the observed difference between simulated and predicted angles. Note that the rheology in Hibler and Schulson (2000) was built by changing the shape of the yield curve *a-posteriori*, while the rheology presented here solves the constitutive equations rigorously.

The teardrop yield curve with a normal flow rule (Rothrock, 1975; Zhang and Rothrock, 2005) is divergent for a wide range of normal stresses and for all practical purposes consists of a continuously differentiable version of the Coulombic yield curve. This asymmetry between divergent and convergent deformation for different normal stresses decreases the effect of confinement on the fracture angle — issue 2 on page 17 — and reduces the fracture angle for any confinement pressure — issue 4. This yield curve does not address issue 1, because adding shear strength in the teardrop yield curve also increases the fracture angle.

As the main disadvantage of a non-normal flow rule we found that it leads to slower convergence of the numerical solver. Solving the momentum equation accurately requires more solver iterations and failure to converge is more frequent than for standard normal-flow-rule rheologies. In our simulations, this numerical issue manifests itself by the presence of multiple and asymmetrical fracture lines despite the fact that our experiment geometry and forcing are exactly symmetrical. This asymmetry is not expected, and is not found with normal flow rules. The fracture lines with a normal flow rule are symmetrical and come in pairs (Ringeisen et al., 2019). In practice, the numerical convergence issue will go unnoticed in simulations using

realistic geometries and time varying wind forcing. In these simulations, while the number of iterations typically used (O(10)) is much smaller than that required for full convergence, at each time-step, a new iteration typically uses the solution of the previous timestep as the initial estimate. With this, together with slowly varying forcing in space and time, the number of solver iterations per forcing cycle is large, in contrast to the fast changing forcing in this study (every timesteps). Whether this behavior (asymmetry and multiple fracture lines) will also be present in realistic simulation using spatially and temporally varying wind forcing remains to be tested.

The following criteria should be considered when building a new rheology: The spatial and temporal scaling of sea-ice deformation should agree with observations (Bouchat and Tremblay, 2017; Hutter et al., 2018); the flow rule should reproduce the correct divergence along LKFs (Stern et al., 1995); the yield curve should include some tensile strength (Coon et al., 2007) and be Coulombic in nature in agreement with observed internal stress invariants from ice stress buoys (Weiss and Schulson, 2009); the distribution of fracture angles should agree with observations (Marko and Thomson, 1977; Erlingsson, 1988; Cunningham et al., 1994; Hutter et al., 2019); the sea ice mechanical strength properties (i.e., yield curve) and deformation (i.e., flow rule for VP rheologies) should vary in time and space depending on additional variables or parameterizations, for example, the time-varying distribution of the contact normals (Balendran and Nemat-Nasser, 1993), floe size distributions (Horvat and Tziperman, 2017; Roach et al., 2018), or a damage parameter (Dansereau et al., 2016; Plante et al., 2020), as per observations and laboratory or numerical experiments (Overland et al., 1998; Hutter et al., 2019).

Although high spatial resolution observations from satellite are available from optical instruments (e.g., from the Landsat or Sentinel programs), higher temporal resolution of sea ice deformation and flow size distributions is still unavailable. The new Sentinel constellation and in-situ observations from the field program MOSAIC may bridge this gap. There is also a knowledge gap in the interplay between yield stresses and the post-fracture deformation in a 2D granular material such as sea ice. This interplay is likely different than for the well studied case of a solid homogeneous 3D block of ice (e.g. Schulson, 2002). Sea ice floating on the ocean surface can "escape" vertically when it forms ridges under confined compression (Hopkins, 1994). This behavior differs from laboratory test with 3D granular material like sand that use axial symmetry. Generally, information about sea ice resistance in different configurations (e.g., confinement) and the resulting fracture angles and deformation (ridging or opening) is also still missing, although some laboratory scale experimental results are available Weiss et al. (2007); Schulson et al. (2006b); Weiss and Schulson (2009). The sea ice flow size distribution varies in time (summer/winter) and space (marginal ice zone/central Arctic) (Rothrock and Thorndike, 1984). These variations change the mechanical properties (e.g., distribution of contact normals) and thermodynamic properties (e.g., lateral melt) of sea ice (Horvat and Tziperman, 2017). Designing more appropriate rheologies for improved high-resolution climate models and more accurate sea ice prediction systems requires consolidated observations of these still unclear physical processes.

## 6 Conclusions

The flow rule, which dictates the post-fracture deformation, has a fundamental effect on the orientation of fractures lines in a viscous-plastic (VP) sea ice model. To test this, we added an elliptical plastic potential (allowing for a non-normal flow rule)

to the standard VP rheology with an elliptical yield curve, therefore modifying the flow rule without changing the yielding stress state. We tested this new rheology with numerical experiments in uni-axial compression using the standard VP model of Hibler (1979). The modeled fracture angles are in agreement with the Roscoe angle, a theory based on experiments with granular materials that includes an angle of dilatancy (Roscoe, 1970; Tremblay and Mysak, 1997). This new rheology partially solves issues raised in an earlier study (Ringeisen et al., 2019). The use of a plastic potential or non-normal flow rule allows for the simulation of smaller fracture angles between pairs of Linear Kinematic Features, in agreement with satellite observations. Because of the elliptical yield curve, the fracture angles still depend on the confinement pressure (Ringeisen et al., 2019), and the elliptical plastic potential only modifies the ratio of divergence relative to shear, but not the direction of deformation at the fracture lines (convergence or divergence). The momentum equations for a rheological model with a non-normal flow rule are more difficult to solve numerically, and produce multiple lines of fractures that are asymmetrical (despite the geometrical symmetry of the problem), in contrast with a model with a normal flow rule. It is necessary to understand the effect of the flow rule on the fracture angle to design VP rheologies for high-resolution sea-ice modeling that both reproduce fracture angles and deformation along the fracture lines and the behavior of sea ice as a granular material.

Designing a rheology for high-resolution simulations requires information about sea ice fracture angles and sea ice strength in a wide range of stress conditions (i.e., compression with or without confinement, pure shear, tension), yet unavailable at high temporal and spatial resolution. The observations of the Multidisciplinary drifting Observatory for the Study of Arctic Climate (MOSAiC, Dethloff et al., 2016) in 2019/2020 may provide valuable data from continuous ice radar imaging, stress sensors, and arrays of drift buoys that will greatly help improve sea ice model dynamics.

*Code availability.* The modified version of MITgcm used in this study is available at https://github.com/dringeis/MITgcm/tree/ell_nnfr

*Author contributions.* DR designed the rheology and implemented the code changes with ML. DR ran the experiments. DR and BT designed the theory linking sea ice rheologies and granular matter theory. DR wrote the manuscript with contributions of BT and ML.

*Competing interests.* The authors declare that they have no conflict of interest.

*Acknowledgements.* The authors would like to thank Véronique Dansereau and Harry Heorton for their review and numerous comments that helped improve this manuscript. The authors also thank Jennifer Hutchings and Yevgeny Aksenov for their comments and involvements as editors. The authors are thankful to Mathieu Plante for his comments on this manuscript, as well as Stephanie Deboeuf and Guillaume Ovarlez for discussions on granular materials rheologies. This project was supported by the Deutsche Forschungsgemeinschaft (DFG) through the International Research Training Group "Processes and impacts of climate change in the North Atlantic Ocean and the Canadian Arctic"

(IRTG 1904 ArcTrain). This work is a contribution to Natural Sciences and Engineering Research Council Discovery Grant awarded to Tremblay.

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
