# Peer review of "Non-normal flow rules affect fracture angles in sea ice viscous-plastic rheologies"

_The Cryosphere, 2020_

## Referee Comment (RC1) · Véronique Dansereau (Referee) · 16 Sep 2020

This paper presents an implementation of a non-normal plastic flow rule in a Viscous-Plastic model with the goal of better representing the observed angles between Linear Kinematic Features in sea ice at the geophysical scale. The paper is overall well written, in a pedagogical way for the theory (section 2) section, which could however be a little more concise in some places. The figures are, for the most, very clear. Here are my major comments/concerns :

• It does not appear clear in the paper what physical process(es) the authors really want to model. In the introduction, it is mentionned that sea ice, both in the pack and the marginal ice zone, is considered as a granular material. No physical justification

is offered for this assumption. The rheology used to model this granular material is one of plastic flow, but the authors do not explain how they reconcile their continuum viscous-plastic model with a granular behavior. The aim is apparently to reproduce fracture angles (repeated terminology for the features simulated by their model), but the authors do not explain the link between plastic flow, fracturation and the mechanical behavior of a granular material, which is an already fractured/fragmented material in which contacts and friction dominate. Later, it seems that the authors refer to shear bands in granular materials as if they were associated with the same processes as a fracturing solid. The Coulomb theory is invoked but it is not clear if it is in the context of friction or fracture. There is therefore much confusion thoughout the paper as to what the authors consider is the mechanical behavior of sea ice : is it caracterized by fracturation? By friction and contacts between already broken up floes? Granular materials like sand are invoked, but is sea ice really assimilated to a sand-like material here? Whatever is assumed, it crucially need to be clarified and all physical concepts untangled throughout the text in a way that makes physical sense.

• In the same line of ideas, the authors seem to base their assumption of sea ice being a granular material on observations supporting fracture angles that are independant of confining pressure. It appears that they aim at developing a model that complies with these observations. However, no reference of observations, neither at the lab nor the geophysical scale, is clearly associated with this statement. One can reasonably wonder if making such observation would be possible in the case of sea ice at the geophysical scale: how would it be possible to determine far field stresses and distinguish between unconfined and confined states? Do unconfined compression leading to fracture even occur in circonstances other than an individual ice floe crashing into a coast? References are lacking here to support this assumption of independance on confinement and should crucially be added.

• Also somewhat contradictory is the fact that the authors use an elliptical yield curve and plastic potential to model a material that they consider as a granular. I understand

this is perhaps temporary and other criterion will eventually be investigated, but in the meantime, are there examples of granular materials that have been observed to follow this kind of yield curve/flow rule? References of such examples would strenghten the paper.

• Another concern is in the interpretation of the results. A model of plastic flow is used here, not a model of fracture (neither heterogeneities, nor elastic interactions, nor a mechanism representing breakage of bonds or damage is included here). In such model, one expects the simulated macroscopic behavior (that of the ice floe in this case) to coincide with the theory prescribed at the local scale, i.e., the constitutive equation, flow rule, etc. Therefore, as pointed out by Hutchings, Heil and Hibler, 2005, if deviations between the simulated angles and the predicted values occured, they would be indicative of numerical errors. Hence, while it is good to verify that the model does indeed reproduce the Roscoe angle within a small RMS error, doesn't it just show that the numerical scheme of the model works? This point needs to be clarified in the text. It would also be important to mention what method is used to estimate the angles from fields such as the ones shown on figure 6.

• Finally, I find that a discussion of previous studies that have presented similar interests and analyses is lacking from the discussion. Hibler and Schulson, 2000, have indeed implemented a non-normal flow rule in the VP model, using a Mohr-Coulomb yield curve with an elliptical cap (''modified Coulombic" curve). They have also found that a non-normal flow rule affects the orientation of deformation features in the VP rheology. This work is cited in the discussion section, but not really discussed in terms of the differences or similarities between both approaches, nor in terms of the advances of the present study compared to this previous one. I suggest clearly stating that is new here and what is the broad relevance of the results. The model of Hibler and Schulson, 2000 has also been used by Hutchings, Heil and Hibler, 2005 who have looked at intersection angles. They have compared simulated angles between the modified Coulombic and the elliptical yield curve. Mentionning these previous results

and comparing them with the current study would be interesting and would strengthen the litterature review and Discussion part of the paper. I therefore recommand major reviews to clarify the important points above before a resubmission. More specific comments that are often linked to these major comments are listed below.

Page 1, lines 8-9: "A newly adapted theory (...) predicts numerical simulations of the fracture angles (...) with a root-mean-square error below 1.3 degrees." This formulation is unclear and needs rephrasing: a newly adapted theory is implemented in the VP model and leads to prediction of the prescribed fracture angle with a RMS error below 1.3 degrees"?. Also, se my main comment about the agreement of the theory with your modeled angles.

Page 1, line 11: I suggest dropping "In conclusion" from your abstract.

Page 1, lines 14-15: "to make the fracture angle independant of (not on) the confining pressure (as in observations). This relates to another of my main comments : what sea ice observations support that fracture angles are independant of the confining pressure? Please give supporting references. Is it even possible to distinguish between fracturing processes ocurring in confined and unconfined conditions in the sea ice cover at the geophysical scale?

Page 1, lines 19-20: "narrow lines of deformation observed in the Arctic sea ice cover, emerge in high-resolution simulations (Kwok, 2001; Hutchings et al., 2005)". It would be relevant to cite more up-to-date works on high-resolution simulations here.

Page 2, line 23 : "The ice strenght locally depends on the ice thickness". This is only partially true: local ice strenght does not depend only on local ice thickness. This sentence perhaps needs some rephrasing.

Page 2, lines 25-27: "In granular media like sea ice (...) Note, that in this study, we consider sea ice to be granular not only in the marginal ice zone, but also in pack ice, where ice floes are densely packed". This again one of my major concern: what is the

basis for this assumption? How do you reconcile this assumption with the fact that your goal is to reproduce fracture angles in sea ice? Does pack ice, newly-formed ice or any ice that is not yet fractured into floes or constituted of agglomerated, refrozen floes always present the characteristics of a granular media? Please explain and also give some support for this assumption.

Page 2, line 28: "This anisotropy". This is unclear. Please define this anisotropy and better explain how it emerges.

Page 2, line 37: The brittle model used in Rampal et al., 2016 is the EB model of Girard et al., 2011. Please modify the reference.

Page 2, line 39: I believe a simpler and scientifically more objective formulation would be "most widely used", instead of de facto standard.

Page 2, lines 48-49: Yes, granular media indeed present shear bands, which are not the same as fractures. Again, please clarify what you want to represent in your model. What is the link between LKFs in sea ice, shear bands in granular media and fractures in solid materials?

Page 2, lines 48-49 vs line 50: "Two classical solutions coexist and set two limit angles for the orientation of fractures: the Coulomb angle (...)". There is something unclear and contradictory between this and the previous sentence. You invoque the Coulomb theory here, in the context of friction or fracturing? I understand it is the later, but please make that clear by answering my previous comment.

Page 3, line 56: I think it would be relevant to make some space and re-introduce the definition of the dilatancy angle here : it would make life easier for the reader and avoid the need to dig for it in another article.

Page 3, line 58: "A general theory derived from experiments with sand that takes into account both the angle of friction (...)". In the case of sand, contact and friction are indeed at play and shear bands are formed. This again adds to the confusion: internal

angle of friction or angle of friction? i.e., fracture or friction? Please clarify.

Page 3, line 60: based on the grain size.

Page 3, lines 67-68: ''a larger dilatancy angle implies a larger grain size, more contact normals, hence more friction''. Can you please include some references that support this?

Page 3, line 73: There is a mistake here, as Weiss and Schulson, 2009 reported observed fracture angles between 20 and 50 degrees. Or did you derived this directly from their estimated internal friction angle, which is fitted to in-situ stress measurements? In the later case, this is then not an observation of fracture angles but a derivation based on some physical assumptions, which are moreover debatable (see Dansereau et al., 2019 and many others), and it should be removed from the list of observations of fracture angles.

Page 3, lines 74-76: You state that uni-axial compression experiments showed that (3) the fracture angle is a function of the confining pressure. How did you determine that without performing bi-axial compression experiments? Is there a typo here?

Page 3, line 75: the ''gradient'' of shear to compressive strenght. Did you mean the ratio?

Page 3, line 76-79: See again my major comment about the apparent confusion between fracturing, friction, granular media, sea ice and a viscous-plastic continuum rheology. I think it is crucial to clarify the links you make between these processes and the motivation of your approach here. This passage in particular leads the reader to believe that your goal is that the VP rheology complies with observations of granular media behavior, because you consider that sea ice at the geophysical scale, in all its different states, is a granular media. If this assumption is at the very basis of your approach, it should be stated earlier in the introduction, (very importantly) along with supporting arguments. This would make the reading and the assesment of your assumptions and

methods by the reader much easier.

Page 3, line 82: "The ratio of shear and divergence along the LKFs allows to infer the dilatancy angle." Again, if one assumes sea ice in any state behaves as a granular material.

Page 3, lines 84-85: "Separating the link between the fracture angle and the flow rule from the yield curve is necessary to design rheologies that are consistent with observed sea ice deformations". Please note that this would be only true for plastic flow rheologies and not applicable nor necessary for rheologies based on elasticity (EB, MEB, Elastic-Decohesive). To be objective, this statement should therefore be modified as "necessary to design plastic flow rheologies that are consistent (...)".

Page 4, line 90: ''In these different classses of models, various rheologies can be defined". This is not true and/or not clear: these are rheological models and therefore they do not include different rheologies. I think that you mean that these different models require the definition of different components: a constitutive relation (all models), a yield/damage curve/criterion (all models including a threshold mechanism, i.e, a change in mechanical behavior) and a flow rule (only plastic flow models). I therefore suggest to rephrase and clarify this passage and the next sentence, that is "in a VP rheology, a yield curve and plastic potential (flow rule) must be defined". In the same line of idea, I do not really see the point of the last sentence of this paragraph. Maybe it can be cut if some rephrasing is made at the beginning of the paragraph?

Page 4, lines 96-97: See my major comment above. Hibler and Schulson, 2000 have indeed used a VP model with a non-normal flow rule and a Mohr-Coulomb yield curve with elliptical cap, or ''modified Coulombic" curve, as cited in your Discussion section. This model has also been used by Hutchings, Heil and Hibler, 2005 (https://doi.org/10.1175/MWR3045.1) who have looked at intersection angles and compared them between the modified Coulombic and the elliptical yield curve. As their approach is therefore close to yours, it would be important and certainly interesting to

explain the similarities and difference between your work and theirs in the litterature review (introduction) section. Please also note that Hibler and Schulson, 2000 do not seem to share your view that the angles of fracture in sea ice at the geophysical scale are independant of confinement, which would be an important point to discuss further.

Page 4, line 100: "viscous-plastic materials" or "a viscous-plastic material", "with any flow rules".

Page 4, line 100: "from the yield curve".

Page 4, lines 101-102: "The new model is tested in simple uni-axial loading experiments". See my major comment above: a quick addition to your work would be to test if your numerical implementation also holds under bi-axial loading conditions, that is, if the angles vary or not with confinement.

Page 4, line 108: ''We consider sea ice as a 2D viscous-plastic material". See my previous major comment: please explain the physical link between this viscous-plastic assumption and that of a granular material.

Page 4, line 113: In your case, the constitutive equation links the vertically integrated stress tensor to the deformation rate, which you introduced on the previous line.

Page 4, lines 17-19: Representing small deformations with a viscous model is rather counter-intuitive, especially for a reader that is familiar with viscous-plastic rheologies (plastic for small, viscous for large deformations). I believe it is important that you explain in more details how a viscous rheology is expected here to represent the small deformations of a solid (time scales, viscosities, etc).

Page 5, line 130 to page 6, line 149: These paragraphs could be shortened by removing or presenting in a more concise manner some general pieces of information.

Page 5, lines 130-131: As it is not the states of stress that are deforming plastically, but the material, this sentence needs some reformulation.

Page 9, line 204: "The slope of the yield curve". And many other missing "the" throughout the text.

Page 10, line 223: How does the no-slip condition at the bottom boundary affect your results compared to the case in which slip is allowed in the x-direction (i.e., by holding only one of the two bottom corners of the domain fixed in x and y)? Such boundary conditions are maybe less representative of a floe that sticks to a coast but would not lead to as much concentration of stresses on the bottom corners of your ice floe (here your Bcs imply some bi-axial compression at the bottom) and hence would put less contraint on the appearance of conjugate faults and on their orientation. I think this would be an interesting and not time-consuming test.

Page 11, line 240: I suggest "more numerically challenging".

Page 11, line 256: ''laboratory experiments". If you compare your results with laboratory experiments, please provide more details on these experiments (e.g., boundary conditions? biaxial or uni-axial compression? on samples with an aspect ratio similar to sea ice, i.e., virtually 2D? on fresh or sea ice?) Were such experiments made by Erlingsson et al., 1988 and Wilchinsky et al., 2010?

Pages 11-13 and caption of figure 6: What is the field represented in figure 6? I assume from the color scale that it is a deformation rate?

Section 4 and figures 6 and 7: How are the angles of the features observed on fields such as shown on figure 6 measured, i.e., estimated? It would be important to mention what method is used.

Result section, figure 7 and page 15, lines 292 and 306-308: "the theory predicts the fracture angles accurately" and "The results illustrate clearly how the yield curve defines the stress for which the ice will deform, that is, the transition between viscous and plastic deformation, and how the relative shape of the plastic potential with respect to the yield curve defines both the type of deformation (convergence or shear) along

the fracture line and the fracture angle. The resulting fracture angles are in excellent agreement with the Roscoe angle predictions (Roscoe, 1970)." There is my major comment about the results. In section 2.3, you describe how the yield curve, flow rule and angles are related in your model. By prescribing the yield curve and plastic potential ellipse ratios, you prescribe locally the angle (Roscoe) of "fractures". Figure 7 shows that at the macro-scale, i.e., the scale of the ice floe you indeed retrieve that angle. What is prescribed at the local scale is what you get at the macro-scale in your model, as expected in a model of plastic flow. Therefore my understanding is that these tests serve to verify that your numerical scheme is OK. Is that the case? To better illustrate that point, it would be relevant to show the (deformation?) fields at different stages of the compression experiment, to illustrate how the features arise in your model.

Page 15, line 300 : "the shape of the plastic potential".

Page 15, line 305 : "this allows decoupling the mechanical strength properties of the material (ice) from its post-fracture behavior". Again the contradiction with the assumption of a granular material, i.e., an already fractured/fragmented material. How do you reconcile these ideas?

Page 15, lines 306-308: "The results illustrate clearly how the yield curve defines the stress for which the ice will deform, that is, the transition between viscous and plastic deformation, and how the relative shape of the plastic potential with respect to the yield curve defines both the type of deformation (convergence or shear) along the fracture line and the fracture angle. The resulting fracture angles are in excellent agreement with the Roscoe angle predictions (Roscoe, 1970)." But you prescribe the yield and plastic potential in your model: why would you not expect what you get to indeed be what you prescribe? In other words, you do not make any distinction between what you prescribe at the micro-scale (scale of your discretization) in your model and your macroscale results and you do not discuss why you expect these behavior to be identical or not : that is missing from your work and interpretation of your continuum

model.

Page 15, point 2: About confinement, shear bands and fractures, see my major comment above.

Page 17, line 382: ''sea ice mechanical strenght properties (yield curve) and deformation (flow rule)''. Again, you write this with the perspective of a VP model, but mechanical strenght properties and deformation are not only determined by the yield criterion and flow rules in other rheological models for sea ice. Please be specific and make this distinction clear. Also, I do not undestand why Dansereau et al., 2016 is cited in this context.

Page 17, lines 387-388: ''So is the combined knowledge of the failure stresses and their associated deformation of sea ice as a 2D granular material''. This is confusing: why then do you base your approach on the assumption of a granular material? This goes along my main comment and really needs to be clarified.

Please also note the supplement to this comment:
https://tc.copernicus.org/preprints/tc-2020-153/tc-2020-153-RC1-supplement.pdf

---

## Referee Comment (RC2) · Harry Heorton (Referee) · 29 Sep 2020

This paper describes the implementation of a non-normal flow rule in the VP sea ice rheology. The equational form of the new rheology is well described and several very useful diagrams are included. The numerical implementation is linked to a theory that links the flow rule and the intersection of failure lines within the medium described. A series of idealised numerical experiments are performed which show that the numerical rheology successfully recreates the fracture intersection angles predicted by the presented theory. The authors follow the experiments with a discussion on the implications of using a non-normal flow rule when designing future sea ice rhelogies. They describe the various challenges when using non-normal flow rules. I find that this paper is well written and a valuable contribution to the modelling of sea ice deformation. It is a very useful introduction to use of non-normal flow rules for sea ice modelling for future work in this area. I recommend this paper for publication after a few questions I have.

First of all can you explain why figure 7a contains both theoretical links between the plastic potential and intersection angle and many numerical experiments that back up the theory but 7b contains relatively few numerical results? I can see several cases where additional results from 7a can be copied to 7b and back up your results. Is it true that the full range of values for 7b are not obtainable due difficulties that the authors discuss in getting the model to converge to a solution for highly non-normal flow? If this is case then please tell us.

Several times in the discussion and results  the authors say that the intersection angle depends on the confining pressure despite the varying non-normal flow rule. I can see no evidence of this in their results. The presented experiments show changing intersection angle with changing flow-rule (varying plastic potential and yield curve eccentricity), but I see no results where they change the confining pressure. Is this from previous work? Or an interpretation of the results that they do present?

General editing points:

Can you please start the paper with a description of what a flow rule is. Then what a normal flow rule is, and the crucially what the main difference physically and theoretically is between a normal and non-normal flow rule. I see that a definition is on line 90, and then further physical descriptions of the flow rule are in the results. The introduction make much more sense if these can come first.

Can you describe what is documented in this study that is novel and new?

L 20 they are also, more importantly, observed

L 21 Here you LKF's influence in many ways but what follows is not a list. Consider re-writing

L 22 Please define what a lead is. Consider starting  with a definition of LKF's that are typically leads or ridges

L 70 Which is the 'standard rheology'? do you mean the VP rheology. Also can you further describe this result. How did Ringeisen find that the angle can't be lower than 30 degrees?

L71 the following list is hard to read. Consider reformatting. Also what does the μ = 0.9, relate to with the Weiss and Schulson reference.
L71 can you confirm that these angles are all comparable? I have found that studies document both the intersection and also the half angle, being the intersection between the fracture and the principal axis of stress.

L80 this paper require a definition for a normal flow rule. This sentence and the following paragraph make little sense without it.

L82 do you mean that the flow rule can be observed by measuring the ratio of shear a divergence along LKF.

L85 were these laboratory observations performed the same way as those of Stern mentioned above?

L89 it will be nice to have the Anisotropic Plastic (Tsamados 2013, 10.1029/2012JC007990) rheology listed here too

L92. Good to see a flow-rule definition here. How does the plastic potential determine the post-fracture deformation? is this through the direction of the principal stress when the yield criterion is reached?

L 115 is f here the Coriolis acceleration as above? Actually can you tell what value was used for the Coriolis acceleration? If it is non-zero (valid to use zero and non-zero for these experiments) then asymmetry will be expected (see comments later)

L120 It is great to read this description of the VP rheology. A really helpful addition.

L 138 is it possible to have a physical description of the plastic potential here? The physical description of what the yield curve represents is very helpful. A similar description of the plastic potential here will be similarly useful. The flow-rule is difficult concept that is explained well here. An additional physical description will make it even better.

L 180 I see that the dilatancy angle was introduced earlier. However it would benefit the paper to include a physical description of 'dilatancy of a granular material' either before or here when it is implemented in the model equations.

L 180 and onwards. This section will benefit from an expanded introduction to the theoretical steps performed. From what I can tell, you use the theory that links dilatancy angle to fracture angle as discussed in the the introduction. You have quantified the dilatancy angle using geometrical description of an arbitrary yield curve and plastic potential. This is expanded through the notation to express the fracture angle as a function of yield curve and plastic potential eccentricity. Is this correct?
If so is the motivation behind the description that it is possible to show how the expected fracture angle is expected to change with changing plastic potential?
Can you be clear what the theory of Roscoe is describing. Is the angle you are obtaining the expected angle of fracture due to minimising some sort of energy potential? Or does it relate to an analytical solution of fracture? The mathematical expansion here is clear to follow, but the reasoning behind why you have shown it is less so.
In figure 4 you describe how the ratio of divergence to shear changes with changing plastic potential. Is this the key effect of the non-normal flow rule? In that by separating the yield curve and plastic potential it is possible to change the ratio of divergent to shear stresses whilst under deformation? But without also change the point of deformation (as in the yield curve) If so please emphasise this point throughout the paper! It makes the non-normal flow rule much clearer for me!

Figure 3 caption - the arrows are described as orange, but appear red to me.

Figure 4. I see red and orange arrows here, and they are correctly described. Can you check figure 3. Do the colours relate between the two figures?

L 222 is the initial ice state entirely uniform? Or did you seed some noise into the initial state?
L 231 did you test at other time and spatial resolutions? Later you comment that fracture angles were shown in a previous study to be independent of model resolution (we found this too). Did you test this for this study too?
L232 is this equation 4 that is solved for?

L233. What are the non-linear and linear problems ? Can you relate these back to the model equations?

L246 So are the simulations only run for 5 seconds of model time? Have you tested how long the model can run for and its overall stability? I read above that you have used excessive computation to ensure the extra complexity of the non-normal flow rule is accounted for. How successful is this approach? Did you find that certain computational setups did not perform well when attempting to solve the equations? Any insight you can share into how to solve these equations will greatly help the sea ice modelling community

L 263 what is average residual norm R? is this a measure of the solution accuracy?

L 282 is the shear strain rate shown anywhere? Are you relating back to figure 6? If so can you say so? Are you saying the relation ship in figure 6 for eF and shear strain rate is also true for the various values of eF in Figure 7? Or is this a theoretical postulation?

L 282 fracture angle or angles plural? Do you you take multiple angles or just one per simulation?

Figure 7 Is it possible to add the red orange and teal umerical simulations to figure 7 b? If you have added the blue dots then the omission of the others makes me wonder how they will fit? I see that you only have multiple values for eG = 4.0. Though there are 2 points for 0.7 and a single point for 2.0 and 1.0. I also see that the full range of eF was not investigated for each eG. What is the reason for this? Is it the limitations of the model? Or did you choose not to in order to keep the simulations physically relevant?

L 305 this line is very informative to what the non-normal flow rule can achieve. Can you put this information into the introduction and abstract please?

L 309 while you have displayed the agreement to Roscoe for the cases of constant eF the case of constant eG (fig 7b) is inconclusive to the reader due to the lack of numerical simulation data points. Is it possible to fill out figure 7b and thus strengthen this statement?

L 313 Can you sort out the parenthesis on the Ringeisen 2019 citation. It currently doesn't read very well.

L 317 is this lack of convergence the reason for the lack of results on figure 7b?

L 319 Can you give a citation a description of how this result with the changing fracture angle with changing stress confinement was obtained? I assume it is not from this study as you have not altered the confinement ratio for any of your simulations. Or are you referring to that the fracture angles change as the loading increases with time?

L 321 How do think this result relates to to laboratory experiments on sea ice where two clear fracture angles were found about a critical confinement ratio? (Golding et al. 2010 1359-6454/$36.00, Schulson 2001 10.2138/gsrmg.51.1.201)

L 341 Is this result about pure shear and angle of 45deg. from the Ip et al. 1191 citation? How was it obtained?

L345 angle - angles

L 363 Is it possible to include a diagram of the various yield curves discussed in this section? This would greatly ease the understanding of your arguments. I'm sure others have included such a diagram in previous work so you may be able to cite such a diagram.

L369 Can you explain why non symmetrical deformation features are unrealistic or present an incorrect solution? Do they also correspond to poor numerical solutions? With a non-linear system of equations such as in all sea ice rheologies, asymmetry is often expected. This relates back to most laboratory experiments on ice deformation and even the ill-posedness of divergent weakening (Gray 1999 10.1175/1520-0485(1999)029<2920:LOHAIP>2.0.CO;2). Also if you use a

non-zero Coriolis acceleration then asymmetry will be expected as the run progresses. What value did you use?

L371 I'm not sure I understand your argument here. Are you saying; poor non-normal flow model convergence won't be an issue in realistic simulations as the numerical solver can't solve the VP rheology anyway? Surely this argument says that there isn't a hope of using non-normal flow VP rheology in realistic simulations?

L396 These issues are not exclusive to high resolution climate modelling. It can be argued they are even more important for current coarse resolution models which are currently used for long climate simulations and typically perform poorly for reproducing ice drift patterns. LKF intersection angles are also observed over basin length scales (Weiss and Schulson 2009) and your discussion in this paper is relevant for modelling sea ice deformation at these length scales.

L406 I am confused by your conclusion here. Where have you shown that the fracture angles depend on the confinement pressure? Where did you change the confinement pressure? Do not Figure 6 and 7 show clear changes in intersection angle with changing plastic potential in accordance with predictions from the theory of Roscoe?

L 409 again I'm not convinced that symmetric solutions are mandatory for a symmetric experiment? Again can you say whether you used a zero or non-zero value for the Coriolis acceleration? If it is non-zero then asymmetry will be expected.

---

## Author Comment (AC1) · 26 Nov 2020

**Authors answer to Véronique Dansereau for tc-2020-153**

November 26, 2020

**Note:**

- The referees comments are shown in black.

- The authors answers are shown in blue.

- **The proposed modifications for the manuscript are shown in bold typeface and colored in gray.**

- **Because we do multiple referenses to the comments of the other referee, the answers to their comments are also presented at the end of this document.**

**R1#1,** This paper presents an implementation of a non-normal plastic flow rule in a Viscous-Plastic model with the goal of better representing the observed angles between Linear Kinematic Features in sea ice at the geophysical scale. The paper is overall well written, in a pedagogical way for the theory (section 2) section, which could however be a little more concise in some places. The figures are, for the most, clear.

We thanks the reviewer for her thorough review of our manuscript. Her comments will improve the clarity and quality of our manuscript. We hope that we address all comments in a satisfactory fashion.

Here are my major comments/concerns :

- **R1#2,** It does not appear clear in the paper what physical process(es) the authors really want to model. In the introduction, it is mentioned that sea ice, both in the pack and the marginal ice zone, is considered as a granular material. No physical justification is offered for this assumption. The rheology used to model this granular material is one of plastic flow, but the authors do not explain how they reconcile their continuum viscous-plastic model with a granular behavior. The aim is apparently to reproduce fracture angles (repeated terminology for the features simulated by their model), but the authors do not explain the link between plastic flow, fracturation and the mechanical behavior of a granular material, which is an already fractured/fragmented material in which contacts and friction dominate. Later, it seems that the authors refer to shear bands in granular materials as if they were associated with the same processes as a fracturing solid. The Coulomb theory is invoked but it is not clear if it is in the context of friction or fracture. There is therefore much confusion throughout the paper as to what the authors consider is the mechanical behavior of sea ice : is it characterized by fracturation? By friction and contacts between already broken up floes? Granular materials like sand are invoked, but is sea ice really assimilated to a sand-like material here? Whatever is assumed, it

crucially need to be clarified and all physical concepts untangled throughout the text in a way that makes physical sense.

- Sea ice is composed to individual floes that vary in size and thickness along seasons and conditions. Sea ice has often been described as a granular material (Overland et al., 1998; Mcnutt and Overland, 2003; Tremblay and Mysak, 1997). In other fields, granular material has been modeled with continuum plastic flow models, considering both the Coulomb theory or the Roscoe theory (Vermeer and De Borst, 1984; Vermeer, 1990; Balendran and Nemat-Nasser, 1993; Mánica et al., 2018).

- We think that we need to consider the ice as a granular material if we want to explain divergence along fracture lines (Stern et al., 1995; Bouchat and Tremblay, 2017). The fact that the elliptical yield curve with normal flow rule (Hibler, 1979) feature compressive states with divergent opening (also when low confinement is applied) (Ringeisen et al., 2019) shows that we can consider granular dynamics to already be present in current VP models. In this manuscript, we investigate a modification of the VP model with elliptical yield curve.

- We do not consider sea ice to behave like sand, but still as a granular material: a 2D granular material. Sea ice is peculiar in the world of physics, because (1) it is bound to the 2D ocean-atmosphere interface by gravity, but can "escape in the vertical dimension" (page 17, line 389) and ridge when bi-axial compression exceeds a critical threshold. Also ice floes, the "*grains*" of sea ice, can brake or refreeze. Therefore, sea ice dynamics exhibits a large spectrum behaviors, including characteristic granular dynamics, for example dilatancy, as well as brittle behavior.

- The terms referring to brittle behavior, such as *fracture angle* or *fracture lines*, might be slightly confusing with the idea of sea ice as a granular material, but we would like to keep them as it is. Here is our reflection:

  * If we agree on the fact that sea ice is already a fractured medium, we study the large scale deformation of a compact ice field, process similar to the creation of fracture in continuous solid.

  * In that case, it makes little sense to us to make a distinction between fracture and friction. This is well described in the abstract of (Wilchinsky and Feltham, 2011): "*Sea ice failure under low-confinement compression is modeled with a linear Coulombic criterion that can describe either fractural failure or frictional granular yield along slip lines.*" The assemblage breaks and floes interact with one another, which can be seen as the microscopic behavior of friction.

  * Furthermore, the creation of LKFs in sea ice was already associated with breaking behavior (Erlingsson, 1991; Marko and Thomson, 1977), the term fracture is repetitively used (Hutchings et al., 2005; Hibler and Schulson, 2000), as well as the fact sea ice is granular medium (Wilchinsky and Feltham, 2011; Hopkins, 1996).

  * Furthermore, for clarity, we would like to keep the same terminology as in the Ringeisen et al. (2019), on which this study is based.

In order to address these points, the following sentences were added:

- "*Note, that in this study, we consider sea ice to be of granular nature not only in the marginal ice zone, but also in pack ice, where ice floes are densely packed. Because sea ice floes are densely packed, we can*

*consider the creation of an LKFs as a fracture process with both fracture and friction (Wilchinsky and Feltham, 2011)."* L26 of the original manuscript.

- We modify the penultimate paragraph of the introduction (see also comment R2#4). It now reads "*In this paper, we investigate the effects of a non-normal flow rule on fracture angles and its use as a means of separating the state of stress (at failure) and the post-fracture deformation. The novelty of this paper is that we study the non-normal flow rule in the context of the standard VP rheological model using a similar shape for the plastic potential (i.e., an ellipse) because (1) it is widely used in the community, and (2) its behavior is well documented (compared to other models), providing a solid basis for comparison. This paper provides a new generalized theoretical framework for developing any viscous-plastic material with normal or non-normal flow rules. To this end, we test the new model in simple uni-axial loading experiments where the relationship between fracture angle and flow-rule can be easily identified."*

- **R1#3,** In the same line of ideas, the authors seem to base their assumption of sea ice being a granular material on observations supporting fracture angles that are independent of confining pressure. It appears that they aim at developing a model that complies with these observations. However, no reference of observations, neither at the lab nor the geophysical scale, is clearly associated with this statement. One can reasonably wonder if making such observation would be possible in the case of sea ice at the geophysical scale: how would it be possible to determine far field stresses and distinguish between unconfined and confined states? Do unconfined compression leading to fracture even occur in circumstances other than an individual ice floe crashing into a coast? References are lacking here to support this assumption of independence of confinement and should crucially be added.

  Concerning the granular matter behavior:

  - Fracture angles (or orientation of the shear bands) that are independent of the confinement pressure are characteristics of granular material, and lead to the use of the Mohr–Coulomb yield criterion.

  - More recent studies showed that shear bands orientations in granular materials increase slightly with confining pressure (Alshibli and Sture, 2000; Han and Drescher, 1993; Desrues and Hammad, 1989, Note that some of these studies show a decrease, but only because they use the complementary angles.). However, this change is very limited: of the order of 5°, with a stress confinement ratio of in the range [0.05-0.5] depending on the confining pressure and the grain size.

  - The magnitude of the change of angle contrasts with the effect of confining pressure with the elliptical yield curve, where a stress-ratio of 0.3 changes the fracture from divergent to convergent and the fracture angle from ca. 34° to 46°.

  Concerning the sea ice behavior:

  - The observations of the same fracture angles at different scale (so probably different stress conditions) by several studies (Erlingsson, 1988; Marko and Thomson, 1977;

Cunningham et al., 1994) is an indication that fracture angles might be independent of the stress conditions, i.e. different confining pressures. New datasets of intersection angles from LKFs tracking show that coulombic fracture in the Arctic sea ice shows a predominant angle (Nils Hutter, personal communications)

- It is correct that, at high confining pressure, the fracture angle probably changes, especially when sea ice reaches a ridging state. This can be seen with the shape of the yield curve observed in Schulson (2004); Weiss and Schulson (2009). Please see also our answer to Reviewer#̃2 in comment R2#40.
- See also our answer to comment R2#39 of Reviewer#̃2.
- Finally, we agree that far field stresses are difficult (or close to impossible) to determine, this is why observing the angle of dilatancy along LKFs could be a good metric to improve sea ice models.

To clarify our manuscript, we make the following modifications:

- We modify our statement: "*. . . namely that shear band orientations and divergent/convergent motion at the slip lines are a function mainly of the shear strength of the material and orientation of the contact normals (or dilatation angle), and that the confining pressure has only limited impact (Alshibli and Sture, 2000; Han and Drescher, 1993; Desrues and Hammad, 1989)*", L79 of the original manuscript.
- The sentence on L321 would now reads "*. . . unlike laboratory experiments with granular materials (e.g. sand) where the fracture angle is weakly sensitive to the confining pressure (Han and Drescher, 1993; Desrues and Hammad, 1989; Alshibli and Sture, 2000).*".
- We add the following statement: "*. . . A 2D material, such as sea ice, can ridge and "escape to the 3rd dimension" after fracture, therefore we expect a change in the fracture angles at large confinement, when the ice is susceptible to ridging (Schulson et al., 2006).*" L325 of the original manuscript.

- **R1#4,** Also somewhat contradictory is the fact that the authors use an elliptical yield curve and plastic potential to model a material that they consider as a granular. I understand this is perhaps temporary and other criterion will eventually be investigated, but in the meantime, are there examples of granular materials that have been observed to follow this kind of yield curve/flow rule? References of such examples would strengthen the paper.

  - As the reviewer stated, the use of elliptical yield curve is transitory, but practical for the main goal of this study: that is, studying the effect of a non-normal flow rule on the angles of fractures, and provide an theoretical explanation for this effect.
  - We use an elliptical yield curve in this study for 2 reasons: (1) Because it is widely used in the sea ice community, for instance 30 out of 34 sea ice models in GCMs participating in CMIP5 use the standard VP model or a modification thereof (Stroeve et al., 2014), and (2) because the behavior of the elliptical yield curve with normal flow rule in uni-axial compression has been recently investigated (Ringeisen et al., 2019), and we want to isolate the effects of using a non-normal flow rule.

- Elliptical yield curve, like the *Von Mises* yield curve, are used in material modeling, especially for ductile materials. Although their formulation is different that of in the sea ice models. Granular materials usually use an incompressible formulation, while sea ice needs a non-zero divergence term to represent open water formation and ridging.

To clarify our manuscript, we make the following modifications:

- *"We use the elliptical yield curve because it is widely used and its behavior better documented than any other models in use in the community. This provides a known reference to study the use of non-associative flow rule. We do no aim to propose here a new VP rheology here but to study the effect of the non-normal flow rule as it could be used in future rheology."* on L335 of the original manuscript.

- **R1#5,** Another concern is in the interpretation of the results. A model of plastic flow is used here, not a model of fracture (neither heterogeneities, nor elastic interactions, nor a mechanism representing breakage of bonds or damage is included here). In such model, one expects the simulated macroscopic behavior (that of the ice floe in this case) to coincide with the theory prescribed at the local scale, i.e., the constitutive equation, flow rule, etc. Therefore, as pointed out by Hutchings et al. (2005), if deviations between the simulated angles and the predicted values occurred, they would be indicative of numerical errors. Hence, while it is good to verify that the model does indeed reproduce the Roscoe angle within a small RMS error, doesn't it just show that the numerical scheme of the model works? This point needs to be clarified in the text. It would also be important to mention what method is used to estimate the angles from fields such as the ones shown on figure 6.

  - In sea ice VP rheology, the angle of fracture is not yet understood. For instance, Roscoe and Coulomb theories gives different angles for the same process. We show here that the flow rule affects the fracture angles, and we explain this influence with a theoretical model, adapted from the Roscoe angle. Similar investigations of the angle of deformation features can be found, for example, in the field of lithosphere geophysical modeling: Lemiale et al. (2008); Kaus (2010).
  - The method used to estimate the angles is presented at the end of Sec. 3.

To clarify our manuscript, we make the following modifications:

- We add on L69 of the original manuscript: *"In the case of a non-normal flow rule, it is unclear which of the three theory (Coulomb, Roscoe, Arthur) predicts the modeled angle of fracture."*
- For comparison and clarity, we add the Coulomb angles predictions on a new version of Fig. 7a, shown below (Figure 1).

[Figure]

Figure 1: The new Fig. 7a *Caption:* **(a)** Fracture angles as function of the plastic potential ellipse ratio $e_G$ for different yield curve ellipse ratios ($e_F = 0.7$, 1.0, 2.0, and 4.0). The markers with ranges are the mean and two standard deviations of the fracture angles. The dashed lines show the prediction from the Roscoe angle (Eq. 28). The arrows indicate the angles predicted by Coulomb theory, which are constant with respect to $e_G$. Colors indicate the value of $e_F$ for lines and markers. The $R^2$ between theory and modeled angles for $e_F = 0.7$, 2.0, and 4.0 are 0.97, 0.95, and 0.97.

- **R1#6,** Finally, I find that a discussion of previous studies that have presented similar interests and analyses is lacking from the discussion. Hibler and Schulson (2000) have indeed implemented a non-normal flow rule in the VP model, using a Mohr-Coulomb yield curve with an elliptical cap (*"modified Coulombic"* curve). They have also found that a non-normal flow rule affects the orientation of deformation features in the VP rheology. This work is cited in the discussion section, but not really discussed in terms of the differences or similarities between both approaches, nor in terms of the advances of the present study compared to this previous one. I suggest clearly stating that is new here and what is the broad relevance of the results. The model of Hibler and Schulson (2000) has also been used by Hutchings et al. (2005) who have looked at intersection angles. They have compared simulated angles between the modified Coulombic and the elliptical yield curve. Mentioning these previous results and comparing them with the current study would be interesting and would strengthen the literature review and Discussion part of the paper.

  - Hibler and Schulson (2000) effectively used a yield curve with a non-normal yield curve. Nevertheless, they link the fracture angles to the slope of the Mohr–Coulomb limbs of the yield curve ($\mu$), and not to the orientation of the flow rule. Also, they did not show an actual fracture creation at high-resolution.

  - Hutchings et al. (2005) investigated the fracture angles with the *modified Coulombic* but did not explain the variations of the fracture angles, and only explained that the difference between theory and experiments comes from numerical convergence.

  - In Ringeisen et al. (2019), we also investigated a modified version of the *modified Coulombic* yield curve.

  - An investigation of Mohr–Coulomb yield curve with non-normal flow rule (Ip et al.,

 in a similar setup is underway, but lies outside of the focus of this work.

To improve our manuscript, we make the following modifications:

- *"Previous studies with a non-normal flow rule (e.g., Hibler and Schulson, 2000; Hutchings et al., 2005) did not explore the effect of a non-normal flow rule on the fracture angles."* on L97 of the original manuscript.

- *"According to Hibler and Schulson (2000), the flow rule may have an effect on the angle of fracture, but the authors limited their case to the framework of flawed ice and did not consider Roscoe's theory of dilatancy. The rheology of Hibler and Schulson (2000) was tested in an idealized experiment more complex than ours (Hutchings et al., 2005), but the effect of using a non-normal flow rule was not explored. The complexity of their setup may explain the observed difference between simulated and predicted angles. Note that the rheology in Hibler and Schulson (2000) was built by changing the shape of the yield curve* a-posteriori, *while the rheology presented here solves the constitutive equations rigorously."* on L361 of the original manuscript.

**I therefore recommend major reviews to clarify the important points above before a resubmission. More specific comments that are often linked to these major comments are listed below.**

**Specific comments:**

**R1#7, Page 1, lines 8-9:** *"A newly adapted theory (...) predicts numerical simulations of the fracture angles (...) with a root-mean-square error below 1.3 degrees."* This formulation is unclear and needs rephrasing: a newly adapted theory is implemented in the VP model and leads to prediction of the prescribed fracture angle with a RMS error below 1.3 degrees"?. Also, see my main comment about the agreement of the theory with your modeled angles.

We rewrite the abstract. Also, see our answer to the the main comment R1#5.

The new abstract reads " *The standard viscous-plastic (VP) sea ice model with an elliptical yield curve and a normal flow rule has at least two issues. First, it does not simulate fracture angles below $30°$ in uni-axial compression leading to a stark contrast with observations of Linear Kinematic Features (LKFs) in the Arctic Ocean. Second, the tight coupling between the fracture angle, post-fracture deformation, and the shape of the yield curve was identified as the reason for this behavior. In this paper, these issues are addressed by removing the normality constraint on the flow rule in the standard VP model in a uni-axial compressive loading setup. To this end, an elliptical plastic potential – that defines the post-fracture deformations, or flow rule – is defined independently of the elliptical yield curve. As a consequence, the post-fracture behavior is decoupled from the mechanical strength properties of the ice. In a newly adapted theory – based on one developed from observations of granular material – the fracture angles depend on both yield curve and plastic potential parameters. This theory predicts fracture angles well below $30°$. Numerical experiments confirm that the flow rule details determine the fracture angle. For instance, a plastic potential with an*

*ellipse aspect ratio smaller than two (i.e., the value of the standard ellipse) gives fracture angles as low as 22°. Implementing an elliptical plastic potential in the standard VP sea ice model requires only small modifications to the code. The model dynamics with the modified rheology, however, are more difficult to solve numerically. An independent plastic potential provides a solution to two issues with the standard VP rheology: it allows for smaller fracture angles that fall within the range of satellite observations, and it decouples the angle of fracture and post-fracture deformation from the shape of the yield curve. The orientation of the post-fracture deformation along the fracture lines (convergence and divergence) is controlled by the shape of the plastic potential. An orientation that is different from the standard VP rheology requires a non-elliptical plastic potential. "*

**R1#8, Page 1, line 11:** I suggest dropping "*In conclusion*" from your abstract.
Corrected as suggested

**R1#9, Page 1, lines 14-15:** "*to make the fracture angle independent of (not on) the confining pressure (as in observations)*". This relates to another of my main comments : what sea ice observations support that fracture angles are independent of the confining pressure? Please give supporting references. Is it even possible to distinguish between fracturing processes occurring in confined and unconfined conditions in the sea ice cover at the geophysical scale?
Please see our answer to the main comment R1#3.
We replace "*independent on*" by "*independent of*"

**R1#10, Page 1, lines 19-20:** "*narrow lines of deformation observed in the Arctic sea ice cover, emerge in high-resolution simulations (Kwok, 2001; Hutchings et al., 2005)*". It would be relevant to cite more up-to-date works on high-resolution simulations here.
The idea is here to cite the seminal studies about LKFs, we are now also citing more recent literature.
We add the following references: (Hutter et al., 2018; Koldunov et al., 2019; Heorton et al., 2018).

**R1#11, Page 2, line 23:** "*The ice strength locally depends on the ice thickness*". This is only partially true: local ice strength does not depend only on local ice thickness. This sentence perhaps needs some rephrasing.
Corrected as suggested
The sentence in the revised manuscript now reads: "*Locally, the ice strength depends on the sea ice state (thickness, concentration, . . . ), which in turn . . .*"

**R1#12, Page 2, lines 25-27:** "*In granular media like sea ice (...) Note, that in this study, we consider sea ice to be granular not only in the marginal ice zone, but also in pack ice, where ice floes are densely packed*". This again one of my major concern: what is the basis for this assumption? How do you reconcile this assumption with the fact that your goal is to reproduce fracture angles in sea ice? Does pack ice, newly-formed ice or any ice that is not yet fractured into floes or constituted of agglomerated, refrozen floes always present the characteristics of a granular media? Please explain and also give some support for this assumption.
We argue that yes, "*pack ice, newly-formed ice or any ice that is not yet fractured into floes or constituted of agglomerated, refrozen floes*" still carry granular characteristics. The anisotropy at subgrid scale is still present in a way that fracture will rarely be created in

straight lines, but will most probably follow the network of weaknesses.

**R1#13, Page 2, line 28:** "*This anisotropy*". This is unclear. Please define this anisotropy and better explain how it emerges.

We modified the text: "*The intersection angles between the LKFs have an influence on the deformation field and, hence, on the local sea ice strength and the emergent sea ice anisotropy (Aksenov and Hibler, 2001). This anisotropy, which emerges as sea ice develops weak and strong areas along LKFs as leads or ridges form locally, then influences ...*"

**R1#14, Page 2, line 37:** The brittle model used in (Rampal et al., 2016) is the EB model of Girard et al. (2011). Please modify the reference.

Corrected as suggested by the reviewer.

**R1#15, Page 2, line 39:** I believe a simpler and scientifically more objective formulation would be "*most widely used*", instead of "*de facto standard*".

"*De facto*" means "*in fact*" or "*in effect*". We are just stating a fact here.

**R1#16, Page 2, lines 48-49:** Yes, granular media indeed present shear bands, which are not the same as fractures. Again, please clarify what you want to represent in your model. What is the link between LKFs in sea ice, shear bands in granular media and fractures in solid materials?

See our answer to the comment R1#2.

**R1#17, Page 2, lines 48-49 vs line 50:** "*Two classical solutions coexist and set two limit angles for the orientation of fractures: the Coulomb angle (...)*". There is something unclear and contradictory between this and the previous sentence. You invoke the Coulomb theory here, in the context of friction or fracturing? I understand it is the later, but please make that clear by answering my previous comment.

We consider the case of fracture, but this applies also a dense pack of ice floes. We do not understand why these two concepts should be separated. The creation of LKFs in sea ice has been referred to as "fracture" in several preceding publications (e.g., Hutchings et al., 2005).

**R1#18, Page 3, line 56:** I think it would be relevant to make some space and re-introduce the definition of the dilatancy angle here: it would make life easier for the reader and avoid the need to dig for it in another article.

Added as suggested

We add the following sentences "*Dilatancy refers to divergence along along shear bands or LKFs that is a function of the distribution of contact normals between individual floes at the sub-grid scales. A positive angle of dilatancy is associated contact normals that (on average) opposes the macroscopic shear motion and divergence along the shear band; while negative dilatancy is associated with a closing of the fracture line and ridging*" on L55 of the original manuscript.

**R1#19, Page 3, line 58:** "*A general theory derived from experiments with sand that takes*

*into account both the angle of friction (...)*". In the case of sand, contact and friction are indeed at play and shear bands are formed. This again adds to the confusion: internal angle of friction or angle of friction? i.e., fracture or friction? Please clarify.

Please see our answer to the major comment R1#2

**R1#20, Page 3, line 60:** based *on* the grain size.
Corrected as suggested

**R1#21, Page 3, lines 67-68:** "*a larger dilatancy angle implies a larger grain size, more contact normals, hence more friction*". Can you please include some references that support this?
We add a citation.
We now refer to Vermeer (1990) at the end of the cited sentence.

**R1#22, Page 3, line 73:** There is a mistake here, as Weiss and Schulson (2009) reported observed fracture angles between 20 and 50 degrees. Or did you derive this directly from their estimated internal friction angle, which is fit to in-situ stress measurements? In the later case, this is then not an observation of fracture angles but a derivation based on some physical assumptions, which are moreover debatable (see Dansereau et al. (2019) and many others), and it should be removed from the list of observations of fracture angles.
Corrected as suggested.

**R1#23, Page 3, lines 74-76:** You state that uni-axial compression experiments showed that (3) the fracture angle is a function of the confining pressure. How did you determine that without performing bi-axial compression experiments? Is there a typo here?
No, this is no typo. Ringeisen et al. (2019) showed that the fracture angles changes with the confining pressure when a elliptical yield curve is used, the forcing was uniaxial but the ice was confined, hence similar to a bi-axial loading.
We modify the text to now read: "*the fracture angle is a function of the confining pressure. The confinement was achieved by adding thinner ice on either side of an ice slab subjected to uni-axial loading.*"

**R1#24, Page 3, line 75:** the "gradient" of shear to compressive strength. Did you mean the ratio?
The fracture angles does not depend on the ratio, but the slope of the tangent to the yield curve (Ringeisen et al., 2019; Pritchard, 1988). This slope determines where the ice will break on the Mohr's circle of stress, i.e., the fracture angle.
We changed the sentence to: "*...the angle of fracture is a function of the gradient of shear strength with respect to compressive strength (i.e. the slope of the yield curve) ...*"

**R1#25, Page 3, line 76-79:** See again my major comment about the apparent confusion between fracturing, friction, granular media, sea ice and a viscous-plastic continuum rheology. I think it is crucial to clarify the links you make between these processes and the motivation of your approach here. This passage in particular leads the reader to believe that your goal is that the VP rheology complies with observations of granular media behavior, because you consider that sea ice at the geophysical scale, in all its different states, is a granular media. If this

assumption is at the very basis of your approach, it should be stated earlier in the introduction, (very importantly) along with supporting arguments. This would make the reading and the assessment of your assumptions and methods by the reader much easier.

See our answer to the comment R1#2.

**R1#26, Page 3, line 82:** "The ratio of shear to divergence along the LKFs allows to infer the dilatancy angle." Again, if one assumes sea ice in any state behaves as a granular material.

We clarify this in the revised manuscript. It is important to note that dilatancy (dilatancy can be positive or negative) in leads is a known fact. If most of the deformation happens in shear, LKFs play a predominant role in thick ice formation (ridging) as well as in thin ice formation

*"The ratio of shear to divergence along the LKFs allows to infer the dilatancy angle when considering sea ice as a granular material."*

**R1#27, Page 3, lines 86-87:** *"Separating the link between the fracture angle and the flow rule from the yield curve is necessary to design rheologies that are consistent with observed sea ice deformations"*. Please note that this would be only true for plastic flow rheologies and not applicable nor necessary for rheologies based on elasticity (EB, MEB, Elastic-Decohesive). To be objective, this statement should therefore be modified as *"necessary to design plastic flow rheologies that are consistent (...)"*.

We correct as suggested.

the sentence in the revised manuscript now reads *"...to design VP rheologies that ..."*

**R1#28, Page 4, line 90:** *"In these different classes of models, various rheologies can be defined"*. This is not true and/or not clear: these are rheological models and therefore they do not include different rheologies. I think that you mean that these different models require the definition of different components: a constitutive relation (all models), a yield/damage curve/criterion (all models including a threshold mechanism, i.e, a change in mechanical behavior) and a flow rule (only plastic flow models). I therefore suggest to rephrase and clarify this passage and the next sentence, that is *"in a VP rheology, a yield curve and plastic potential (flow rule) must be defined"*. In the same line of idea, I do not really see the point of the last sentence of this paragraph. Maybe it can be cut if some rephrasing is made at the beginning of the paragraph?

A VP model with a different yield curve and/or a different flow rule can describe a different physics in the modeled material. A VP rheology with a Mohr-Coulomb yield curve (e.g. Tremblay and Mysak, 1997) will create different results than the one with an elliptical yield curve. The last statement is important for this paper, because it stresses the fact that changing the flow rule changes the system dynamics.

**R1#29, Page 4, lines 96-97:** See my major comment above. Hibler and Schulson (2000) have indeed used a VP model with a non-normal flow rule and a Mohr-Coulomb yield curve with elliptical cap, or *"modified Coulombic"* curve, as cited in your Discussion section. This model has also been used by (Hutchings et al., 2005) (https://doi.org/10.1175/MWR3045.1) who have looked at intersection angles and compared them between the modified Coulombic and the elliptical yield curve. As their approach is therefore close to yours, it would be important and certainly interesting to explain the similarities and difference between your work and theirs in the literature review (introduction) section. Please also note that Hibler and Schulson (2000)

do not seem to share your view that the angles of fracture in sea ice at the geophysical scale are independent of confinement, which would be an important point to discuss further.

See our answer to the major comment R1#6.

**R1#30, Page 4, line 100:** *"viscous-plastic materials"* or *"a viscous-plastic material"*, *"with any flow rules"*.

Corrected as suggested

**R1#31, Page 4, line 100:** "from the yield curve".

Corrected as suggested

**R1#32, Page 4, lines 101-102:** "The new model is tested in simple uni-axial loading experiments". See my major comment above: a quick addition to your work would be to test if your numerical implementation also holds under bi-axial loading conditions, that is, if the angles vary or not with confinement.

See our answer to the general comment R1#3 as well as comment R2#2 and R2#39 from Reviewer #2

**R1#33, Page 4, line 108:** ''We consider sea ice as a 2D viscous-plastic material". See my previous major comment: please explain the physical link between this viscous-plastic assumption and that of a granular material.

See our answer to the general comment R1#2

**R1#34, Page 4, line 113:** In your case, the constitutive equation links the vertically integrated stress tensor to the deformation rate, which you introduced on the previous line.

Yes exactly. For clarity, we prefer repeating *"stress tensor"*. However the term *"rate"* with *"deformation tensor"* was missing.

We modify as suggested: *"The constitutive equations link the vertically integrated stress tensor $\sigma$ to the strain rate tensor $\dot{\epsilon}$"*

**R1#35, Page 4, lines 117-119:** Representing small deformations with a viscous model is rather counter-intuitive, especially for a reader that is familiar with viscous-plastic rheologies (plastic for small, viscous for large deformations). I believe it is important that you explain in more details how a viscous rheology is expected here to represent the small deformations of a solid (time scales, viscosities, etc).

Effectively, this VP models differs from other Viscous-Plastic models, e.g. Bingham plastic, which include a yield condition (rigid solid) and then deforms as a viscous plastic with a linear relationship between viscosity and strain. We add more details to our description of viscous behavior in the last paragraph of Sec. 2.1, on L155

We add the following text on line 157 of the original manuscript *"VP sea-ice models typically cap the viscosity at*

$$\zeta_{\max} = \frac{P}{2\Delta_{\min}} = \left(2.5 \times 10^8 \, \text{s}\right) \cdot P$$

and $\eta_{\max} = \frac{\zeta_{\max}}{e_G^2}$ *to regularize the momentum equations. When this regularization is in effect, $\zeta$ and $\eta$ are independent of the deformation field ($\Delta$) and the stress divergence reduces to harmonic viscosity with constant coefficients. $\Delta_{\min} = 2 \times 10^{-9}\,s^{-1}$ (Hibler, 1979, 1977) translates to a deformation time scale of almost 16 years. Therefore, viscous deformations are slow and negligible with respect to the plastic deformations, and VP rheologies are almost purely plastic. The viscous behavior is a consequence of regularizing the viscosities rather than an implementation of a physical behavior.*"

**R1#36, Page 5, line 130 to page 6, line 149:** These paragraphs could be shortened by removing or presenting in a more concise manner some general pieces of information.

We would like to keep it in the present form because we think it is a useful description of VP rheology.

**R1#37, Page 5, lines 130–131:** As it is not the states of stress that are deforming plastically, but the material, this sentence needs some reformulation.

Corrected as suggested by the reviewer.

"*The yield curve represents the stress states for which sea ice deforms plastically while enclosing the stress states for slow viscous deformation.*"

**R1#38, Page 9, line 204:** "The slope of the yield curve". And many other missing "the" throughout the text.

Corrected as suggested. We thank the reviewer for pointing all these out to us.

**R1#39, Page 10, line 223:** How does the no-slip condition at the bottom boundary affect your results compared to the case in which slip is allowed in the x-direction (i.e., by holding only one of the two bottom corners of the domain fixed in x and y)? Such boundary conditions are maybe less representative of a floe that sticks to a coast but would not lead to as much concentration of stresses on the bottom corners of your ice floe (here your Bcs imply some bi-axial compression at the bottom) and hence would put less constraint on the appearance of conjugate faults and on their orientation. I think this would be an interesting and not time-consuming test.

In Ringeisen et al. (2019), we already investigated the effect of the no- and free-slip condition, and we showed that the configuration used here does not influence the angle of fracture, as indicated on L248 on the original manuscript.

**R1#40, Page 11, line 240:** I suggest "more numerically challenging".

Corrected as suggested.

**R1#41, Page 11, line 256:** ''laboratory experiments". If you compare your results with laboratory experiments, please provide more details on these experiments (e.g., boundary conditions? biaxial or uni-axial compression? on samples with an aspect ratio similar to sea ice, i.e., virtually 2D? on fresh or sea ice?) Were such experiments made by Erlingsson (1988) and Wilchinsky et al. (2010)?

Corrected as suggested by the reviewer.

In the corrected manuscript, this sentence now reads: "*The fractures form*

*a diamond shape, similar to the shapes observed at large scales (Erlingsson, 1988), in laboratory experiments (Schulson, 2001), and modeled with DEM models (Wilchinsky et al., 2010) or other continuous sea ice models (Ringeisen et al., 2019; Heorton et al., 2018)."*

**R1#42, Pages 11-13 and caption of figure 6:** What is the field represented in figure 6? I assume from the color scale that it is a deformation rate?

The field shown here is the shear deformation $\dot{\epsilon}_{\mathrm{II}}$.

We clarify this in the caption: **"Diamond-shaped fracture pattern in the shear deformation field $\dot{\epsilon}_{\mathrm{II}}$ for $e_F = 2.0$ and three different values of $e_G$ after five seconds of simulation."**

**R1#43, Section 4 and figures 6 and 7:** How are the angles of the features observed on fields such as shown on figure 6 measured, i.e., estimated? It would be important to mention what method is used.

This is described in Section 3 *Experimental setup and numerical scheme*, Line 245 to Line 250.

**R1#44, Result section, figure 7 and page 15, lines 292 and 306-308:** "the theory predicts the fracture angles accurately" and "The results illustrate clearly how the yield curve defines the stress for which the ice will deform, that is, the transition between viscous and plastic deformation, and how the relative shape of the plastic potential with respect to the yield curve defines both the type of deformation (convergence or shear) along the fracture line and the fracture angle. The resulting fracture angles are in excellent agreement with the Roscoe angle predictions (Roscoe, 1970)." There is my major comment about the results. In section 2.3, you describe how the yield curve, flow rule and angles are related in your model. By prescribing the yield curve and plastic potential ellipse ratios, you prescribe locally the angle (Roscoe) of "fractures". Figure 7 shows that at the macro-scale, i.e., the scale of the ice floe you indeed retrieve that angle. What is prescribed at the local scale is what you get at the macro-scale in your model, as expected in a model of plastic flow. Therefore my understanding is that these tests serve to verify that your numerical scheme is OK. Is that the case? To better illustrate that point, it would be relevant to show the (deformation?) fields at different stages of the compression experiment, to illustrate how the features arise in your model.

We show the fracture after 5 seconds of simulation, in order to get the initial fracture and avoid more complex interactions that might create more fractures (see Fig. 6 in (Ringeisen et al., 2019)). Please see our answer to the general comment R1#4

**R1#45, Page 15, line 300:** ''the shape of the plastic potential".
Corrected as suggested.

**Page 15, line 305:** "this allows decoupling the mechanical strength properties of the material (ice) from its post-fracture behavior". Again the contradiction with the assumption of a granular material, i.e., an already fractured/fragmented material. How do you reconcile these ideas?

See our answer to general comment R1#2

**R1#46, Page 15, lines 306-308:** ''The results illustrate clearly how the yield curve defines the stress for which the ice will deform, that is, the transition between viscous and plastic deformation, and how the relative shape of the plastic potential with respect to the yield curve defines both the type of deformation (convergence or shear) along the fracture line and the fracture angle. The resulting fracture angles are in excellent agreement with the Roscoe angle predictions (Roscoe, 1970)." But you prescribe the yield and plastic potential in your model: why would you not expect what you get to indeed be what you prescribe? In other words, you do not make any distinction between what you prescribe at the micro-scale (scale of your discretization) in your model and your macroscale results and you do not discuss why you expect these behavior to be identical or not : that is missing from your work and interpretation of your continuum model.

See our answer to general comment R1#5

**R1#47, Page 15, point 2:** About confinement, shear bands and fractures, see my major comment above.

As for the other comments raised about relationship between fracture angles and confinement (R1#3, R2#2), this behavior is linked to the elliptical nature of the yield curve.

We add a reference to our study showing how the confinement changes the fracture angles with an elliptical yield curve: "*This behavior cannot be eliminated with an elliptical plastic potential, as the normal stress along the LKFs increases with confining pressure and the flow rule changes from divergence to convergence (Ringeisen et al., 2019).*"

**R1#48, Page 17, line 382:** "sea ice mechanical strength properties (yield curve) and deformation (flow rule)". Again, you write this with the perspective of a VP model, but mechanical strength properties and deformation are not only determined by the yield criterion and flow rules in other rheological models for sea ice. Please be specific and make this distinction clear. Also, I do not understand why Dansereau et al. (2016) is cited in this context.

We refer to Dansereau et al. (2016) in this context because the way the damage parameters act as the history of the model deformation is very interesting, and could be a representation of the state of the local ice (broken/unbroken), i.e. "*sea ice mechanical strength properties (yield curve)*" cited before.

We reformulate the sentence on L382 of the manuscript "*...; the sea ice mechanical strength properties (i.e., yield curve) and deformation (i.e., flow rule for VP rheologies) should vary in time and space depending on, for example, the time-varying distribution of the contact normals, floe size distributions, or the damage parameter, as per observations and laboratory or numerical experiments (Overland et al., 1998; Hutter et al., 2019; Horvat and Tziperman, 2017; Roach et al., 2018; Balendran and Nemat-Nasser, 1993; Dansereau et al., 2016; Plante et al., 2020).*"

**R1#49, Page 17, lines 387-388:** "So is the combined knowledge of the failure stresses and their associated deformation of sea ice as a 2D granular material". This is confusing: why then do you base your approach on the assumption of a granular material? This goes along my main comment and really needs to be clarified.

If deformation data are available from satellite observations, we still have little knowledge about the stress associated to these observations. This is especially true when these deformation lead to ridging and creation of open-water. Also, most of the laboratory data investigate 3D continuous ice, we are not quite sure if these results can be extrapolated to sea ice, i.e. we are

missing knowledge about 2D fractured materials behavior. See also our answer to comment R1#2.

We reformulate"*. . . higher temporal resolution of sea ice deformation and flow size distributions is still unavailable. There is also a knowledge gap in the interplay between yield stresses and the post-fracture deformation in a 2D granular material such as sea ice. This interplay is likely different than for the well studied case of a solid homogeneous 3D block of ice (e.g. Schulson, 2002).*" on L387 of the original manuscript.

**Answer to tc-2020-153-RC2 – Harry Heorton**

This paper describes the implementation of a non-normal flow rule in the VP sea ice rheology. The equational form of the new rheology is well described and several very useful diagrams are included. The numerical implementation is linked to a theory that links the flow rule and the intersection of failure lines within the medium described. A series of idealized numerical experiments are performed which show that the numerical rheology successfully recreates the fracture intersection angles predicted by the presented theory. The authors follow the experiments with a discussion on the implications of using a non-normal flow rule when designing future sea ice rheologies. They describe the various challenges when using non-normal flow rules. I find that this paper is well written and a valuable contribution to the modeling of sea ice deformation. It is a very useful introduction to use of non-normal flow rules for sea ice modeling for future work in this area. I recommend this paper for publication after a few questions I have.

We would like to thank the reviewer for the review of our manuscript. The many suggestions and comments will, without doubt, increase the quality of this manuscript.

**R2#1,** First of all can you explain why figure 7a contains both theoretical links between the plastic potential and intersection angle and many numerical experiments that back up the theory but 7b contains relatively few numerical results? I can see several cases where additional results from 7a can be copied to 7b and back up your results. Is it true that the full range of values for 7b are not obtainable due difficulties that the authors discuss in getting the model to converge to a solution for highly non-normal flow? If this is case then please tell us.

Figure 7b was intended to illustrate how the fracture angle changes when the plastic potential stays the same, but the yield curve changes. This was not the intended goal of this paper, as we wanted to focus on the effect of a varying flow rule at constant yield curve. We added the few point to show that the fit is still very good, only these few points could be reported. We would need to do many more simulations to populate this figure. To avoid confusion, we decided to remove the few points on this figure and emphasize the fact that it is shown for illustration. Please see our answer to comments R2#34 and R2#36.

**R2#2,** Several times in the discussion and results the authors say that the intersection angle depends on the confining pressure despite the varying non-normal flow rule. I can see no evidence of this in their results. The presented experiments show changing intersection angle with changing flow rule (varying plastic potential and yield curve eccentricity), but I see no results where they change the confining pressure. Is this from previous work? Or an interpretation of the results that they do present?

The fact that the angles depends on the confining pressure with a elliptical yield curve was explained in Ringeisen et al. (2019). Because the yield curve is still an ellipse here, there is no reason that this would change. We added a sentence to clarify this point. Please see our answer to comments R2#39 and R1#3.

**General editing points:**

**R2#3,** Can you please start the paper with a description of what a flow rule is. Then what a normal flow rule is, and the crucially what the main difference physically and theoretically is between a normal and non-normal flow rule. I see that a definition is on line 90, and then

further physical descriptions of the flow rule are in the results. The introduction make much more sense if these can come first.

We followed the reviewer's suggestion and reorganize the introduction.

We reorder the introduction by moving the paragraph starting by "*This paper focuses on VP rheologies. Different...*" before the one starting by "*LKFs have been studied for...*"

**R2#4,** Can you describe what is documented in this study that is novel and new?

Corrected as suggested

We make the following modifications

- We modify the abstract, see our modifications following comment R1#7.

- We modify the penultimate paragraph of the introduction (see also the answer to comment R1#2). It now reads "*In this paper, we investigate the effects of a non-normal flow rule on fracture angles and its use as a means of separating the state of stress (at failure) and the post-fracture deformation. The novelty of this paper is that we study the non-normal flow rule in the context of the standard VP rheological model using a similar shape for the plastic potential (i.e., an ellipse) because (1) it is widely used in the community, and (2) its behavior is well documented (compared to other models), providing a solid basis for comparison. This paper provides a new generalized theoretical framework for developing any viscous-plastic material with normal or non-normal flow rules. To this end, we test the new model in simple uni-axial loading experiments where the relationship between fracture angle and flow-rule can be easily identified.*".

**R2#5, L20** they are also, more importantly, observed

We are not sure what is meant here. We already state in the same sentence that LKFs are observed, and they emerge in high-resolution simulations.

We try to clarify: "*Linear Kinematic Features (LKFs), narrow lines of deformation, are observed in the Arctic sea ice cover, and also emerge in high-resolution simulations (Kwok, 2001; Hutchings et al., 2005).*"

**R2#6, L21** Here your LKF's influence in many ways but what follows is not a list. Consider re-writing

Corrected as suggested.

We rewrite as a list "*...: heat and matter exchange take place primarily over open water (Badgley, 1965), salt rejection during ice formation in leads creates dense water and influences the thermohaline circulation (Nguyen et al., 2011, 2012; Itkin et al., 2015), and the ice strength locally depends on the ice thickness, which in turn is affected by sea ice fracture with thermodynamical growth in opening leads and with local dynamical growth during ridge formation.*"

**R2#7, L22** Please define what a lead is. Consider starting with a definition of LKF's that are typically leads or ridges

Corrected as suggested.

"*LKFs can form in divergence, creating stretches of open water or leads, or in convergence, creating piles of ice or ridges. (Stern et al., 1995)*"

**R2#8, L70** Which is the 'standard rheology'? do you mean the VP rheology. Also can you further describe this result. How did Ringeisen find that the angle can't be lower than 30 degrees?

We meant the standard VP rheology, i.e. the VP rheology with elliptical yield curve and normal flow rule.

We clarify by adding "*Standard VP rheology*" on L70 of the original manuscript. We also add "*, as shown by idealized experiments and theory (Ringeisen et al., 2019).*" on L71 of the original manuscript.

**R2#9, L71** the following list is hard to read. Consider reformatting. Also what does the $\mu = 0.9$, relate to with the Weiss and Schulson reference.

We decided to remove this citation from the list following the comment R1#22 of reviewer #1.

**R2#10, L71** can you confirm that these angles are all comparable? I have found that studies document both the intersection and also the half angle, being the intersection between the fracture and the principal axis of stress.

We can confirm that these angles are measured the same way, i.e. they are the half angles, as for our study. (Hutter and Losch, 2020) used intersection angles, we divided the angles they reported by two in this list.

**R2#11, L80** this paper requires a definition for a normal flow rule. This sentence and the following paragraph make little sense without it.

We add a definition of the normal and normal flow rule in the introduction, when the VP model is introduced.

"*A VP rheology is composed of (1) a yield curve that defines the stress at which deformation changes from viscous to plastic, and (2) the flow rule that defines the nature of the deformation (i.e. convergence, divergence, and shear). The flow rule can be normal to the yield curve (a normal flow rule), or in any other direction (a non-normal flow rule).*"

**R2#12, L82** do you mean that the flow rule can be observed by measuring the ratio of shear a divergence along LKF.

Yes, the ratio of shear and divergence can be measure along the LKFs and give indications on the flow rule.

We modify the sentence as "*The ratio of shear and divergence along the LKFs is a measure for the orientation of the flow rule, hence the dilatancy angle.*"

**R2#13, L85** were these laboratory observations performed the same way as those of Stern mentioned above?

Observations in Stern et al. (1995) are in Arctic sea ice, not from experiments, we add a sentence to clarify this. According to reviewer 1 (R1#13) the retrieval of the flow rule in Weiss et al. (2007) might be questionable and we decided to remove the sentence.

"*Observations of sea ice drift in the Arctic show that most of the deformation takes place in shear, that is, 98% of deformation show more shear than divergence or convergence (Stern et al., 1995).*"

**R2#14, L89** it will be nice to have the Anisotropic Plastic (Tsamados et al., 2013) rheology listed here too

Added

We added the following entry to the list of rheological frameworks: *"..., Elastic-Anisotropic-Plastic (EAP) (Tsamados et al., 2013), or Maxwell-..."*.

**R2#15, L92.** Good to see a flow-rule definition here. How does the plastic potential determine the postfracture deformation? is this through the direction of the principal stress when the yield criterion is reached?

The flow rule is perpendicular to the plastic potential, as stated on page 3, L93. We will reorder the introduction and this will appear sooner in the introduction.

**R2#16, L115** is f here the Coriolis acceleration as above? Actually can you tell what value was used for the Coriolis acceleration? If it is non-zero (valid to use zero and non-zero for these experiments) then asymmetry will be expected (see comments later)

Yes $f$ is here the coriolis parameter. However we use $f = 0$ in this study.

In the Sec. 3, L231 of the original manuscript, we add the sentence. *"For simplicity, $f = 0$."*

**R2#17, L120** It is great to read this description of the VP rheology. A really helpful addition.

Thanks!

**R2#18, L138** is it possible to have a physical description of the plastic potential here? The physical description of what the yield curve represents is very helpful. A similar description of the plastic potential here will be similarly useful. The flow-rule is a difficult concept that is explained well here. An additional physical description will make it even better.

In the new version of the manuscript, we add a few sentences describing the flow rule in physical terms.

*"The flow rule represent the direction of deformation in the grid cell. The orientation of the flow rule in the reference ($\dot{\epsilon}_I$,$\dot{\epsilon}_{II}$), as shown in orange in Fig. 1, indicates if the grid cell deforms convergence ($\dot{\epsilon}_I < 0$) or divergence ($\dot{\epsilon}_I > 0$) and shear ($\dot{\epsilon}_{II}$)."*

**R2#19, L180** I see that the dilatancy angle was introduced earlier. However it would benefit the paper to include a physical description of 'dilatancy of a granular material' either before or here when it is implemented in the model equations.

We add a sentence describing the physical process of dilatancy.

*"Dilatancy is the motion normal to a shear band as a result of grain-to-grain contacts opposing the motion in shear. In other words, the interaction between grains on both sides of the shear band opposes the shear motion and creates divergent motion."*

**R2#20, L180 and onwards.** This section will benefit from an expanded introduction to the theoretical steps performed. From what I can tell, you use the theory that links dilatancy angle to fracture angle as discussed in the introduction. You have quantified the dilatancy angle using geometrical description of an arbitrary yield curve and plastic potential. This is expanded through the notation to express the fracture angle as a function of yield curve and

plastic potential eccentricity. Is this correct? If so is the motivation behind the description that it is possible to show how the expected fracture angle is expected to change with changing plastic potential?

This is absolutely correct. The construction for the normal flow rule in blue on Fig. 3b shows what happens if the plastic potential is different, the fracture angle $\theta$ is different.

We add the following text on L180 of the original manuscript: "*To adapt the Roscoe angles to sea ice modeling, we proceed as follows: (1) the stress state on the yield curve (point p on Fig. 3a) defines the position and size the Mohr's circle at fracture (blue circle on Fig. 3b), (2) the slope of the plastic potential determines the point on the Mohr's circle where deformation takes place, that is, the slope directly predicts the fracture angle $\theta$ as function of the dilatancy angle $\delta$ (per Roscoe theory, Fig. 3b). For the special case of uni-axial compression, we (A) determine the stress state on the yield curve for uni-axial compression as function of the yield curve ellipse ratio $e_F$, and (B) compute the slope of the plastic potential at that stress state as function of the plastic potential ellipse ratio $e_G$. Finally, we combine (2) and (B) to compute the theoretical prediction for the fracture angle as function of ellipse ratios $e_G$ and $e_F$.*".

**R2#21,** Can you be clear what the theory of Roscoe is describing. Is the angle you are obtaining the expected angle of fracture due to minimizing some sort of energy potential? Or does it relate to an analytical solution of fracture? The mathematical expansion here is clear to follow, but the reasoning behind why you have shown it is less so.

The Roscoe angle is base on observations of deformation of granular materials in the lab, and is explained as the direction of "*zero-extension lines*"

We add the following sentence "*Based on laboratory experiments, the zero-extension lines, also called velocity characteristics, seem to be a better predictor for the orientation of shear bands in granular materials than the stress characteristics (Roscoe, 1970).*".

**R2#22,** In figure 4 you describe how the ratio of divergence to shear changes with changing plastic potential. Is this the key effect of the non-normal flow rule? In that by separating the yield curve and plastic potential it is possible to change the ratio of divergent to shear stresses whilst under deformation? But without also changing the point of deformation (as in the yield curve) If so please emphasize this point throughout the paper! It makes the non-normal flow rule much clearer for me!

Yes it is exactly the point.

We make two modifications to add this explanation of the non-normal flow rule:

- "*By doing this, we will change the orientation of the flow rule, without changing the deformation stress state (see Fig. 2 and Fig. 4 for some examples).*" on L160.

- "*. . . with an elliptical yield curve, modifying the flow rule without changing the deformation stress state.*" on L401.

**R2#23, Figure 3 caption** - the arrows are described as orange, but appear red to me.
The mistake was in the caption, the arrow is red. The caption is modified accordingly.
The caption now reads: "*The red arrow is. . .*"

**R2#24, Figure 4** I see red and orange arrows here, and they are correctly described. Can you check

We checked – the colors are correct here.

**R2#25, figure 3** Do the colours relate between the two figures?

Yes, red represent the case with non-normal flow rule, the blue one represent the case with a normal flow rule. We add a precision in the caption of figure 3.

The caption was modified to read " *b) Mohr's circle for the fracture state p in a) (for normal in blue and for non-normal flow rule in red) in the fracture plane of reference ($\sigma$, $\tau$)*"

**R2#26, L222** is the initial ice state entirely uniform? Or did you seed some noise into the initial state?

Yes the initial state is with uniform ice (thickness and concentration).

The sentence is now "*...Following Ringeisen et al. (2019), we load a rectangular ice floe of 8 km by 25 km with a uniform thickness of $h = 1\,\mathrm{m}$ and a uniform sea ice concentration of $A = 100\%$ (see Fig. 5). ...*"

**R2#27, L231** did you test at other time and spatial resolutions? Later you comment that fracture angles were shown in a previous study to be independent of model resolution (we found this too). Did you test this for this study too?

No, we did not test the dependence of scale and resolution here, only in Ringeisen et al. (2019). There is no reason to think that this would change, no scale nor resolution are included in the formulation of the rheology.

**R2#28, L232** is this equation 4 that is solved for?

Yes, we add a reference

We add "*...Eq. (4) ...*".

**R2#29, L233** What are the non-linear and linear problems ? Can you relate these back to the model equations?

The non-linear problem is linearized and solved iteratively. The non-linear problem is then updated with the intermediate solution from the linear iterations, and then the cycle continues. We do not describe here further the formulation of the solver, it would be outside of the scope of this work, and relatively long. However, it is described in several studies (Zhang and Hibler, 1997).

We add *"For the linearized problem within each "non-linear" iteration ...".*

**R2#30, L246** So are the simulations only run for 5 seconds of model time? Have you tested how long the model can run for and its overall stability? I read above that you have used excessive computation to ensure the extra complexity of the non-normal flow rule is accounted for. How successful is this approach? Did you find that certain computational setups did not perform well when attempting to solve the equations? Any insight you can share into how to solve these equations will greatly help the sea ice modeling community

We tried this rheology in a pan-arctic setup ($2\,\mathrm{km}$) with an integration time of the order of one year and did not experience any problem (not shown). As we show, a lot of iterations are needed for this experiment, but much less are needed in realistic setups. The actual computational details are not relevant to this paper, but will be discussed in a different manuscript

describing realistic applications, where they are relevant.

**R2#31, L263** what is average residual norm R? is this a measure of the solution accuracy?
We mean the $L_2$-norm of the residual of the non-linear equations.

We modify the sentence on **L263** of the original manuscript to start by "*For instance, the $L_2$ norm of the residual of the non-linear equations*" on **L263** of the original manuscript.

**R2#32, L282** is the shear strain rate shown anywhere? Are you relating back to figure 6? If so can you say so? Are you saying the relationship in figure 6 for $e_F$ and shear strain rate is also true for the various values of eF in Figure 7? Or is this a theoretical postulation?
We were referring to Fig. 6. This is something we observe in the simulations, Figure 6 gives an example.

The sentence starts now with "*For $e_G > e_F$, the shear strain rate increases along the LKFs (see Fig. 6c) and...*". The same way we change the sentence on **L283-284** as "*...less shear along the LKFs (see Fig. 6c), and the fracture ...*"

**R2#33, L282** fracture angle or angles plural? Do you you take multiple angles or just one per simulation?
We measure multiple angles, then compute the average and standard-deviation. In this case, "*angles*" would be the correct conjugation
We correct "*...and the fracture angles tend toward ...*"

**R2#34, Figure 7** Is it possible to add the red orange and teal numerical simulations to figure 7b? If you have added the blue dots then the omission of the others makes me wonder how they will fit? I see that you only have multiple values for $e_G = 4.0$. Though there are 2 points for 0.7 and a single point for 2.0 and 1.0. I also see that the full range of $e_F$ was not investigated for each $e_G$. What is the reason for this? Is it the limitations of the model? Or did you choose not to in order to keep the simulations physically relevant?
This would need new simulations, data points on figure 7a cannot be reported on figure 7b (excepted the three points for $e_G = 4.0$. The figure 7b was intended for illustration only, the blue points for $e_G = 4.0$ were added to show that the theoretical prediction still fit. Because it shows the same formula, the predictions should be as accurate in 7a than in 7b. Changing $e_F$ at constant $e_G$ is not as interesting as the inverse, because both the flow rule and the fracture stress state change (see Fig. 4). We add an sentence in the figure caption to clarify this.

In the caption, we add: "*Theoretical predictions of the fracture angle as function of $e_F$ with a constant $e_G$, for indication. As shown on Fig. 4, if $e_F$ is modified, both the stress state and the flow rule change, resulting in a more complex behavior.*"

**R2#35, L305** this line is very informative to what the non-normal flow rule can achieve. Can you put this information into the introduction and abstract please?
We have added a similar sentence to the abstract, please see R1#7.

**R2#36, L309** while you have displayed the agreement to Roscoe for the cases of constant $e_F$ the case of constant $e_G$ (fig 7b) is inconclusive to the reader due to the lack of numerical simulation data points. Is it possible to fill out figure 7b and thus strengthen this statement?
Filling it is totally possible, but would mean a non-negligible amount of new simulations.

We think this is unnecessary, it would just mean reversing the Eq. (27), we showed the case of $e_F = 4$ to show that it does correspond to the Roscoe Angle as well. Please see also our answer to comment R2#1.

**R2#37, L313** Can you sort out the parenthesis on the Ringeisen 2019 citation. It currently doesn't read very well.

Corrected as suggested.

**R2#38, L317** is this lack of convergence the reason for the lack of results on figure 7b?

No, figure 7b is shown to illustrate of the evolution of the fracture angles when $e_G$ is kept constant and $e_F$ changes. See our answer to the question R2#35

**R2#39, L319** Can you give a citation for a description of how this result with the changing fracture angle with changing stress confinement was obtained? I assume it is not from this study as you have not altered the confinement ratio for any of your simulations. Or are you referring to that the fracture angles change as the loading increases with time?

We are referring to the experiments in Ringeisen et al. (2019, Sec. 3.2.2, Fig. 8), where the elliptical yield curve with normal flow rule is used. We add this citation.

We add the citation as such: "*Because of the elliptical shape of the yield curve, the angle of fracture in the standard VP model changes with confining pressure (Ringeisen et al., 2019, Sec. 3.2.2, Fig. 8) unlike laboratory experiments with granular materials (e.g. sand) where the fracture angle is relatively insensitive to the confining pressure (Alshibli and Sture, 2000).*" Note also our answer to Comment R1#3 of Reviewer #1.

**R2#40, L321** How do think this result relates to laboratory experiments on sea ice where two clear fracture angles were found above a critical confinement ratio? (Golding et al., 2010; Schulson, 2002)

Thanks for the reference, we now add a short discussion in the text.

We add the sentence "*Laboratory experiments show that behavior and yield stresses in sea ice change above a critical confinement ratio (Golding et al., 2010; Schulson, 2002). It is still not clear whether these results can be extrapolated to the modeling sea ice as a 2D medium at the geophysical scale, although several common features can be found (Schulson, 2002).*" on L325 of the original manuscript

**R2#41, L341** Is this result about pure shear and angle of 45°. From the Ip et al. (1991) citation? How was it obtained?

This is not a result, but the prediction from the Roscoe theory when a pure shear flow rule is used. When the flow rule tends to be only oriented in shear, the fracture tends to 45°. We add a few words to reflect this.

The sentence now reads: "*With the Mohr–Coulomb yield curve with a pure shear flow rule (Ip et al., 1991), the Roscoe angles predicts a fracture angle approximately equal to 45°, independently of the slope of the yield curve.*"

**R2#42, L345** angle - angles

Corrected as suggested

*"...would allow for a different angle of fracture with shear ..."*

**R2#43, L363** Is it possible to include a diagram of the various yield curves discussed in this section? This would greatly ease the understanding of your arguments. I'm sure others have included such a diagram in previous work so you may be able to cite such a diagram.

We add a figure to show the alternative yield curve and their flow rule.

A new figure Fig. 8, showing the different yield curves and their flow rule, is added to the new version of the manuscript. It is shown below as Figure 2.

[Figure]

Figure 2: New Fig. 8. *Caption:* Alternative yield curves and flow rules: The Mohr-Coulomb yield curve with shear, non-normal, flow rule (blue Ip et al., 1991), the modified coulombic yield curve with normal (elliptic part) and non-normal (linear part) flow rule (orange, Hibler and Schulson, 2000), and the teardrop yield curve with a normal flow rule (red, Zhang and Rothrock, 2005). The elliptical yield curve with $e_F = 2.0$ is shown for reference (black thin line). $P$ is the compressive ice strength, and $T$ the tensile ice strength.

**R2#44, L369** Can you explain why non symmetrical deformation features are unrealistic or present an incorrect solution? Do they also correspond to poor numerical solutions? With a non-linear system of equations such as in all sea ice rheologies, asymmetry is often expected. This relates back to most laboratory experiments on ice deformation and even the ill-posedness of divergent weakening (Gray, 1999). Also if you use a non-zero Coriolis acceleration then asymmetry will be expected as the run progresses. What value did you use?

- Because our experiment is fully symmetrical (forcing, ice initial state, ...), the fracture pattern should not show asymmetries. This expectation is met by most of the simulations with the elliptical yield curve with normal flow rule (See Figure 6b or in Ringeisen et al. (2019))

- There is no Coriolis force in these experiments that would break the symmetry. We add this detail to the experiment description, see our answer to comment R2#16.

We add the following sentence: *"This asymmetry is not expected, and is not found with normal flow rules, therefore we assume is stems from the non-normal flow."*

**R2#45, L371** I'm not sure I understand your argument here. Are you saying; poor non-normal flow model convergence won't be an issue in realistic simulations as the numerical solver can't solve the VP rheology anyway? Surely this argument says that there isn't a hope of using non-normal flow VP rheology in realistic simulations?

Our argument is:

- The forcing in high-resolution simulations of sea ice changes on long timescales compared to the timestep of the ice dynamics

- Therefore the solver starts from an already good solution from the previous timestep. This way, the solution of the momentum equation is accurate enough even at high-resolution.

- In the experiment here, the forcing increases fast, at every timestep, so the solver does not benefit from the previous timesteps.

The sentence now reads "*The poorer numerical convergence in practice will go unnoticed in high-resolution simulations using realistic geometries: While the number of iterations typically used (O(10)) is much smaller than that required for full convergence, at each time-step, a new iteration typically use the solution from the previous timestep as initial conditions.*"

**R2#46, L396** These issues are not exclusive to high resolution climate modeling. It can be argued they are even more important for current coarse resolution models which are currently used for long climate simulations and typically perform poorly for reproducing ice drift patterns. LKF intersection angles are also observed over basin length scales (Weiss and Schulson, 2009) and your discussion in this paper is relevant for modeling sea ice deformation at these length scales.

We agree with your analysis. We modify our statement as follows.

"... to design new rheologies for more accurate climate models as well as more precises sea ice predictions."

**R2#47, L406** I am confused by your conclusion here. Where have you shown that the fracture angles depend on the confinement pressure? Where did you change the confinement pressure? Do not Figure 6 and 7 show clear changes in intersection angle with changing plastic potential in accordance with predictions from the theory of Roscoe?

We are referring to the results presented in Ringeisen et al. (2019), where we showed that the fracture angles depend on confinement because of the elliptical shape of the yield curve. We state this more precisely in the new version of the manuscript.

"Because of the elliptical yield curve, the fracture angles still depend on the confinement pressure (Ringeisen et al., 2019), and the elliptical plastic potential does not modify the direction of deformation at the fracture lines (convergence or divergence), only the ratio of divergence relative to shear."

**R2#48, L409** again I'm not convinced that symmetric solutions are mandatory for a symmetric experiment? Again can you say whether you used a zero or non-zero value for the Coriolis acceleration? If it is non-zero then asymmetry will be expected.

The Coriolis force is zero in our experimental setup (see R2#16)

---

## Referee Report (RR1)

Note bis:
• The referees comments are shown in black.
• *The authors answers are shown in italic.*
• *The proposed modifications for the manuscript are shown in italic as well.*
• The referee's new comments are shown in red.

Thank you for your response to my comments on your paper. The modifications you have made improve readability. In particular, the abstract is much clearer and the addition of the definition of the different angles makes the reading easier.

You have provided some elements of answer to my main comments in your responses, in particular, to the first comment (R1-2). However, I find that the changes you have made accordingly (in the introduction in particular) are not sufficient to address the point I wanted to make with this comment, which is: you need to state and explain clearly the assumptions behind your work.

In other words, the goal is not to present these assumptions to me, as a reviewer, but to your readers. Therefore, I would strongly suggest that you put the list of arguments you have presented to me in R1-2 in the text (eg., around page 2, lines 32 to 39), to explain that sea ice present both brittle and granular behaviors, but that here you consider it to be a granular (already fractured) media in the context of shearing band angles at the regional to global scale (i.e., the scale of sea ice models). In think this would really help following your line of thoughts, understand your approach and strenghten the manuscript.

More generally, my point is: your paper adresses an issue that will be of interest for a very specific group of sea ice modellers concerned with the details of its mechanical behavior and numerical representation. These people know what sea ice is, know about VP and will most probably be knowledgeable in mechanics (i.e., on granular vs plastic vs brittle behavior and models). I believe that what they need is to be guided through the physical assumptions that you make to be convinced that your approach is physical and relevant to their modelling.

I therefore suggest a ''major'' revision in the sense that I think some, perhaps locally substantial, changes need to be made to the introduction in particular, but you already have brought up some references and a bullet point list of your arguments to me in your review, so introducing them in the text to support your approach should not be too time-consuming.

This will likely lenghten the text. Consequently, and in the line of idea of my previous comment (that people who will read your paper to improve their sea ice simulations will probably know VP), I suggest below some cuts to generic elements in section 2.1 that would make it shorter, while still keeping in mind that you wanted to keep a full description of VP.

There are three more precise points on which I would like to have your comments or answer:

1) one unanswered question: what is the method to evaluate the angle from your simulated fields (i.e., what does the Measure Tool from GIMP, what is the method and the related errors)? Please briefly summarize it in the text.

2) in my point of view, the introduction (around page 2, lines 40-52) still lacks an explanation on why you think VP is an appropriate rheology for a granular media (could be short).

3) my question in R1-5 remains unanswered and I have rephrased it below to make it clearer.

I have also added some questions and comments about your responses and put some minor comments at the end of this review.

R1#1, This paper presents an implementation of a non-normal plastic flow rule in a Viscous-Plastic model with the goal of better representing the observed angles between Linear Kinematic Features in sea ice at the geophysical scale. The paper is overall well written, in a pedagogical way for the theory (section 2) section, which could however be a little more concise in some places. The figures are, for the most, clear.

*We thanks the reviewer for her thorough review of our manuscript. Her comments will improve the clarity and quality of our manuscript. We hope that we address all comments in a satisfactory fashion.*

Here are my major comments/concerns :
• R1#2, It does not appear clear in the paper what physical process(es) the authors really want to model. In the introduction, it is mentioned that sea ice, both in the pack and the marginal ice zone, is considered as a granular material. No physical justification is offered for this assumption. The rheology used to model this granular material is one of plastic flow, but the authors do not explain how they reconcile their continuum viscous-plastic model with a granular behavior. The aim is apparently to reproduce fracture angles (repeated terminology for the features simulated by their model), but the authors do not explain the link between plastic flow, fracturation and the mechanical behavior of a granular material, which is an already fractured/fragmented material in which contacts
and friction dominate. Later, it seems that the authors refer to shear bands in granular materials as if they were associated with the same processes as a fracturing solid. The Coulomb theory is invoked but it is not clear if it is in the context of friction or fracture. There is therefore much confusion throughout the paper as to what the authors consider is the mechanical behavior of sea ice : is it characterized by fracturation? By friction and contacts between already broken up floes? Granular materials like sand are invoked, but is sea ice really assimilated to a sand-like material here? Whatever is assumed, it crucially needs to be clarified and all physical concepts untangled throughout the text in a way that makes physical sense.

*• Sea ice is composed to individual floes that vary in size and thickness along seasons and conditions. Sea ice has often been described as a granular material (Overland et al., 1998; Mcnutt and Overland, 2003; Tremblay and Mysak, 1997). In other fields, granular material has been modeled with continuum plastic flow models, considering both the Coulomb theory or the Roscoe theory (Vermeer and De Borst, 1984; Vermeer, 1990; Balendran and Nemat-Nasser, 1993; Mánica et al., 2018).*
Yes indeed.

*• We think that we need to consider the ice as a granular material if we want to explain divergence along fracture lines (Stern et al., 1995; Bouchat and Tremblay, 2017).*
Why? You need to extend on this.

*The fact that the elliptical yield curve with normal flow rule (Hibler, 1979) feature compressive states with divergent opening (also when low confinement is applied) (Ringeisen et al., 2019) shows that we can consider granular dynamics to already be present in current VP models. In this manuscript, we investigate a modification of the VP model with elliptical yield curve.*

*• We do not consider sea ice to behave like sand, but still as a granular material: a 2D granular material. Sea ice is peculiar in the world of physics, because (1) it is bound to the 2D ocean-atmosphere interface by gravity, but can "escape in the vertical dimension" (page 17, line 389) and ridge when bi-axial compression exceeds a critical threshold. Also ice floes, the "grains" of sea ice, can brake or refreeze. Therefore, sea ice dynamics exhibits a large spectrum behaviors, including characteristic granular dynamics, for example dilatancy, as well as brittle behavior.*
*• The terms referring to brittle behavior, such as fracture angle or fracture lines, might be slightly confusing with the idea of sea ice as a granular material, but we would like to keep them as it is. Here is our reflection:*
*✶ If we agree on the fact that sea ice is already a fractured medium, we study the large scale deformation of a compact ice field, process similar to the creation of fracture in continuous solid.*
*✶ In that case, it makes little sense to us to make a distinction between fracture and friction. This is well described in the abstract of (Wilchinsky and Feltham, 2011): "Sea ice failure under low-confinement compression is modeled with a linear Coulombic criterion that can describe either fractural fahilure or frictional granular yield along slip lines." The assemblage breaks and floes interact with*

*one another, which can be seen as the microscopic behavior of friction.*

Of course both fracture and friction are present within sea ice. But please note that the Coulomb theory has a very different interpretation for fracture than for friction, although the equations are the same. I made that comment because is a difference that you should be aware of and not mix-up in the text because it brings a lot of confusion.

✶ *Furthermore, the creation of LKFs in sea ice was already associated with break-ing behavior (Erlingsson, 1991; Marko and Thomson, 1977), the term fracture is repetitively used (Hutchings et al., 2005; Hibler and Schulson, 2000), as well as the fact sea ice is granular medium (Wilchinsky and Feltham, 2011; Hopkins, 1996).*

Which part are you modelling? The ''breaking'' behavior or the granular regime? I assume it is the granular regime, but please make this distinction in the text (see my my comment above).

✶ *Furthermore, for clarity, we would like to keep the same terminology as in the Ringeisen et al. (2019), on which this study is based.*

*In order to address these points, the following sentences were added:*
*• "Note, that in this study, we consider sea ice to be of granular nature not only in the marginal ice zone, but also in pack ice, where ice floes are densely packed. Because sea ice floes are densely packed, we can 2consider the creation of an LKFs as a fracture process with both frac-ture and friction (Wilchinsky and Feltham, 2011)." L26 of the original manuscript.*
*• We modify the penultimate paragraph of the introduction (see also com-ment R2#4). It now reads "In this paper, we investigate the effects of a non-normal flow rule on fracture angles and its use as a means of separating the state of stress (at failure) and the post-fracture defor-mation. The novelty of this paper is that we study the non-normal flow rule in the context of the standard VP rheological model using a similar shape for the plastic potential (i.e., an ellipse) because (1) it is widely used in the community, and (2) its behavior is well documented (com-pared to other models), providing a solid basis for comparison. This paper provides a new generalized theoretical framework for developing any viscous-plastic material with normal or non-normal flow rules. To this end, we test the new model in simple uni-axial loading experiments where the relationship between fracture angle and flow-rule can be easily identified."*

• R1#3, In the same line of ideas, the authors seem to base their assumption of sea ice being a granular material on observations supporting fracture angles that are independent of confining pressure. It appears that they aim at developing a model that complies with these observations. However, no reference of observations, neither at the lab nor the geophysical scale, is clearly associated with this statement. One can reasonably wonder if making such observation would be possible in the case of sea ice at the geophysical scale: how would it be possible to determine far field stresses and distinguish between unconfined and confined states? Do unconfined compression leading to fracture even occur in circumstances other than an individual ice floe crashing into a coast? References are lacking here to support this assumption of independence of confinement and should crucially be added.

*Concerning the granular matter behavior:*
*• Fracture angles (or orientation of the shear bands) that are independent of the confinement pressure are characteristics of granular material, and lead to the use of the Mohr–Coulomb yield criterion.*
*• More recent studies showed that shear bands orientations in granular materials in-crease slightly with confining pressure (Alshibli and Sture, 2000; Han and Drescher, 1993; Desrues and Hammad, 1989, Note that some of these studies show a decrease, but only because they use the complementary angles.). However, this change is very limited: of the order of 5 ∘ , with a stress confinement ratio of in the range [0.05-0.5] depending on the confining pressure and the grain size.*

Please note that at least *Desrues and Hammad, 1989* used sand  in their (3D, not 2D) experiments which is very different asa material than sea ice (in terms of the dispersion of grain sizes, friction, 3D vs 2D), hence you should be carefull with the statement that shear band angles in granular material do not vary with confining pressure.

*• The magnitude of the change of angle contrasts with the effect of confining pressure*
*with the elliptical yield curve, where a stress-ratio of 0.3 changes the fracture from*
*divergent to convergent and the fracture angle from ca. 34 ◦ to 46 ◦.*
*Concerning the sea ice behavior:*
*• The observations of the same fracture angles at different scale (so probably different*
*stress conditions) by several studies (Erlingsson, 1988; Marko and Thomson, 1977;*
*3Cunningham et al., 1994) is an indication that fracture angles might be independent*
*of the stress conditions, i.e. different confining pressures. New datasets of intersec-*
*tion angles from LKFs tracking show that coulombic fracture in the Arctic sea ice*
*shows a predominant angle (Nils Hutter, personal communications)*
*• It is correct that, at high confining pressure, the fracture angle probably changes,*
*especially when sea ice reaches a ridging state. This can be seen with the shape of*
*the yield curve observed in Schulson (2004); Weiss and Schulson (2009). Please see*
*also our answer to Reviewer#2 in comment R2#40.*
*• See also our answer to comment R2#39 of Reviewer#2.*
*• Finally, we agree that far field stresses are difficult (or close to impossible) to de-*
*termine, this is why observing the angle of dilatancy along LKFs could be a good*
*metric to improve sea ice models.*
*To clarify our manuscript, we make the following modifications:*
*• We modify our statement: ". . . namely that shear band orientations and*
*divergent/convergent motion at the slip lines are a function mainly of*
*the shear strength of the material and orientation of the contact nor-*
*mals (or dilatation angle), and that the confining pressure has only lim-*
*ited impact (Alshibli and Sture, 2000; Han and Drescher, 1993; Desrues*
*and Hammad, 1989)", L79 of the original manuscript.*
*• The sentence on L321 would now reads "... unlike laboratory experiments*
*with granular materials (e.g. sand) where the fracture angle is weakly*
*sensitive to the confining pressure (Han and Drescher, 1993; Desrues*
*and Hammad, 1989; Alshibli and Sture, 2000)".*
*• We add the following statement: ". . . A 2D material, such as sea ice,*
*can ridge and "escape to the 3rd dimension" after fracture, therefore*
*we expect a change in the fracture angles at large confinement, when*
*the ice is susceptible to ridging (Schulson et al., 2006)." L325 of the*
*original manuscript.*

Again, my point is that you need to state and explain, **in the text,** the assumptions you make and then refer to the litterature supporting your approach. For instance, here, you start by your main statement, ''we consider sea ice as a granular material'', then, ''and as such we consider that shear bands vary weakly with confining pressure'', citing the references you give here in your response.

I really believe that this will help the reader understand your thought process **and relate to studies they already know of**.

To make the link between studies on granular media and the behavior of sea ice and to support your assumption, it would be highly relevant to include a figure, eg., of the predominant angle you say is observed by Niels Hutter. Would that be possible? Or is it the range  20-25 you later cite in your paper from Hutter and Losh 2020?
Otherwise, citing what you included here in bullet points (''The observations of the same fracture angles at different scale (so probably different stress conditions) by several studies (Erlingsson, 1988; Marko and Thomson, 1977; Cunningham et al., 1994) is an indication that fracture angles might be independent of the stress conditions, i.e. different confining pressures.'', etc) would be a start.

• R1#4, Also somewhat contradictory is the fact that the authors use an elliptical yield curve and plastic potential to model a material that they consider as a granular. I understand this is perhaps temporary and other criterion will eventually be investigated, but in the meantime, are there examples of granular materials that have been observed to follow this kind of yield curve/flow rule? References of such examples would strengthen the paper.

• As the reviewer stated, the use of elliptical yield curve is transitory, but practical
for the main goal of this study: that is, studying the effect of a non-normal flow rule
on the angles of fractures, and provide an theoretical explanation for this effect.
• We use an elliptical yield curve in this study for 2 reasons: (1) Because it is widely
used in the sea ice community, for instance 30 out of 34 sea ice models in GCMs
participating in CMIP5 use the standard VP model or a modification thereof (Stroeve
et al., 2014), and (2) because the behavior of the elliptical yield curve with normal
flow rule in uni-axial compression has been recently investigated (Ringeisen et al.,
2019), and we want to isolate the effects of using a non-normal flow rule.
• Elliptical yield curve, like the Von Mises yield curve, are used in material modeling,
especially for ductile materials. Although their formulation is different that of in
the sea ice models. Granular materials usually use an incompressible formulation,
while sea ice needs a non-zero divergence term to represent open water formation
and ridging.
To clarify our manuscript, we make the following modifications:
• "We use the elliptical yield curve because it is widely used and its behav-
ior better documented than any other models in use in the community.
This provides a known reference to study the use of non-associative
flow rule. We do no aim to propose here a new VP rheology here but
to study the effect of the non-normal flow rule as it could be used in
future rheology." on L335 of the original manuscript.

Thank you for this addition. I would modify the sentence for improved clarity as ''it is widely used for sea ice and its behavior is better documented than any other yield curve used in the sea ice community'' and add ''because the behavior of the elliptical yield curve with normal flow rule in uni-axial compression has been recently investigated (Ringeisen et al., 2019), and we want to isolate the effects of using a non-normal flow rule'' so that the reader understands that your papers are related (and that you want to use the same terminology).

• R1#5, Another concern is in the interpretation of the results. A model of plastic flow is used here, not a model of fracture (neither heterogeneities, nor elastic interactions, nor a mechanism representing breakage of bonds or damage is included here). In such model, one expects the simulated macroscopic behavior (that of the ice floe in this case) to coincide with the theory prescribed at the local scale, i.e., the constitutive equation, flow rule, etc. Therefore, as pointed out by Hutchings et al. (2005), if deviations between the simulated angles and the predicted values occurred, they would be indicative of numerical errors. Hence, while it is good to verify that the model does indeed reproduce the Roscoe angle within a small RMS error, doesn't it just show that the numerical scheme of the model works? This point needs to be clarified in the text. It would also be important to mention what method is used to estimate the angles from fields such as the ones shown on figure 6.

• In sea ice VP rheology, the angle of fracture is not yet understood. For instance,
Roscoe and Coulomb theories gives different angles for the same process. We show
here that the flow rule affects the fracture angles, and we explain this influence with
a theoretical model, adapted from the Roscoe angle. Similar investigations of the
angle of deformation features can be found, for example, in the field of lithosphere
geophysical modeling: Lemiale et al. (2008); Kaus (2010).
• The method used to estimate the angles is presented at the end of Sec. 3.
To clarify our manuscript, we make the following modifications:
• We add on L69 of the original manuscript: "In the case of a non-normal
flow rule, it is unclear which of the three theory (Coulomb, Roscoe,
Arthur) predicts the modeled angle of fracture."
• For comparison and clarity, we add the Coulomb angles predictions on a
new version of Fig. 7a, shown below (Figure 1).
5a)
e F = 0.7
e F = 1.0
e F = 2.0
e F = 4.0
e G = e F
Roscoe
Coulomb

80

70

60

50

40

30

20

0

0

0.5

1.5 2 2.5 3 3.5 4 4.5

Plastic potential ellipse ratio e G

Figure 1: The new Fig. 7a Caption: (a) Fracture angles as function of the plastic potential ellipse ratio $e_G$ for different yield curve ellipse ratios ($e_F$ = 0.7, 1.0, 2.0, and 4.0). The markers with ranges are the mean and two standard deviations of the fracture angles. The dashed lines show the prediction from the Roscoe angle (Eq. 28). The arrows indicate the angles predicted by Coulomb theory, which are constant with respect to $e_G$. Colors indicate the value of $e_F$ for lines and markers. The $R^2$ between theory and modeled angles for $e_F$ = 0.7, 2.0, and 4.0 are 0.97, 0.95, and 0.97.

I will try to formulate my question more concisely : I wonder why, if you prescribe e_G and e_F locally in your model, you do not necessarily expect the macroscopic behavior (in terms of the simulated angle in your rectangular sample) to correspond to your equation 30? What are the reasons why the simulated and theoretical angle could differ, if any?

See my related question below: how does the fracture evolves in your model (in the first 5 seconds of the simulation)?

• R1#6, Finally, I find that a discussion of previous studies that have presented similar interests and analyses is lacking from the discussion. Hibler and Schulson (2000) have indeed implemented a non-normal flow rule in the VP model, using a Mohr-Coulomb yield curve with an elliptical cap ("modified Coulombic" curve). They have also found that a non-normal flow rule affects the orientation of deformation features in the VP rheology. This work is cited in the discussion section, but not really discussed in terms of the differences or similarities between both approaches, nor in terms of the advances of the present study compared to this previous one. I suggest clearly stating that is new here and what is the broad relevance of the results. The model of Hibler and Schulson (2000) has also been used by Hutchings et al. (2005) who have looked at intersection angles. They have compared simulated angles between the modified Coulombic and the elliptical yield curve. Mentioning these previous results and comparing them with the current study would be interesting and would strengthen the literature review and Discussion part of the paper.

• Hibler and Schulson (2000) effectively used a yield curve with a non-normal yield curve. Nevertheless, they link the fracture angles to the slope of the Mohr–Coulomb limbs of the yield curve ($\mu$), and not to the orientation of the flow rule. Also, they did not show an actual fracture creation at high-resolution.
• Hutchings et al. (2005) investigated the fracture angles with the modified Coulombic but did not explain the variations of the fracture angles, and only explained that the difference between theory and experiments comes from numerical convergence.
• In Ringeisen et al. (2019), we also investigated a modified version of the modified Coulombic yield curve.
• An investigation of Mohr–Coulomb yield curve with non-normal flow rule (Ip et al., 61991) in a similar setup is underway, but lies outside of the focus of this work.
To improve our manuscript, we make the following modifications:
• "Previous studies with a non-normal flow rule (e.g., Hibler and Schulson, 2000; Hutchings et al., 2005) did not explore the effect of a non-normal flow rule on the fracture angles." on L97 of the original manuscript.
• "According to Hibler and Schulson (2000), the flow rule may have an effect on the angle of fracture, but the authors limited their case to the framework of flawed ice and did not consider Roscoe's theory of dilatancy. The rheology of Hibler and Schulson (2000) was tested in an

*idealized experiment more complex than ours (Hutchings et al., 2005),
but the effect of using a non-normal flow rule was not explored. The
complexity of their setup may explain the observed difference between
simulated and predicted angles. Note that the rheology in Hibler and
Schulson (2000) was built by changing the shape of the yield curve
a-posteriori, while the rheology presented here solves the constitutive
equations rigorously." on L361 of the original manuscript.
I therefore recommend major reviews to clarify the important points above
before a resubmission. More specific comments that are often linked to these
major comments are listed below.*

Specific comments:
R1#7, Page 1, lines 8-9: "A newly adapted theory (…) predicts numerical simulations of
the fracture angles (…) with a root-mean-square error below 1.3 degrees." This formulation is
unclear and needs rephrasing: a newly adapted theory is implemented in the VP model and
leads to prediction of the prescribed fracture angle with a RMS error below 1.3 degrees"?. Also,
see my main comment about the agreement of the theory with your modeled angles.

*We rewrite the abstract. Also, see our answer to the the main comment R1#5.
The new abstract reads " The standard viscous-plastic (VP) sea ice model with
an elliptical yield curve and a normal flow rule has at least two issues. First, it
does not simulate fracture angles below 30 ∘ in uni-axial compression leading to
a stark contrast with observations of Linear Kinematic Features (LKFs) in the
Arctic Ocean. Second, the tight coupling between the fracture angle, post-fracture
deformation, and the shape of the yield curve was identified as the reason for
this behavior. In this paper, these issues are addressed by removing the normality
constraint on the flow rule in the standard VP model in a uni-axial compressive
loading setup. To this end, an elliptical plastic potential – that defines the post-
fracture deformations, or flow rule – is defined independently of the elliptical
yield curve. As a consequence, the post-fracture behavior is decoupled from the
mechanical strength properties of the ice. In a newly adapted theory – based on
one developed from observations of granular material – the fracture angles de-
pend on both yield curve and plastic potential parameters. This theory predicts
fracture angles well below 30 ∘. Numerical experiments confirm that the flow rule
details determine the fracture angle. For instance, a plastic potential with an
7ellipse aspect ratio smaller than two (i.e., the value of the standard ellipse) gives
fracture angles as low as 22 ∘. Implementing an elliptical plastic potential in the
standard VP sea ice model requires only small modifications to the code. The
model dynamics with the modified rheology, however, are more difficult to solve
numerically. An independent plastic potential provides a solution to two issues
with the standard VP rheology: it allows for smaller fracture angles that fall
within the range of satellite observations, and it decouples the angle of fracture
and post-fracture deformation from the shape of the yield curve. The orienta-
tion of the post-fracture deformation along the fracture lines (convergence and
divergence) is controlled by the shape of the plastic potential. An orientation
that is different from the standard VP rheology requires a non-elliptical plastic
potential. "*

R1#8, Page 1, line 11: I suggest dropping "In conclusion" from your abstract.
*Corrected as suggested*

R1#9, Page 1, lines 14-15: "to make the fracture angle independent of (not on) the confining pressure (as in
observations)". This relates to another of my main comments : what sea ice observations support that fracture angles
are independent of the confining pressure? Please give supporting references. Is it even possible to distinguish between
fracturing processes occurring in confined and unconfined conditions in the sea ice cover at the geophysical scale?
*Please see our answer to the main comment R1#3.
We replace "independent on" by "independent of "*
See my response to R1-3: support for this assumption and references should be included in the text (intro).

R1#10, Page 1, lines 19-20: "narrow lines of deformation observed in the Arctic sea icecover, emerge in high-resolution simulations (Kwok, 2001; Hutchings et al., 2005)". It would be relevant to cite more up-to-date works on high-resolution simulations here. The idea is here to cite the seminal studies about LKFs, we are now also citing more recent literature.

*We add the following references: (Hutter et al., 2018; Koldunov et al., 2019; Heorton et al., 2018).*

Page 1, line 33: LKFs do not emerge only in high-resolution simulations (e.x., 10, 20, 40, + km is sufficient in NeXtSIM) depending on the rheology used. You should modify this sentence accordingly.

R1#11, Page 2, line 23: "The ice strength locally depends on the ice thickness". This is only partially true: local ice strength does not depend only on local ice thickness. This sentence perhaps needs some rephrasing.

*Corrected as suggested*

*The sentence in the revised manuscript now reads: "Locally, the ice strength*
*depends on the sea ice state (thickness, concentration, . . . ), which in turn . . . "*

R1#12, Page 2, lines 25-27: "In granular media like sea ice (...) Note, that in this study, we consider sea ice to be granular not only in the marginal ice zone, but also in pack ice, where ice floes are densely packed ". This again one of my major concern: what is the basis for this assumption? How do you reconcile this assumption with the fact that your goal is to reproduce fracture angles in sea ice? Does pack ice, newly-formed ice or any ice that is not yet fractured into floes or constituted of agglomerated, refrozen floes always present the characteristics of a granular media? Please explain and also give some support for this assumption.

*We argue that yes, "pack ice, newly-formed ice or any ice that is not yet fractured into*
*floes or constituted of agglomerated, refrozen floes" still carry granular characteristics. The*
*anisotropy at subgrid scale is still present in a way that fracture will rarely be created in*
*8straight lines, but will most probably follow the network of weaknesses.*

Agreed, but non-straight fracture lines are not a characteristic of granular material only: they occur in any heterogeneous quasi-brittle material. See my response to R1-3: you need to state clearly that sea ice present both brittle and granular behaviors and in which of these regimes you place your study.

R1#13, Page 2, line 28: "This anisotropy". This is unclear. Please define this anisotropy and better explain how it emerges.

*We modified the text: "The intersection angles between the LKFs have an influence on the deformation field and, hence, on the local sea ice strength and the emergent sea ice anisotropy (Aksenov and Hibler, 2001). This anisotropy, which emerges as sea ice develops weak and strong areas along LKFs as leads or ridges form locally, then influences . . . "*

R1#14, Page 2, line 37: The brittle model used in (Rampal et al., 2016) is the EB model of Girard et al. (2011). Please modify the reference.

*Corrected as suggested by the reviewer.*

Page 2, line 70: The rheology in Rampal et al., 2016 being the same as in Girard et al., 2011, I would remove the reference to Rampal et al., 2016 (repetition).

R1#15, Page 2, line 39: I believe a simpler and scientifically more objective formulation would be "most widely used ", instead of "de facto standard ".

*"De facto" means "in fact" or "in effect". We are just stating a fact here.*

Page 2, line 72: I still think that an objective sentence would replace ''standard'' by ''most widely used'' (your next sentence supports just that) or *de facto* by ''practically''. It is not *a fact* that the sea ice community has defined a standard rheology :)

R1#16, Page 2, lines 48-49: Yes, granular media indeed present shear bands, which are not the same as fractures. Again, please clarify what you want to represent in your model. What is the link between LKFs in sea ice, shear bands in granular media and fractures in solid materials?

*See our answer to the comment R1#2.*

R1#17, Page 2, lines 48-49 vs line 50: "Two classical solutions coexist and set two limit angles for the orientation of fractures: the Coulomb angle (...)". There is something unclear and contradictory between this and the previous sentence. You invoke the Coulomb theory here, in the context of friction or fracturing? I understand it is the later, but please make that clear by answering my previous comment.

*We consider the case of fracture, but this applies also a dense pack of ice floes. We do not understand why these two concepts should be separated. The creation of LKFs in sea ice has been referred to as "fracture" in several preceding publications (e.g., Hutchings et al., 2005).*

The Coulomb theory (originally for friction) has been adapted and extensively used to describe fracturing in brittle materials, but these are two completely different phenomena (friction and fracture) and so is the interpretation of this theory in terms of angles. This is why these two concepts should be separated. See my response to R1-3: you just need to state more clearly in a short sentence what you are describing: shear bands in a granular media or brittle fracturing, so that the reader follows your line of thought.

R1#18, Page 3, line 56: I think it would be relevant to make some space and re-introduce the definition of the dilatancy angle here: it would make life easier for the reader and avoid the need to dig for it in another article.
Added as suggested
*We add the following sentences "Dilatancy refers to divergence along along shear bands or LKFs that is a function of the distribution of contact normals between individual floes at the sub-grid scales. A positive angle of dilatancy is associated contact normals that (on average) opposes the macroscopic shear motion and divergence along the shear band; while negative dilatancy is associated with a closing of the fracture line and ridging " on L55 of the original manuscript.*

R1#19, Page 3, line 58: "A general theory derived from experiments with sand that takes into account both the angle of friction (...)". In the case of sand, contact and friction are indeed at play and shear bands are formed. This again adds to the confusion: internal angle of friction or angle of friction? i.e., fracture or friction? Please clarify.
*Please see our answer to the major comment R1#2*

R1#20, Page 3, line 60: based on the grain size.
*Corrected as suggested*

R1#21, Page 3, lines 67-68: "a larger dilatancy angle implies a larger grain size, more contact normals, hence more friction". Can you please include some references that support this?
*We add a citation.*
*We now refer to Vermeer (1990) at the end of the cited sentence.*

R1#22, Page 3, line 73: There is a mistake here, as Weiss and Schulson (2009) reported observed fracture angles between 20 and 50 degrees. Or did you derive this directly from their estimated internal friction angle, which is fit to in-situ stress measurements? In the later case, this is then not an observation of fracture angles but a derivation based on some physical assumptions, which are moreover debatable (see Dansereau et al. (2019) and many others), and it should be removed from the list of observations of fracture angles.
*Corrected as suggested.*

R1#23, Page 3, lines 74-76: You state that uni-axial compression experiments showed that (3) the fracture angle is a function of the confining pressure. How did you determine that without performing bi-axial compression experiments? Is there a typo here?
No, this is no typo. Ringeisen et al. (2019) showed that the fracture angles changes with the confining pressure when a elliptical yield curve is used, the forcing was uniaxial but the ice was confined, hence similar to a bi-axial loading. We modify the text to now read: "the fracture angle is a function of the con-fining pressure. The confinement was achieved by adding thinner ice on either side of an ice slab subjected to uni-axial loading."
I see. It would be clearer and shorter if you wrote ''compression experiments with uni-axial loading and lateral confinement added via the addition of thinner ice (Ringeisen et al., 2019)'' because uni-axial compression experiments with confinement are in fact bi-axial compression experiments.

R1#24, Page 3, line 75: the ''gradient'' of shear to compressive strength. Did you mean the ratio?
*The fracture angles does not depend on the ratio, but the slope of the tangent to the yield curve (Ringeisen et al., 2019; Pritchard, 1988). This slope determines where the ice will break on the Mohr's circle of stress, i.e., the fracture angle.*
*We changed the sentence to: ". . . the angle of fracture is a function of the gradient of shear strength with respect to compressive strength (i.e. the slope of the yield curve) . . . "*

R1#25, Page 3, line 76-79: See again my major comment about the apparent confusion between fracturing, friction, granular media, sea ice and a viscous-plastic continuum rheology.
I think it is crucial to clarify the links you make between these processes and the motivation of your approach here. This passage in particular leads the reader to believe that your goal is that the VP rheology complies with observations of granular media behavior, because you consider that sea ice at the geophysical scale, in all its different states, is a granular media. If this assumption is at the very basis of your approach, it should be stated earlier in the introduction, (very importantly) along with supporting arguments. This would make the reading and the assessment of your assumptions and methods by the reader much easier.
*See our answer to the comment R1#2.*

R1#26, Page 3, line 82: "The ratio of shear to divergence along the LKFs allows to infer the dilatancy angle." Again, if one assumes sea ice in any state behaves as a granular material.
*We clarify this in the revised manuscript. It is important to note that dilatancy (dilatancy can be positive or negative) in leads is a known fact. If most of the deformation happens in*
*shear, LKFs play a predominant role in thick ice formation (ridging) as well as in thin ice formation*
*"The ratio of shear to divergence along the LKFs allows to infer the dilatancy*
*angle when considering sea ice as a granular material."*

R1#27, Page 3, lines 86-87: "Separating the link between the fracture angle and the flow rule from the yield curve is necessary to design rheologies that are consistent with observed sea ice deformations". Please note that this would be only true for plastic flow rheologies and not applicable nor necessary for rheologies based on elasticity (EB, MEB, Elastic-Decohesive). To be objective, this statement should therefore be modified as "necessary to design plastic flow
rheologies that are consistent (...)".
*We correct as suggested.*
*the sentence in the revised manuscript now reads ". . . to design VP rheologies*
*that . . . "*

R1#28, Page 4, line 90: ''In these different classes of models, various rheologies can be defined ''. This is not true and/or not clear: these are rheological models and therefore they do not include different rheologies. I think that you mean that these different models require the definition of different components: a constitutive relation (all models), a yield/damage curve/criterion (all models including a threshold mechanism, i.e, a change in mechanical be-havior) and a flow rule (only plastic flow models). I therefore suggest to rephrase and clarify this passage and the next sentence, that is "in a VP rheology, a yield curve and plastic potential (flow rule) must be defined ''. In the same line of idea, I do not really see the point of the last sentence of this paragraph. Maybe it can be cut if some rephrasing is made at the beginning of the paragraph?
*A VP model with a different yield curve and/or a different flow rule can describe a different*
*physics in the modeled material. A VP rheology with a Mohr-Coulomb yield curve (e.g. Trem-*
*blay and Mysak, 1997) will create different results than the one with an elliptical yield curve.*
*The last statement is important for this paper, because it stresses the fact that changing the*
*flow rule changes the system dynamics.*
Again this is not clear: a rheological model has its own rheology, that determines if it is elastic, plastic, viscous, etc. A model with a different yield curve will lead to different results with the same constitutive equation indeed but does not change the relationship stress-deformation. The flow rule problem concern plastic models only. The sentence should therefore read ''In these different classes of models, various mechanical components can be defined" or ''in the VP sea ice model, various yield curves and flow rules can be defined''.

R1#29, Page 4, lines 96-97: See my major comment above. Hibler and Schulson (2000) have indeed used a VP model with a non-normal flow rule and a Mohr-Coulomb yield curve with elliptical cap, or "modified Coulombic" curve, as cited in your Discussion section. This model has also been used by (Hutchings et al., 2005) (https://doi.org/10.1175/MWR3045.1) who have looked at intersection angles and compared them between the modified Coulombic and the elliptical yield curve. As their approach is therefore close to yours, it would be important and certainly interesting to explain the similarities and difference between your work and theirs in the literature review (introduction) section. Please also note that Hibler and Schulson (2000) do not seem to share your view that the angles of fracture in sea ice at the geophysical scale are independent of confinement, which would be an important point to discuss further.
*See our answer to the major comment R1#6.*

R1#30, Page 4, line 100: "viscous-plastic materials" or "a viscous-plastic material ",
"with any flow rules".

*Corrected as suggested*

R1#31, Page 4, line 100: "from the yield curve".
*Corrected as suggested*

R1#32, Page 4, lines 101-102: "The new model is tested in simple uni-axial loading experiments". See my major comment above: a quick addition to your work would be to test if your numerical implementation also holds under bi-axial loading conditions, that is, if the angles vary or not with confinement.
*See our answer to the general comment R1#3 as well as comment R2#2 and R2#39 from*
*Reviewer #2*

R1#33, Page 4, line 108: ''We consider sea ice as a 2D viscous-plastic material". See my previous major comment: please explain the physical link between this viscous-plastic assumption and that of a granular material.
*See our answer to the general comment R1#2*
This comment is not clearly answered in R1-2 and should be included somewhere in the introduction (see my major comment above).

R1#34, Page 4, line 113: In your case, the constitutive equation links the vertically integrated stress tensor to the deformation rate, which you introduced on the previous line.
*Yes exactly. For clarity, we prefer repeating "stress tensor ". However the term "rate" with*
*"deformation tensor " was missing.*
*We modify as suggested: "The constitutive equations link the vertically inte-*
*grated stress tensor σ to the strain rate tensor #"*
.

R1#35, Page 4, lines 117-119: Representing small deformations with a viscous model is rather counter-intuitive, especially for a reader that is familiar with viscous-plastic rheologies (plastic for small, viscous for large deformations). I believe it is important that you explain in more details how a viscous rheology is expected here to represent the small deformations of a solid (time scales, viscosities, etc).
*Effectively, this VP models differs from other Viscous-Plastic models, e.g. Bingham plastic,*
*which include a yield condition (rigid solid) and then deforms as a viscous plastic with a linear*
*relationship between viscosity and strain. We add more details to our description of viscous*
*behavior in the last paragraph of Sec. 2.1, on L155*
*We add the following text on line 157 of the original manuscript "VP sea-ice*
*models typically cap the viscosity at*
*ζ max =*
*#*
*#*
*P*
*= 2.5 × 10 8 s · P*
*2Δ min*
*12and η max = ζ max*
*to regularize the momentum equations. When this regulariza-*
*e 2 G*
*tion is in effect, ζ and η are independent of the deformation field (Δ) and*
*the stress divergence reduces to harmonic viscosity with constant coefficients.*
*Δ min = 2 × 10 −9 s −1 (Hibler, 1979, 1977) translates to a deformation time scale of*
*almost 16 years. Therefore, viscous deformations are slow and negligible with*
*respect to the plastic deformations, and VP rheologies are almost purely plastic.*
*The viscous behavior is a consequence of regularizing the viscosities rather than*
*an implementation of a physical behavior."*

R1#36, Page 5, line 130 to page 6, line 149: These paragraphs could be shortened by removing or presenting in a more concise manner some general pieces of information.
*We would like to keep it in the present form because we think it is a useful description of*
*VP rheology.*

R1#37, Page 5, lines 130–131: As it is not the states of stress that are deforming plastically, but the material, this sentence needs some reformulation.
*Corrected as suggested by the reviewer.*

*"The yield curve represents the stress states for which sea ice deforms plastically while enclosing the stress states for slow viscous deformation."*

R1#38, Page 9, line 204: "The slope of the yield curve". And many other missing "the" throughout the text.
*Corrected as suggested. We thank the reviewer for pointing all these out to us.*

R1#39, Page 10, line 223: How does the no-slip condition at the bottom boundary affect your results compared to the case in which slip is allowed in the x-direction (i.e., by holding only one of the two bottom corners of the domain fixed in x and y)? Such boundary conditions are maybe less representative of a floe that sticks to a coast but would not lead to as much concentration of stresses on the bottom corners of your ice floe (here your Bcs imply some bi-axial compression at the bottom) and hence would put less constraint on the appearance of conjugate faults and on their orientation. I think this would be an interesting and not time-consuming test.
*In Ringeisen et al. (2019), we already investigated the effect of the no- and free-slip condition, and we showed that the configuration used here does not influence the angle of fracture, as indicated on L248 on the original manuscript.*

R1#40, Page 11, line 240: I suggest "more numerically challenging".
*Corrected as suggested.*

R1#41, Page 11, line 256: ''laboratory experiments". If you compare your results with laboratory experiments, please provide more details on these experiments (e.g., boundary conditions? biaxial or uni-axial compression? on samples with an aspect ratio similar to sea ice, i.e., virtually 2D? on fresh or sea ice?) Were such experiments made by Erlingsson (1988) and Wilchinsky et al. (2010)?
*Corrected as suggested by the reviewer.*
*In the corrected manuscript, this sentence now reads: "The fractures forma diamond shape, similar to the shapes observed at large scales (Erlingsson, 1988), in laboratory experiments (Schulson, 2001), and modeled with DEM models (Wilchinsky et al., 2010) or other continuous sea ice models (Ringeisen et al., 2019; Heorton et al., 2018)."*

R1#42, Pages 11-13 and caption of figure 6: What is the field represented in figure 6? I assume from the color scale that it is a deformation rate?
*The field shown here is the shear deformation $\dot{\epsilon}_{II}$.*
*We clarify this in the caption: "Diamond-shaped fracture pattern in the shear deformation field $\dot{\epsilon}_{II}$ for $e_F = 2.0$ and three different values of $e_G$ after five seconds of simulation."*

R1#43, Section 4 and figures 6 and 7: How are the angles of the features observed on fields such as shown on figure 6 measured, i.e., estimated? It would be important to mention what method is used.
*This is described in Section 3 Experimental setup and numerical scheme, Line 245 to Line 250.*
Please add a short description on **how** the GIMP Measure Tool estimates (automatically or not?) the angle from the simulated fields (method, errors?).

R1#44, Result section, figure 7 and page 15, lines 292 and 306-308: "the theory predicts the fracture angles accurately" and "The results illustrate clearly how the yield curve defines the stress for which the ice will deform, that is, the transition between viscous and plastic deformation, and how the relative shape of the plastic potential with respect to the yield curve defines both the type of deformation (convergence or shear) along the fracture line and the fracture angle. The resulting fracture angles are in excellent agreement with the Roscoe angle predictions (Roscoe, 1970)." There is my major comment about the results. In section 2.3, you describe how the yield curve, flow rule and angles are related in your model. By prescribing the yield curve and plastic potential ellipse ratios, you prescribe locally the angle (Roscoe) of "fractures". Figure 7 shows that at the macro-scale, i.e., the scale of the ice floe you indeed retrieve that angle. What is prescribed at the local scale is what you get at the macro-scale in your model, as expected in a model of plastic flow. Therefore my understanding is that these tests serve to verify that your numerical scheme is OK. Is that the case? To better illustrate that point, it would be relevant to show the (deformation?) fields at different stages of the compression experiment, to illustrate how the features arise in your model.
*We show the fracture after 5 seconds of simulation, in order to get the initial fracture and avoid more complex interactions that might create more fractures (see Fig. 6 in (Ringeisen et al., 2019)). Please see our answer to the general comment R1#4*
The sense of my question was : what happens within the first 5 seconds of the simulation? (see my major comment above).

R1#45, Page 15, line 300: ''the shape of the plastic potential''.
*Corrected as suggested.*

Page 15, line 305: "this allows decoupling the mechanical strength properties of thematerial (ice) from its post-fracture behavior". Again the contradiction with the assumption of a granular material, i.e., an already fractured/fragmented material. How do you reconcile these ideas?
*See our answer to general comment R1#2*

R1#46, Page 15, lines 306-308: ''The results illustrate clearly how the yield curve defines the stress for which the ice will deform, that is, the transition between viscous andplastic deformation, and how the relative shape of the plastic potential with respect to the yield curve defines both the type of deformation (convergence or shear) along the fracture line and the fracture angle. The resulting fracture angles are in excellent agreement with the Roscoe angle predictions (Roscoe, 1970)." But you prescribe the yield and plastic potential in your model: why would you not expect what you get to indeed be what you prescribe? In other words, you do not make any distinction between what you prescribe at the micro-scale (scale of your discretization) in your model and your macroscale results and you do not discuss why you expect these behavior to be identical or not : that is missing from your work and interpretation of your continuum model.
*See our answer to general comment R1#5*
Please see my major comment above.

R1#47, Page 15, point 2: About confinement, shear bands and fractures, see my major comment above.
*As for the other comments raised about relationship between fracture angles and confinement (R1#3, R2#2), this behavior is linked to the elliptical nature of the yield curve. We add a reference to our study showing how the confinement changes the fracture angles with an elliptical yield curve: "This behavior cannot be eliminated with an elliptical plastic potential, as the normal stress along the LKFs increases with confining pressure and the flow rule changes from divergence to convergence (Ringeisen et al., 2019)."*

R1#48, Page 17, line 382: "sea ice mechanical strength properties (yield curve) and deformation (flow rule)". Again, you write this with the perspective of a VP model, but mechanical strength properties and deformation are not only determined by the yield criterion and flow rules in other rheological models for sea ice. Please be specific and make this distinction clear. Also, I do not understand why Dansereau et al. (2016) is cited in this context.
*We refer to Dansereau et al. (2016) in this context because the way the damage parameters*
*act as the history of the model deformation is very interesting, and could be a representation of*
*the state of the local ice (broken/unbroken), i.e. "sea ice mechanical strength properties (yield*
*curve)" cited before.*
*We reformulate the sentence on L382 of the manuscript ". . ; the sea ice me-*
*chanical strength properties (i.e., yield curve) and deformation (i.e., flow rule*
*for VP rheologies) should vary in time and space depending on, for example, the*
*time-varying distribution of the contact normals, floe size distributions, or the*
*damage parameter, as per observations and laboratory or numerical experiments*
*(Overland et al., 1998; Hutter et al., 2019; Horvat and Tziperman, 2017; Roach*
*et al., 2018; Balendran and Nemat-Nasser, 1993; Dansereau et al., 2016; Plante*
*et al., 2020)."*
I see your point, but the numerical experiments in Dansereau et al., 2016 does not show that mechanical properties involved in the damage criteria should depend on the damage itself. Instead this dependancy was not added in the model because of lack of agreement in (e.g., experimental) supports, hence my reaction to this sentence. Maybe this sentence is just not clear and should be rephrased?

R1#49, Page 17, lines 387-388: "So is the combined knowledge of the failure stresses and their associated deformation of sea ice as a 2D granular material". This is confusing: why then do you base your approach on the assumption of a granular material? This goes along my main comment and really needs to be clarified.
*If deformation data are available from satellite observations, we still have little knowledge*
*about the stress associated to these observations. This is especially true when these deformation*
*lead to ridging and creation of open-water. Also, most of the laboratory data investigate 3D*
*continuous ice, we are not quite sure if these results can be extrapolated to sea ice, i.e. we are*
*15missing knowledge about 2D fractured materials behavior. See also our answer to comment R1#2.*
*We reformulate ". . . higher temporal resolution of sea ice deformation and flow*
*size distributions is still unavailable. There is also a knowledge gap in the inter-*

*play between yield stresses and the post-fracture deformation in a 2D granular material such as sea ice. This interplay is likely different than for the well studied case of a solid homogeneous 3D block of ice (e.g. Schulson, 2002)." on L387 of the original manuscript.*

**Other minor comments**

Page 1, line 32: ''In granular media like sea ice''... then ''Note that in this study, we consider sea ice to be of granular nature''. See my response to your answer to my major comment above. You should first state that you make the assumption that sea ice is mostly of granular nature and give some references supporting this assumption. Hence reverse the two first sentences here.
And then next sentence : ''For this reason, we  consider here...''

Page 2, lines 40-42: To avoid repetition: ''Other models represent sea ice (...)'' and then skip VP as an example.

Page 2, line 43: ''In these different classes of models, various rheologies can be specified''. ? This sentence still does not make sense. I think it could be just removed without impacting the text.

Page 2, line 44: ''The yield curve defines the stress criteria for the transition from  viscous deformations to  plastic deformations''. The deformations are not necessery small or large so I would remove these adjectives. Also, I would add at the beginning ''In the VP sea ice model'' so that the reader understand that this is inverted compared to standard visco-plastic rheologies (see my previous comment about this). Another solution is to move the sentence on page 2, lines 53 to 55 here to make this distinction clear.

Page 2, lines 49-50: ''It is important to note that two PLASTIC models ...''

Page 3, lines 90-91: Again, and in agreement to your response to my main comment, I would not focus too much on sand and would remove this sentence, which is I beleive just an example of the previous one.

Page 4, line 93: ''The theory'' change to ''the concept''?

Page 4, lines 112-113: ''in contrast to observing stress which requires in-situ measurements''. This comparison is not really relevant here or it would need a longer description. I think it could simply be cut to make the text shorter.

Page 4, line 122: ''for comparision'' add *with previous simulations*.

Page 4, line 123: ''sea ice as a granular material ''.

Page 4, line 125: ''uni-axial compression  VP simulations''.

Page 5, line 141: ''In an ideal plastic model, the stresses are independent of the strain rates''. This part of the sentence could be cut (VP is by definition not an ideal plastic but a viscous-plastic model so this is implicit).

Page 5, line 152: ''Some other state variables are a function of P ; for instance, the tensile strength T is usually defined as T = k t · P , where the tensile factor k t > 0 (König Beatty and Holland, 2010). Others are not, such as the ellipse aspect ratio (Hibler, 1979) or the internal angle of friction (Ip et al., 1991)'' Thes sentences is not directly relevant to what you do in your paper and could be cut.

Page 5, line 154: ''For two-dimensional sea ice, stress is a rank two tensor; thus, it has four components.'' This is very generic and can be cut: if you have mentionned that you consider sea ice to be 2D in the intro it is already implied.

Page 6, line 158: ''The yield curve can be represented in principal stress (σ 1 and σ 2 ) or stress invariants space (σ I and σ II ).'' Again, this can be cut.

Page 7, line 190-191: ''and VP rheologies can be considered as ideal plastic''. Please make the distinction! only the (converged) VP model for sea ice could be considered as ideal plastic, on the time scales relevant for sea ice modelling, **not** standard visco-plastic rheologies (like I said, the viscous vs plastic behavior is inverted with respect to sea ice VP).

Page 12, line 294: ''The angle of each fracture ''.

Page 18, lines 420-423: Thank you for adding a comparision to the work of Hibler and Schulson, 2000 and Hutchings et al., 2005. However, I think it would be more relevant to present it in the introduction, to position your work in the context of what has been done in the sea ice community.

---

## Editor Decision (ED1)

This paper describes the implementation of a non-normal flow rule in the VP sea ice rheology. The equational form of the new rheology is well described and several very useful diagrams are included. The numerical implementation is linked to a theory that links the flow rule and the intersection of failure lines within the medium described. A series of idealised numerical experiments are performed which show that the numerical rheology successfully recreates the fracture intersection angles predicted by the presented theory. The authors follow the experiments with a discussion on the implications of using a non-normal flow rule when designing future sea ice rhelogies. They describe the various challenges when using non-normal flow rules. I find that this paper is well written and a valuable contribution to the modelling of sea ice deformation. It is a very useful introduction to use of non-normal flow rules for sea ice modelling for future work in this area. I recommend this paper for publication after a few questions I have.

First of all can you explain why figure 7a contains both theoretical links between the plastic potential and intersection angle and many numerical experiments that back up the theory but 7b contains relatively few numerical results? I can see several cases where additional results from 7a can be copied to 7b and back up your results. Is it true that the full range of values for 7b are not obtainable due difficulties that the authors discuss in getting the model to converge to a solution for highly non-normal flow? If this is case then please tell us.

Several times in the discussion and results the authors say that the intersection angle depends on the confining pressure despite the varying non-normal flow rule. I can see no evidence of this in their results. The presented experiments show changing intersection angle with changing flowrule (varying plastic potential and yield curve eccentricity), but I see no results where they change the confining pressure. Is this from previous work? Or an interpretation of the results that they do present?

General editing points:

Can you please start the paper with a description of what a flow rule is. Then what a normal flow rule is, and the crucially what the main difference physically and theoretically is between a normal and non-normal flow rule. I see that a definition is on line 90, and then further physical descriptions of the flow rule are in the results. The introduction make much more sense if these can come first.

Can you describe what is documented in this study that is novel and new?

L 20 they are also, more importantly, observed

L 21 Here you LKF's influence in many ways but what follows is not a list. Consider re-writing

L 22 Please define what a lead is. Consider starting with a definition of LKF's that are typically leads or ridges

L 70 Which is the 'standard rheology'? do you mean the VP rheology. Also can you further describe this result. How did Ringeisen find that the angle can't be lower than 30 degrees?

L71 the following list is hard to read. Consider reformatting. Also what does the  $\mu$  = 0.9, relate to with the Weiss and Schulson reference.

L71 can you confirm that these angles are all comparable? I have found that studies document both the intersection and also the half angle, being the intersection between the fracture and the principal axis of stress.

L80 this paper require a definition for a normal flow rule. This sentence and the following paragraph make little sense without it.

L82 do you mean that the flow rule can be observed by measuring the ratio of shear a divergence along LKF.

L85 were these laboratory observations performed the same way as those of Stern mentioned above?

L89 it will be nice to have the Anisotropic Plastic (Tsamados 2013, 10.1029/2012JC007990) rheology listed here too

L92. Good to see a flow-rule definition here. How does the plastic potential determine the postfracture deformation? is this through the direction of the principal stress when the yield criterion is reached?

L 115 is f here the Coriolis acceleration as above? Actually can you tell what value was used for the Coriolis acceleration? If it is non-zero (valid to use zero and non-zero for these experiments) then asymmetry will be expected (see comments later)

L120 It is great to read this description of the VP rheology. A really helpful addition.

L 138 is it possible to have a physical description of the plastic potential here? The physical description of what the yield curve represents is very helpful. A similar description of the plastic potential here will be similarly useful. The flow-rule is difficult concept that is explained well here. An additional physical description will make it even better.

L 180 I see that the dilatancy angle was introduced earlier. However it would benefit the paper to include a physical description of 'dilatancy of a granular material' either before or here when it is implemented in the model equations.

L 180 and onwards. This section will benefit from an expanded introduction to the theoretical steps performed. From what I can tell, you use the theory that links dilatancy angle to fracture angle as discussed in the the introduction. You have quantified the dilatancy angle using geometrical description of an arbitrary yield curve and plastic potential. This is expanded through the notation to express the fracture angle as a function of yield curve and plastic potential eccentricity. Is this correct?

If so is the motivation behind the description that it is possible to show how the expected fracture angle is expected to change with changing plastic potential?

Can you be clear what the theory of Roscoe is describing. Is the angle you are obtaining the expected angle of fracture due to minimising some sort of energy potential? Or does it relate to an analytical solution of fracture? The mathematical expansion here is clear to follow, but the reasoning behind why you have shown it is less so.

In figure 4 you describe how the ratio of divergence to shear changes with changing plastic potential. Is this the key effect of the non-normal flow rule? In that by separating the yield curve and plastic potential it is possible to change the ratio of divergent to shear stresses whilst under deformation? But without also change the point of deformation (as in the yield curve) If so please emphasise this point throughout the paper! It makes the non-normal flow rule much clearer for me!

Figure 3 caption - the arrows are described as orange, but appear red to me.

Figure 4. I see red and orange arrows here, and they are correctly described. Can you check figure 3. Do the colours relate between the two figures?

L 222 is the initial ice state entirely uniform? Or did you seed some noise into the initial state? L 231 did you test at other time and spatial resolutions? Later you comment that fracture angles were shown in a previous study to be independent of model resolution (we found this too). Did you test this for this study too?

L232 is this equation 4 that is solved for?

L233. What are the non-linear and linear problems ? Can you relate these back to the model equations?

L246 So are the simulations only run for 5 seconds of model time? Have you tested how long the model can run for and its overall stability? I read above that you have used excessive computation to ensure the extra complexity of the non-normal flow rule is accounted for. How successful is this approach? Did you find that certain computational setups did not perform well when attempting to solve the equations? Any insight you can share into how to solve these equations will greatly help the sea ice modelling community

L 263 what is average residual norm R? is this a measure of the solution accuracy?

L 282 is the shear strain rate shown anywhere? Are you relating back to figure 6? If so can you say so? Are you saying the relation ship in figure 6 for eF and shear strain rate is also true for the various values of eF in Figure 7? Or is this a theoretical postulation?

L 282 fracture angle or angles plural? Do you you take multiple angles or just one per simulation?

Figure 7 Is it possible to add the red orange and teal umerical simulations to figure 7 b? If you have added the blue dots then the omission of the others makes me wonder how they will fit? I see that you only have multiple values for eG = 4.0. Though there are 2 points for 0.7 and a single point for 2.0 and 1.0. I also see that the full range of eF was not investigated for each eG. What is the reason for this? Is it the limitations of the model? Or did you choose not to in order to keep the simulations physically relevant?

L 305 this line is very informative to what the non-normal flow rule can achieve. Can you put this information into the introduction and abstract please?

L 309 while you have displayed the agreement to Roscoe for the cases of constant eF the case of constant eG (fig 7b) is inconclusive to the reader due to the lack of numerical simulation data points. Is it possible to fill out figure 7b and thus strengthen this statement?

L 313 Can you sort out the parenthesis on the Ringeisen 2019 citation. It currently doesn't read very well.

L 317 is this lack of convergence the reason for the lack of results on figure 7b?

L 319 Can you give a citation a description of how this result with the changing fracture angle with changing stress confinement was obtained? I assume it is not from this study as you have not altered the confinement ratio for any of your simulations. Or are you referring to that the fracture angles change as the loading increases with time?

L 321 How do think this result relates to to laboratory experiments on sea ice where two clear fracture angles were found about a critical confinement ratio? (Golding et al. 2010 1359-6454/\$36.00, Schulson 2001 10.2138/gsrmg.51.1.201)

L 341 Is this result about pure shear and angle of 45deg. from the Ip et al. 1191 citation? How was it obtained?

L345 angle - angles

L 363 Is it possible to include a diagram of the various yield curves discussed in this section? This would greatly ease the understanding of your arguments. I'm sure others have included such a diagram in previous work so you may be able to cite such a diagram.

L369 Can you explain why non symmetrical deformation features are unrealistic or present an incorrect solution? Do they also correspond to poor numerical solutions? With a non-linear system of equations such as in all sea ice rheologies, asymmetry is often expected. This relates back to most laboratory experiments on ice deformation and even the ill-posedness of divergent weakening (Gray 1999 10.1175/1520-0485(1999)029<2920:LOHAIP>2.0.CO;2). Also if you use a

non-zero Coriolis acceleration then asymmetry will be expected as the run progresses. What value did you use?

L371 I'm not sure I understand your argument here. Are you saying; poor non-normal flow model convergence won't be an issue in realistic simulations as the numerical solver can't solve the VP rheology anyway? Surely this argument says that there isn't a hope of using non-normal flow VP rheology in realistic simulations?

L396 These issues are not exclusive to high resolution climate modelling. It can be argued they are even more important for current coarse resolution models which are currently used for long climate simulations and typically perform poorly for reproducing ice drift patterns. LKF intersection angles are also observed over basin length scales (Weiss and Schulson 2009) and your discussion in this paper is relevant for modelling sea ice deformation at these length scales.

L406 I am confused by your conclusion here. Where have you shown that the fracture angles depend on the confinement pressure? Where did you change the confinement pressure? Do not Figure 6 and 7 show clear changes in intersection angle with changing plastic potential in accordance with predictions from the theory of Roscoe?

L 409 again I'm not convinced that symmetric solutions are mandatory for a symmetric experiment? Again can you say whether you used a zero or non-zero value for the Coriolis acceleration? If it is non-zero then asymmetry will be expected.

This paper presents an implementation of a non-normal plastic flow rule in a Viscous-Plastic model with the goal of better representing the observed angles between Linear Kinematic Features in sea ice at the geophysical scale. The paper is overall well written, in a pedagogical way for the theory (section 2) section, which could however be a little more concise in some places. The figures are, for the most, clear. Here are my major comments/concerns :

- It does not appear clear in the paper what physical process(es) the authors really want to model. In the introduction, it is mentionned that sea ice, both in the pack and the marginal ice zone, is considered as a granular material. No physical justification is offered for this assumption. The rheology used to model this granular material is one of plastic flow, but the authors do not explain how they reconcile their continuum viscous-plastic model with a granular behavior. The aim is apparently to reproduce *fracture* angles (repeated terminology for the features simulated by their model), but the authors do not explain the link between plastic flow, fracturation and the mechanical behavior of a granular material, which is an already fractured/fragmented material in which contacts and friction dominate. Later, it seems that the authors refer to shear bands in granular materials as if they were associated with the same processes as a fracturing solid. The Coulomb theory is invoked but it is not clear if it is in the context of friction or fracture. There is therefore much confusion thoughout the paper as to what the authors consider is the mechanical behavior of sea ice : is it caracterized by fracturation? By friction and contacts between already broken up floes? Granular materials like sand are invoked, but is sea ice really assimilated to a sand-like material here? Whatever is assumed, it crucially need to be clarified and all physical concepts untangled throughout the text in a way that makes physical sense.
- In the same line of ideas, the authors seem to base their assumption of sea ice being a granular material on observations supporting fracture angles that are independant of confining pressure. It appears that they aim at developping a model that complies with these observations. However, no reference of observations, neither at the lab nor the geophysical scale, is clearly associated with this statement. One can reasonably wonder if making such observation would be possible in the case of sea ice at the geophysical scale: how would it be possible to determine far field stresses and distinguish between unconfined and confined states? Do unconfined compression leading to fracture even occur in circonstances other than an individual ice floe crashing into a coast? References are lacking here to support this assumption of independance on confinement and should crucially be added.
- Also somewhat contradictory is the fact that the authors use an elliptical yield curve and plastic potential to model a material that they consider as a granular. I understand this is perhaps temporary and other criterion will eventually be investigated, but in the meantime, are there examples of granular materials that have been observed to follow this kind of yield curve/flow rule? References of such examples would strenghten the paper.
- Another concern is in the interpretation of the results. A model of plastic flow is used here, not a model of fracture (neither heterogeneities, nor elastic interactions, nor a mechanism representing breakage of bonds or damage is included here). In such model, one expects the simulated macroscopic behavior (that of the ice floe in this case) to coincide with the theory prescribed at the local scale, i.e., the constitutive equation, flow rule, etc. Therefore, as pointed out by *Hutchings, Heil and Hibler, 2005*, if deviations between the simulated angles and the predicted values occured, they would be indicative of numerical errors. Hence, while it is good to verify that the model does indeed reproduce the Roscoe angle within a small RMS error, doesn't it just show that the numerical scheme of the model works? This point needs to be clarified in the text. It would also be important to mention what method is used to estimate the angles from fields such as the ones shown on figure 6.
- Finally, I find that a discussion of previous studies that have presented similar interests and analyses is lacking from the discussion. *Hibler and Schulson, 2000*, have indeed implemented a non-normal flow rule in the VP model, using a Mohr-Coulomb yield curve with an elliptical cap ("modified Coulombic" curve). They have also found that a non-normal flow rule affects the orientation of deformation features in the VP rheology. This work is cited in the discussion section, but not really discussed in terms of the differences or similarities between both approaches, nor in terms of the advances of the present study compared to this previous one. I suggest clearly stating that is new here and what is the broad relevance of the results. The model of *Hibler and Schulson, 2000* has also been used by *Hutchings, Heil and Hibler, 2005* who have looked at intersection angles. They have compared simulated angles between the modified Coulombic and the elliptical yield curve. Mentionning these previous results and comparing them with the current study would be interesting and would strengthen the litterature review and Discussion part of the paper.

I therefore recommand major reviews to clarify the important points above before a resubmission. More specific comments that are often linked to these major comments are listed below.

**Page 1, lines 8-9:** "A newly adapted theory (...) predicts numerical simulations of the fracture angles (...) with a rootmean-square error below 1.3 degrees." This formulation is unclear and needs rephrasing: a newly adapted theory is implemented in the VP model and leads to prediction of the prescribed fracture angle with a RMS error below 1.3 degrees"?. Also, se my main comment about the agreement of the theory with your modeled angles.

Page 1, line 11: I suggest dropping "In conclusion" from your abstract.

**Page 1, lines 14-15:** "to make the fracture angle independant of (not on) the confining pressure (as in observations). This relates to another of my main comments : what sea ice observations support that fracture angles are independant of the confining pressure? Please give supporting references. Is it even possible to distinguish between fracturing processes ocurring in confined and unconfined conditions in the sea ice cover at the geophysical scale?

**Page 1, lines 19–20:** "narrow lines of deformation observed in the Arctic sea ice cover, emerge in high-resolution simulations (*Kwok, 2001; Hutchings et al., 2005*)". It would be relevant to cite more up-to-date works on high-resolution simulations here.

Page 2, line 23 : "The ice strenght locally depends on the ice thickness". This is only partially true: local ice strenght does not depend only on local ice thickness. This sentence perhaps needs some rephrasing.

Page 2, lines 25-27: "In granular media like sea ice (...) Note, that in this study, we consider sea ice to be granular not only in the marginal ice zone, but also in pack ice, where ice floes are densely packed". This again one of my major concern: what is the basis for this assumption? How do you reconcile this assumption with the fact that your goal is to reproduce *fracture* angles in sea ice? Does pack ice, newly-formed ice or any ice that is *not yet fractured* into floes or constituted of agglomerated, refrozen floes always present the characteristics of a granular media? Please explain and also give some support for this assumption.

Page 2, line 28: "This anisotropy". This is unclear. Please define this anisotropy and better explain how it emerges.

Page 2, line 37: The brittle model used in *Rampal et al., 2016* is the EB model of *Girard et al., 2011*. Please modify the reference.

**Page 2, line 39:** I believe a simpler and scientifically more objective formulation would be "most widely used", instead of *de facto* standard.

**Page 2, lines 48-49:** Yes, granular media indeed present shear bands, which are not the same as fractures. Again, please clarify what you want to represent in your model. What is the link between LKFs in sea ice, shear bands in granular media and fractures in solid materials?

**Page 2, lines 48-49 vs line 50:** "Two classical solutions coexist and set two limit angles for the orientation of fractures: the Coulomb angle (...)". There is something unclear and contradictory between this and the previous sentence. You invoque the Coulomb theory here, in the context of friction or fracturing? I understand it is the later, but please make that clear by answering my previous comment.

**Page 3**, **line 56**: I think it would be relevant to make some space and re-introduce the definition of the dilantancy angle here : it would make life easier for the reader and avoid the need to dig for it in another article.

**Page 3**, **line 58**: "A general theory derived from experiments with sand that takes into account both the angle of friction (...)". In the case of sand, contact and friction are indeed at play and shear bands are formed. This again adds to the confusion: internal angle of friction or angle of friction? i.e., fracture or friction? Please clarify.

Page 3, line 60: based *on* the grain size.

**Page 3, lines 67-68:** "a larger dilatancy angle implies a larger grain size, more contact normals, hence more friction". Can you please include some references that support this?

**Page 3, line 73:** There is a mistake here, as *Weiss and Schulson, 2009* reported observed fracture angles between 20 and 50 degrees. Or did you derived this directly from their estimated internal friction angle, which is fitted to insitu stress measurements? In the later case, this is then not an observation of fracture angles but a derivation based on some physical assumptions, which are moreover debatable (see *Dansereau et al., 2019* and many others), and it should be removed from the list of observations of fracture angles.

**Page 3, lines 74-76:** You state that uni-axial compression experiments showed that (3) the fracture angle is a function of the confining pressure. How did you determine that without performing bi-axial compression experiments? Is there a typo here?

Page 3, line 75: the "gradient" of shear to compressive strenght. Did you mean the ratio?

**Page 3**, **line 76-79**: See again my major comment about the apparent confusion between fracturing, friction, granular media, sea ice and a viscous-plastic continuum rheology. I think it is crucial to clarify the links you make between these processes and the motivation of your approach here. This passage in particular leads the reader to believe that your goal is that the VP rheology complies with observations of granular media behavior, because you consider that sea ice at the geophysical scale, in all its different states, is a granular media. If this assumption is at the very basis of your approach, it should be stated earlier in the introduction, (very importantly) along with supporting arguments. This would make the reading and the assessment of your assumptions and methods by the reader much easier.

Page 3, line 82: "The ratio of shear and divergence along the LKFs allows to infer the dilatancy angle." Again, if one assumes sea ice in any state behaves as a granular material.

**Page 3**, **lines 84-85**: "Separating the link between the fracture angle and the flow rule from the yield curve is necessary to design rheologies that are consistent with observed sea ice deformations". Please note that this would be only true for plastic flow rheologies and not applicable nor necessary for rheologies based on elasticity (EB, MEB, Elastic-Decohesive). To be objective, this statement should therefore be modified as "necessary to design plastic flow rheologies that are consistent (...)".

**Page 4**, **line 90**: "In these different classses of models, various rheologies can be defined". This is not true and/or not clear: these are rheological models and therefore they do not include different rheologies. I think that you mean that these different models require the definition of different components: a constitutive relation (all models), a yield/damage curve/criterion (all models including a threshold mechanism, i.e., a change in mechanical behavior) and a flow rule (only plastic flow models). I therefore suggest to rephrase and clarify this passage and the next sentence, that is "in a VP rheology, a yield curve and plastic potential (flow rule) must be defined". In the same line of idea, I do not really see the point of the last sentence of this paragraph. Maybe it can be cut if some rephrasing is made at the beginning of the paragraph?

Page 4, lines 96-97: See my major comment above. *Hibler and Schulson, 2000* have indeed used a VP model with a non-normal flow rule and a Mohr-Coulomb yield curve with elliptical cap, or "modified Coulombic" curve, as cited in your Discussion section. This model has also been used by *Hutchings, Heil and Hibler, 2005* (https://doi.org/10.1175/MWR3045.1) who have looked at intersection angles and compared them between the modified Coulombic and the elliptical yield curve. As their approach is therefore close to yours, it would be important and certainly interesting to explain the similarities and difference between your work and theirs in the litterature review (introduction) section. Please also note that *Hibler and Schulson, 2000* do not seem to share your view that the angles of fracture in sea ice at the geophysical scale are independent of confinement, which would be an important point to discuss further.

Page 4, line 100: "viscous-plastic materials" or "a viscous-plastic material", "with any flow rules".

Page 4, line 100: "from *the* yield curve".

**Page 4, lines 101-102:** "The new model is tested in simple uni-axial loading experiments". See my major comment above: a quick addition to your work would be to test if your numerical implementation also holds under bi-axial loading conditions, that is, if the angles vary or not with confinement.

**Page 4, line 108:** "We consider sea ice as a 2D viscous-plastic material". See my previous major comment: please explain the physical link between this viscous-plastic assumption and that of a granular material.

**Page 4, line 113:** In your case, the constitutive equation links the *vertically integrated* stress tensor to the deformation rate, which you introduced on the previous line.

**Page 4**, **lines 17-19**: Representing small deformations with a viscous model is rather counter-intuitive, especially for a reader that is familiar with viscous-plastic rheologies (plastic for small, viscous for large deformations). I believe it is important that you explain in more details how a viscous rheology is expected here to represent the small deformations of a solid (time scales, viscosities, etc).

Page 5, line 130 to page 6, line 149: These paragraphs could be shortened by removing or presenting in a more concise manner some general pieces of information.

**Page 5, lines 130-131:** As it is not the states of stress that are deforming plastically, but the material, this sentence needs some reformulation.

Page 9, line 204: "The slope of *the* yield curve". And many other missing "the" throughout the text.

**Page 10, line 223:** How does the no-slip condition at the bottom boundary affect your results compared to the case in which slip is allowed in the x-direction (i.e., by holding only one of the two bottom corners of the domain fixed in x and y)? Such boundary conditions are maybe less representative of a floe that sticks to a coast but would not lead to as much concentration of stresses on the bottom corners of your ice floe (here your Bcs imply some bi-axial compression at the bottom) and hence would put less contraint on the appearance of conjugate faults and on their orientation. I think this would be an interesting and not time-consuming test.

Page 11, line 240: I suggest "more *numerically* challenging".

**Page 11, line 256:** "laboratory experiments". If you compare your results with laboratory experiments, please provide more details on these experiments (e.g., boundary conditions? biaxial or uni-axial compression? on samples with an aspect ratio similar to sea ice, i.e., virtually 2D? on fresh or sea ice?) Were such experiments made by *Erlingsson et al.*, *1988* and *Wilchinsky et al.*, *2010*?

Pages 11-13 and caption of figure 6: What is the field represented in figure 6? I assume from the color scale that it is a deformation rate?

Section 4 and figures 6 and 7: How are the angles of the features observed on fields such as shown on figure 6 measured, i.e., estimated? It would be important to mention what method is used.

**Result section, figure 7 and page 15, lines 292 and 306-308:** "the theory predicts the fracture angles accurately" and "The results illustrate clearly how the yield curve defines the stress for which the ice will deform, that is, the transition between viscous and plastic deformation, and how the relative shape of the plastic potential with respect to the yield curve defines both the type of deformation (convergence or shear) along the fracture line and the fracture angle. The resulting fracture angles are in excellent agreement with the Roscoe angle predictions (Roscoe, 1970)."

There is my major comment about the results. In section 2.3, you describe how the yield curve, flow rule and angles are related in your model. By prescribing the yield curve and plastic potential ellipse ratios, you prescribe locally the angle (Roscoe) of "fractures". Figure 7 shows that at the macro-scale, i.e., the scale of the ice floe you indeed retrieve that angle. What is prescribed at the local scale is what you get at the macro-scale in your model, as expected in a model of plastic flow. Therefore my understanding is that these tests serve to verify that your numerical scheme is OK. Is that the case?

To better illustrate that point, it would be relevant to show the (deformation?) fields at different stages of the compression experiment, to illustrate how the features arise in your model.

Page 15, line 300 : "the shape of *the* plastic potential".

**Page 15, line 305 :** "this allows decoupling the mechanical strength properties of the material (ice) from its post-fracture behavior". Again the contradiction with the assumption of a granular material, i.e., an already fractured/fragmented material. How do you reconcile these ideas?

**Page 15, lines 306-308:** "The results illustrate clearly how the yield curve defines the stress for which the ice will deform, that is, the transition between viscous and plastic deformation, and how the relative shape of the plastic potential with respect to the yield curve defines both the type of deformation (convergence or shear) along the

fracture line and the fracture angle. The resulting fracture angles are in excellent agreement with the Roscoe angle predictions (Roscoe, 1970).''

But you *prescribe* the yield and plastic potential in your model: why would you not expect what you get to indeed be what you prescribe? In other words, you do not make any distinction between what you prescribe at the micro-scale (scale of your discretization) in your model and your macroscale results and you do not discuss why you expect these behavior to be identical or not : that is missing from your work and interpretation of your continuum model.

Page 15, point 2: About confinement, shear bands and fractures, see my major comment above.

**Page 17, line 382:** "sea ice mechanical strenght properties (yield curve) and deformation (flow rule)". Again, you write this with the perspective of a VP model, but mechanical strenght properties and deformation are not only determined by the yield criterion and flow rules in other rheological models for sea ice. Please be specific and make this distinction clear. Also, I do not undestand why *Dansereau et al., 2016* is cited in this context.

**Page 17, lines 387-388:** "So is the combined knowledge of the failure stresses and their associated deformation of sea ice as a 2D granular material". This is confusing: why then do you base your approach on the assumption of a granular material? This goes along my main comment and really needs to be clarified.

---

## Author Response (AR2)

**Authors answer to Veronique Dansereau comments for tc-2020-153 - Round 2**

March 24, 2021

Dear Editor,

You will find below our answers to the referee's comments. This answer document is composed of a main part summarizing the changes made to the manuscript, and a supplement which includes a detailed point-by-point answer to the reviewer.

We thank Véronique Dansereau for her additional comments and thorough review of the revised manuscript. Her comments and suggestions improve our manuscript.

Yours sincerely,

Damien Ringeisen, On behalf of all three authors.

**Main modifications**

The comments from the reviewer can be summarized as follows:

- 1. State clearly which process is modeled here, brittle behavior or granular behavior.
  - We added a statement clarifying that we model sea ice as a granular material, or in other terms, a fractured system. (L35–41)
- 2. The definition of "rheologies" is not clear and needs to be improved.
  - We have included a clearer definition of the terms *rheological models* and *rheologies*. Specifically, we define the physical behavior as the "*rheological model*" (e.g., Viscous-Plastic (VP) ), and "*a rheology*" as a given set of constitutive equations within a rheological model. A VP rheology is defined by the shape of the yield curve and the orientation of the flow rule (i.e., the shape of the plastic potential). (L43–47)
- 3. State why the viscous-plastic rheological model is suitable for modelling sea ice as a granular material.
  - We added a statement explaining why the VP rheological model is suitable for modeling sea ice as a granular material (L55–60):
    - (1) It includes a yield condition for the transition between small quasi-rigid viscous deformations and large plastic deformations.
    - (2) It includes a plastic flow rule which allows to represent divergence or convergence along the shear lines, i.e., the dilatancy observed in granular materials.
- 4. Explain why you would expect an angle different from the Roscoe angles  $\theta_R$ .
  - We compare different concepts (Coulomb and Roscoe) for the orientation of LKFs with a non-normal flow rule. In previous studies with non-normal flow rules, the effect of the non-normality was not considered. We rephrased two sentences to make this point clearer. (L100–102) and (L224–L226)
- 5. Add a statement justifying why LKFs intersection angles should not depend on confinement.
  - We added references reporting similar fracture angles at different geographical locations, indicating that the fracture angle may not be influenced by confining pressure. (L116–L117)
- 6. Make Section 2.1 shorter for your specialized audience.
  - We decided not to shorten Section 2.1 for the sake of completeness and to make the article more accessible to the general reader.
- 7. State how the angles are evaluated and what is the accuracy.
  - We now specify that the angles are measured manually and that the accuracy is  $\pm 1^{\circ}$ . (L301–302)
- 8. Describe what happens in the first five seconds of the simulation.
  - We include a sentence describing that the fracture is created on the first timestep and develops during the rest of simulation. (L303–307)

Other minor answers and corrections are kept in the supplement.

**Supplement – Point-by-point answer**

Note:

- The referees comments from the 1st round are shown in black and numerated with  $\mathbf{R1}$ .
- The authors answers and modifications from the 1st round are shown in italic black.
- The referee's comments for the 2nd round are shown in red and numerated with **R1.2**.
- The authors answers for the 2nd round are shown in blue, with the modifications shown in italic.

We also remove the comments from the 1st round that were not subject to a new comment or answer from the reviewer during the 2nd round.

**R1.2#1,** Thank you for your response to my comments on your paper. The modifications you have made improve readability. In particular, the abstract is much clearer and the addition of the definition of the different angles makes the reading easier.

You have provided some elements of answer to my main comments in your responses, in particular, to the first comment (R1-2). However, I find that the changes you have made accordingly (in the introduction in particular) are not sufficient to address the point I wanted to make with this comment, which is: you need to state and explain clearly the assumptions behind your work.

In other words, the goal is not to present these assumptions to me, as a reviewer, but to your readers. Therefore, I would strongly suggest that you put the list of arguments you have presented to me in R1-2 in the text (e.g., around page 2, lines 32 to 39), to explain that sea ice present both brittle and granular behaviors, but that here you consider it to be a granular (already fractured) media in the context of shearing band angles at the regional to global scale (i.e., the scale of sea ice models). In think this would really help following your line of thoughts, understand your approach and strengthen the manuscript.

We have restructured the introduction and included new material to address the reviewer's comments – without making it longer. The revised introduction includes the fact that sea ice has granular material properties as well as brittle behavior and that we model sea ice as a granular material.

We rewrite the 1st and 2nd paragraph of the introduction on L35 of the revised manuscript:

"Sea ice plays a significant role in the energy budget of the climate system and therefore has a strong influence on future climate projections. Sea ice dynamics are located primarily along narrow lines of deformation, called Linear Kinematic Features (LKFs), where floes slide along and grind against each other. LKFs can form in divergence, creating stretches of open water or leads, or in convergence, creating piles of ice or ridges. LKFs in the Arctic sea ice cover influence the Earth system in many ways: heat and moisture exchange take place primarily over open water (Badgley, 1965), and salt rejection during ice formation in leads creates dense water and influences the thermohaline circulation (Nguyen et al., 2011, 2012; Itkin et al., 2015). Locally, the ice strength depends on the sea ice state (e.g., thickness, concentration, and damage), which in turn is affected by sea ice fracture with thermodynamic growth in opening leads and with local dynamical growth during ridge formation. One observable and quantifiable feature of LKFs in Arctic sea ice is the intersection angles between individual LKFs. The LKFs have an influence on the local ice strength, emergent anisotropy and future deformation in the pack ice, and therefore sea ice mass balance (Aksenov and Hibler, 2001). Reproducing the LKFs patterns, density, and orientation is important for accurate sea ice and climate projections at high-resolution.

LKFs are ubiquitous features of granular media, and sea ice is often described as such a granular material (Overland et al., 1998; Erlingsson, 1988; Anderson, 1942; Schall and van Hecke, 2010). Similar to the crumbling of rocks, sea ice also exhibits brittle fracture, as floes break into smaller pieces. Brittle behavior adds a level of complexity because it implies that models must represent both the dynamics of intact ice (brittle — fracture or elastic regime) and the dynamics of a fractured system (granular — friction or plastic regime) (Handin, 1969). The dominant deformation process along LKFs is shear. Sometimes this shear is associated with non-zero divergence, and this divergence along shear bands is referred to a dilatancy (Stern et al., 1995). Granular theory can explain the dilatancy along LKFs. In this work, we consider sea ice as a granular material and focus on the dynamics of the fractured system."

More generally, my point is: your paper addresses an issue that will be of interest for a very specific group of sea ice modellers concerned with the details of its mechanical behavior and numerical representation. These people know what sea ice is, know about VP and will most probably be knowledgeable in mechanics (i.e., on granular vs plastic vs brittle behavior and models). I believe that what they need is to be guided through the physical assumptions that you make to be convinced that your approach is physical and relevant to their modelling. We decided to keep most of the material in section 2.1 for the sake of completeness and to make the article more accessible to the general reader, as opposed to more specific and targeted at a small audience.

I therefore suggest a "major" revision in the sense that I think some, perhaps locally substantial, changes need to be made to the introduction in particular, but you already have brought up some references and a bullet point list of your arguments to me in your review, so introducing them in the text to support your approach should not be too time consuming. This will likely lengthen the text. Consequently, and in the line of idea of my previous comment (that people who will read your paper to improve their sea ice simulations will probably know VP), I suggest below some cuts to generic elements in section 2.1 that would make it shorter, while still keeping in mind that you wanted to keep a full description of VP. We remove small parts of Section 2.1 as suggested by the reviewer (See comments R1.2#23–

There are three more precise points on which I would like to have your comments or answer:

 one unanswered question: what is the method to evaluate the angle from your simulated fields (i.e., what does the Measure Tool from GIMP, what is the method and the related errors)? Please briefly summarize it in the text. The angles are measured manually and the accuracy is ±1°. We modify a sentence in the

revised manuscript to clarify this (See R1.2#19 below).

R1.2#39).

2. in my point of view, the introduction (around page 2, lines 40-52) still lacks an explanation on why you think VP is an appropriate rheology for a granular media (could be short).

We now include a sentence clarifying why VP is an appropriate rheology for a granular media

We add on L55 of the revised manuscript: "The Viscous-Plastic rheology is an appropriate

continuum rheology for modelling sea ice as a granular material because it includes (1) a 2D yield condition for plastic deformation defining the internal stress stress for fracture medium starts deforming, and (2) a flow rule that allows to represent the divergent and convergent motion along shear lines, that is, the dilatancy. Continuum plastic flow models are often used in other scientific fields to model granular geo-materials (Vermeer and De Borst, 1984; Mánica et al., 2018)."

3. my question in R1-5 remains unanswered and I have rephrased it below to make it clearer. Here, we compare two concepts for the orientation of shear lines in sea ice VP models with non-normal flow rules (Coulomb and Roscoe). Previous studies did not consider the effect of non-normal flow rules of the orientation of LKFs. We revised the introduction to make this point clearer (See R1.2#9 below).

I have also added some questions and comments about your responses and put some minor comments at the end of this review.

Here are my major comments/concerns :

- $\mathbf{R1}\#2$ , It does not appear clear in the paper what physical process(es) the authors really want to model. In the introduction, it is mentioned that sea ice, both in the pack and the marginal ice zone, is considered as a granular material. No physical justification is offered for this assumption. The rheology used to model this granular material is one of plastic flow, but the authors do not explain how they reconcile their continuum viscousplastic model with a granular behavior. The aim is apparently to reproduce fracture angles (repeated terminology for the features simulated by their model), but the authors do not explain the link between plastic flow, fracturation and the mechanical behavior of a granular material, which is an already fractured/fragmented material in which contacts and friction dominate. Later, it seems that the authors refer to shear bands in granular materials as if they were associated with the same processes as a fracturing solid. The Coulomb theory is invoked but it is not clear if it is in the context of friction or fracture. There is therefore much confusion throughout the paper as to what the authors consider is the mechanical behavior of sea ice : is it characterized by fracturation? By friction and contacts between already broken up floes? Granular materials like sand are invoked, but is sea ice really assimilated to a sand-like material here? Whatever is assumed, it crucially need to be clarified and all physical concepts untangled throughout the text in a way that makes physical sense.
  - Sea ice is composed to individual floes that vary in size and thickness along seasons and conditions. Sea ice has often been described as a granular material (Overland et al., 1998; Mcnutt and Overland, 2003; Tremblay and Mysak, 1997). In other fields, granular material has been modeled with continuum plastic flow models, considering both the Coulomb theory or the Roscoe theory (Vermeer and De Borst, 1984; Vermeer, 1990; Balendran and Nemat-Nasser, 1993; Mánica et al., 2018).
    R1.2#2, Yes indeed.
  - We think that we need to consider the ice as a granular material if we want to explain divergence along fracture lines (Stern et al., 1995; Bouchat and Tremblay, 2017).
     R1.2#3, Why? You need to extend on this.

We observe far-field compressive fractures that create LKFs with opening/closing, i.e., some dilatancy. Granular mechanics can explain these non-zero divergence along LKFs during far-field convergent events.

We add on L39 of the revised manuscript "The dominant deformation process along LKFs is shear. Sometimes this shear is associated with non-zero divergence, and this divergence along shear bands is referred to as dilatancy (Stern et al., 1995). Granular theory can explain the dilatancy along LKFs."

The fact that the elliptical yield curve with normal flow rule (Hibler, 1979) feature compressive states with divergent opening (also when low confinement is applied) (Ringeisen et al., 2019) shows that we can consider granular dynamics to already be present in current VP models. In this manuscript, we investigate a modification of the VP model with elliptical yield curve.

- We do not consider sea ice to behave like sand, but still as a granular material: a 2D granular material. Sea ice is peculiar in the world of physics, because (1) it is bound to the 2D ocean-atmosphere interface by gravity, but can "escape in the vertical dimension" (page 17, line 389) and ridge when bi-axial compression exceeds a critical threshold. Also ice floes, the "grains" of sea ice, can brake or refreeze. Therefore, sea ice dynamics exhibits a large spectrum behaviors, including characteristic granular dynamics, for example dilatancy, as well as brittle behavior.
- The terms referring to brittle behavior, such as fracture angle or fracture lines, might be slightly confusing with the idea of sea ice as a granular material, but we would like to keep them as it is. Here is our reflection:
  - \* If we agree on the fact that sea ice is already a fractured medium, we study the large scale deformation of a compact ice field, process similar to the creation of fracture in continuous solid.
  - \* In that case, it makes little sense to us to make a distinction between fracture and friction. This is well described in the abstract of (Wilchinsky and Feltham, 2011): "Sea ice failure under low-confinement compression is modeled with a linear Coulombic criterion that can describe either fractural failure or frictional granular yield along slip lines." The assemblage breaks and floes interact with one another, which can be seen as the microscopic behavior of friction.

**R1.2#4**, Of course both fracture and friction are present within sea ice. But please note that the Coulomb theory has a very different interpretation for fracture than for friction, although the equations are the same. I made that comment because is a difference that you should be aware of and not mix-up in the text because it brings a lot of confusion.

We now state in the revised introduction that we model sea ice as granular material (See R1.2#2 above)

\* Furthermore, the creation of LKFs in sea ice was already associated with breaking behavior (Erlingsson, 1991; Marko and Thomson, 1977), the term fracture is repetitively used (Hutchings et al., 2005; Hibler and Schulson, 2000), as well as the fact sea ice is granular medium (Wilchinsky and Feltham, 2011; Hopkins, 1996).

**R1.2#5**, Which part are you modelling? The "breaking" behavior or the granular regime? I assume it is the granular regime, but please make this distinction in the text (see my my comment above).

We model the granular regime. We clarify this in the revised introduction (See R1.2#1 above)

\* Furthermore, for clarity, we would like to keep the same terminology as in the Ringeisen et al. (2019), on which this study is based.

In order to address these points, we modify the manuscript:

- "Note, that in this study, we consider sea ice to be of granular nature not only in the marginal ice zone, but also in pack ice, where ice floes are densely packed. For this reason, we can consider the creation of an LKF as a process that involves both fracture and friction (Wilchinsky and Feltham, 2011)." on L33 of the revised manuscript.
- We modify the penultimate paragraph of the introduction (see also comment R2#4). It now reads "In this paper, we investigate the effects of a non-normal flow rule on fracture angles. We use the non-normal flow rule as a means of separating the state of stress (at failure) and the post-fracture deformation. To this end, we study the non-normal flow rule in the context of the standard VP rheological model using a similar shape for the plastic potential (i.e., an ellipse) because (1) the ellipse is widely used in the community, and (2) its behavior is well documented (compared to other models), providing a solid basis for comparison. For these two reasons, we use the elliptical yield curve despite the fact that it is not the most appropriate yield curve to model sea ice as a granular material like sea ice. This paper provides a new generalized theoretical framework for any viscous-plastic material with normal or non-normal flow rules. Following Ringeisen et al. (2019), we test the new model in simple uni-axial loading experiments where the relationship between fracture angle and flow-rule can be easily identified."
- R1#3, In the same line of ideas, the authors seem to base their assumption of sea ice being a granular material on observations supporting fracture angles that are independent of confining pressure. It appears that they aim at developing a model that complies with these observations. However, no reference of observations, neither at the lab nor the geophysical scale, is clearly associated with this statement. One can reasonably wonder if making such observation would be possible in the case of sea ice at the geophysical scale: how would it be possible to determine far field stresses and distinguish between unconfined and confined states? Do unconfined compression leading to fracture even occur in circumstances other than an individual ice floe crashing into a coast? References are lacking here to support this assumption of independence of confinement and should crucially be added.

Concerning the granular matter behavior:

- Fracture angles (or orientation of the shear bands) that are independent of the confinement pressure are characteristics of granular material, and lead to the use of the Mohr-Coulomb yield criterion.
- More recent studies showed that shear bands orientations in granular materials increase slightly with confining pressure (Alshibli and Sture, 2000; Han and Drescher, 1993; Desrues and Hammad, 1989, Note that some of these studies show a decrease, but only because they use the complementary angles.). However, this change is very limited: of the order of 5°, with a stress confinement ratio of in the range [0.05-0.5] depending on the confining pressure and the grain size.

**R1.2#6**, Please note that at least Desrues and Hammad, 1989 used sand in their (3D, not 2D) experiments which is very different as a material than sea ice (in terms of the dispersion of grain sizes, friction, 3D vs 2D), hence you should be carefull

with the statement that shear band angles in granular material do not vary with confining pressure.

We agree. Unfortunately, to our knowledge, there is no laboratory experiments looking at the deformation of sea ice with different confinement pressure.

• The magnitude of the change of angle contrasts with the effect of confining pressure with the elliptical yield curve, where a stress-ratio of 0.3 changes the fracture from divergent to convergent and the fracture angle from ca. 34° to 46°.

Concerning the sea ice behavior:

- The observations of the same fracture angles at different scale (so probably different stress conditions) by several studies (Erlingsson, 1988; Marko and Thomson, 1977; Cunningham et al., 1994) is an indication that fracture angles might be independent of the stress conditions, i.e. different confining pressures. New datasets of intersection angles from LKFs tracking show that coulombic fracture in the Arctic sea ice shows a predominant angle (Nils Hutter, personal communications)
- It is correct that, at high confining pressure, the fracture angle probably changes, especially when sea ice reaches a ridging state. This can be seen with the shape of the yield curve observed in Schulson (2004); Weiss and Schulson (2009). Please see also our answer to Reviewer  $\tilde{\#}2$  in comment R2#40.
- See also our answer to comment R2#39 of Reviewer $\tilde{\#}2$ .
- Finally, we agree that far field stresses are difficult (or close to impossible) to determine, this is why observing the angle of dilatancy along LKFs could be a good metric to improve sea ice models.

To clarify our manuscript, we make the following modifications:

- We modify our statement: "... namely that shear band orientations and divergent or convergent motion at the slip lines are a function mainly of the shear strength of the material and orientation of the contact normals (or dilatancy angle), and that the confining pressure has only a limited effect (Alshibli and Sture, 2000; Han and Drescher, 1993; Desrues and Hammad, 1989).", L107 of the revised manuscript.
- The sentence on L369 now reads "... unlike laboratory experiments with granular materials (e.g., sand) where the fracture angle is only weakly sensitive to the confining pressure (Han and Drescher, 1993; Desrues and Hammad, 1989; Alshibli and Sture, 2000).".
- We modify the following statement: "... A 2D material, such as sea ice, can ridge and "escape to the 3rd dimension" after fracture. Therefore, we expect a change in the fracture angles at large confinement. Laboratory experiments show this behavior and yield stresses in sea ice change above a critical confinement ratio (Golding et al., 2010; Schulson, 2002). It is still not clear whether these results can be extrapolated to the modeling sea ice as a 2D medium at the geophysical scale, although several common features can be found (Schulson, 2002)." L375 of the revised manuscript.

R1.2#7, Again, my point is that you need to state and explain, in the text, the assumptions you make and then refer to the literature supporting your approach. For instance, here, you start by your main statement, "we consider sea ice as a granular

material", then, "and as such we consider that shear bands vary weakly with confining pressure", citing the references you give here in your response.

I really believe that this will help the reader understand your thought process **and relate to studies they already know of**.

To make the link between studies on granular media and the behavior of sea ice and to support your assumption, it would be highly relevant to include a figure, e.g., of the predominant angle you say is observed by Nils Hutter. Would that be possible? Or is it the range 20-25 you later cite in your paper from Hutter and Losch 2020? Otherwise, citing what you included here in bullet points ("The observations of the same fracture angles at different scale (so probably different stress conditions) by several studies (Erlingsson, 1988; Marko and Thomson, 1977; Cunningham et al., 1994) is an indication that fracture angles might be independent of the stress conditions, i.e. different confining pressures.", etc) would be a start.

The revised introduction clarifies that we model sea ice as granular material. (See R1.2#1 above).

The distribution of intersection angles is subject two studies currently being written. Therefore, we would prefer to keep this private for now. But citing the aforementioned studies is this context is a good option, we thank the reviewer for the suggestion.

We add in L115 of the revised manuscript "The fracture angles are similar in different regions of the Arctic with different background stress conditions (Erlingsson, 1988; Marko and Thomson, 1977; Cunningham et al., 1994). This observation supports the hypothesis that the angle of fracture is independent of the confining pressure."

- R1#4, Also somewhat contradictory is the fact that the authors use an elliptical yield curve and plastic potential to model a material that they consider as a granular. I understand this is perhaps temporary and other criterion will eventually be investigated, but in the meantime, are there examples of granular materials that have been observed to follow this kind of yield curve/flow rule? References of such examples would strengthen the paper.
  - As the reviewer stated, the use of elliptical yield curve is transitory, but practical for the main goal of this study: that is, studying the effect of a non-normal flow rule on the angles of fractures, and provide an theoretical explanation for this effect.
  - We use an elliptical yield curve in this study for 2 reasons: (1) Because it is widely used in the sea ice community, for instance 30 out of 34 sea ice models in GCMs participating in CMIP5 use the standard VP model or a modification thereof (Stroeve et al., 2014), and (2) because the behavior of the elliptical yield curve with normal flow rule in uni-axial compression has been recently investigated (Ringeisen et al., 2019), and we want to isolate the effects of using a non-normal flow rule.
  - Elliptical yield curve, like the Von Mises yield curve, are used in material modeling, especially for ductile materials. Although their formulation is different that of in the sea ice models. Granular materials usually use an incompressible formulation, while sea ice needs a non-zero divergence term to represent open water formation and ridging.

To clarify our manuscript, we make the following modifications:

• "We discuss the elliptical yield curve here because it the most commonly used one and its behavior is better documented than any other model in use in the community. This provides a known reference for studying the use of non-associated flow rules. Our goal is to provide a reference for the future development of viscous-plastic rheologies with non-normal flow rules rather than suggest a new VP rheology." on L390 of the revised manuscript.

**R1.2#8**, Thank you for this addition. I would modify the sentence for improved clarity as "it is widely used for sea ice and its behavior is better documented than any other yield curve used in the sea ice community" and add "because the behavior of the elliptical yield curve with normal flow rule in uni-axial compression has been recently investigated (Ringeisen et al., 2019), and we want to isolate the effects of using a non-normal flow rule" so that the reader understands that your papers are related (and that you want to use the same terminology).

This is clarified later in the same paragraph. For this reason, we opted to keep this paragraph as it is.

- R1#5, Another concern is in the interpretation of the results. A model of plastic flow is used here, not a model of fracture (neither heterogeneities, nor elastic interactions, nor a mechanism representing breakage of bonds or damage is included here). In such model, one expects the simulated macroscopic behavior (that of the ice floe in this case) to coincide with the theory prescribed at the local scale, i.e., the constitutive equation, flow rule, etc. Therefore, as pointed out by Hutchings et al. (2005), if deviations between the simulated angles and the predicted values occurred, they would be indicative of numerical errors. Hence, while it is good to verify that the model does indeed reproduce the Roscoe angle within a small RMS error, doesn't it just show that the numerical scheme of the model works? This point needs to be clarified in the text. It would also be important to mention what method is used to estimate the angles from fields such as the ones shown on figure 6.
  - In sea ice VP rheology, the angle of fracture is not yet understood. For instance, Roscoe and Coulomb theories gives different angles for the same process. We show here that the flow rule affects the fracture angles, and we explain this influence with a theoretical model, adapted from the Roscoe angle. Similar investigations of the angle of deformation features can be found, for example, in the field of lithosphere geophysical modeling: Lemiale et al. (2008); Kaus (2010).
  - The method used to estimate the angles is presented at the end of Sec. 3.

To clarify our manuscript, we make the following modifications:

• We add on L94 of the revised manuscript: "The effects of a non-normal flow rule for sea-ice rheologies (as in e.g., Hibler and Schulson, 2000; Hutchings et al., 2005) on the fracture angles have not been explored. Therefore, it is unknown which of the three theories (Coulomb, Roscoe, Arthur) provide the most accurate prediction for this case." • For comparison and clarity, we add the Coulomb angles predictions on a new version of Fig. 7a, shown below (Figure 1).

**R1.2#9,** I will try to formulate my question more concisely : I wonder why, if you prescribe  $e_G$  and  $e_F$  locally in your model, you do not necessarily expect the macroscopic behavior (in terms of the simulated angle in your rectangular sample) to correspond to your equation 30? What are the reasons why the simulated and theoretical angle could differ, if any?

See my related question below: how does the fracture evolves in your model (in the first 5 seconds of the simulation)?

As we stated before, we compare the results of experiments with two different theories, Roscoe and Coulomb. We show that Roscoe fits the experimental data better. Other papers using a non-normal flow rule do not consider the effect of the flow rule on the angles, only the effect of the yield curve.

Concerning the behavior before 5 s, the fracture is immediately created at the 1st timestep (See R1.2#20 below).

We add the following sentence on L224 of the manuscript, at the beginning of Section 2.3: "The Roscoe angles can then be compared to the Coulomb angles, as defined in Ringeisen et al. (2019), and the results from the idealized experiments in Section 4.".

We rewrite on L100 of the revised manuscript "So far, only the yield curve has been thought to affect the orientation of LKFs (as in e.g., Hibler and Schulson, 2000; Hutchings et al., 2005; Wang, 2006), and the effects of a non-normal flow rule for sea-ice rheologies on the fracture angles have not been considered."

I therefore recommend major reviews to clarify the important points above before a resubmission. More specific comments that are often linked to these major comments are listed below.

**Specific comments:**

 $\mathbf{R1}\#\mathbf{9}$ , Page 1, lines 14-15: "to make the fracture angle independent of (not on) the confining pressure (as in observations)". This relates to another of my main comments : what sea ice observations support that fracture angles are independent of the confining pressure? Please give supporting references. Is it even possible to distinguish between fracturing processes occurring in confined and unconfined conditions in the sea ice cover at the geophysical scale?

Please see our answer to the main comment R1#3.

We replace "independent on" by "independent of"

**R1.2#10**, See my response to R1-3: support for this assumption and references should be included in the text (intro).

We included some references in the introduction that support these assumptions (See R1.2#7 above).

 $\mathbf{R1}\#\mathbf{10}$ , Page 1, lines 19-20: "narrow lines of deformation observed in the Arctic sea ice cover, emerge in high-resolution simulations (Kwok, 2001; Hutchings et al., 2005)". It would be relevant to cite more up-to-date works on high-resolution simulations here.

The idea is here to cite the seminal studies about LKFs, we are now also citing more recent literature.

We add the following references: (Hutter et al., 2018; Koldunov et al., 2019; Heorton et al., 2018).

**R1.2#11,** Page 1, line 33: LKFs do not emerge only in high-resolution simulations (e.x., 10, 20, 40, + km is sufficient in NeXtSIM) depending on the rheology used. You should modify this sentence accordingly.

We add a sentence on L64 of the revised manuscript "LKFs emerge clearly in plastic flow models at high resolution (Hutchings et al., 2005; Hutter et al., 2018; Koldunov et al., 2019). VP models reproduce observed intermittency and spatial localization even without brittle fracture dynamics (Bouchat and Tremblay, 2017; Hutter et al., 2018), albeit at higher resolution than Maxwell-Elasto-Brittle models (e.g., Rampal et al., 2019)."

R1#12, Page 2, lines 25-27: "In granular media like sea ice (...) Note, that in this study, we consider sea ice to be granular not only in the marginal ice zone, but also in pack ice, where ice floes are densely packed". This again one of my major concern: what is the basis for this assumption? How do you reconcile this assumption with the fact that your goal is to reproduce fracture angles in sea ice? Does pack ice, newly-formed ice or any ice that is not yet fractured into floes or constituted of agglomerated, refrozen floes always present the characteristics of a granular media? Please explain and also give some support for this assumption.

We argue that yes, "pack ice, newly-formed ice or any ice that is not yet fractured into floes or constituted of agglomerated, refrozen floes" still carry granular characteristics. The anisotropy at subgrid scale is still present in a way that fracture will rarely be created in straight lines, but will most probably follow the network of weaknesses.

**R1.2#12,** Agreed, but non-straight fracture lines are not a characteristic of granular material only: they occur in any heterogeneous quasi-brittle material. See my response to R1-3: you need to state clearly that sea ice present both brittle and granular behaviors and in which of these regimes you place your study.

In this paper, we consider the dynamics of sea ice as a fractured system, or granular material (See R1.2#1 above)

R1#14, Page 2, line 37: The brittle model used in (Rampal et al., 2016) is the EB model of Girard et al. (2011). Please modify the reference.

Corrected as suggested by the reviewer.

**R1.2#13**, Page 2, line 70: The rheology in Rampal et al., 2016 being the same as in Girard et al., 2011, I would remove the reference to Rampal et al., 2016 (repetition).

Corrected as suggested.

R1#15, Page 2, line 39: I believe a simpler and scientifically more objective formulation would be "most widely used", instead of "de facto standard".

"De facto" means "in fact" or "in effect". We are just stating a fact here.

**R1.2#14**, Page 2, line 72: I still think that an objective sentence would replace "standard" by "most widely used" (your next sentence supports just that) or de facto by "practically". It is not a fact that the sea ice community has defined a standard rheology :)

We absolutely agree with this last sentence, and this is exactly why "de facto standard" is the good formulation. "De facto: existing in fact, although perhaps not intended, legal, or accepted" (Cambridge Dictionary), we mean that VP was not defined as a standard, but grew to be one (i.e., in almost every climate models — for now). This is a Bottom-Up decision, in contrast with a "de jure standard", which is a Top-Down decision, when the sea ice community would decide to make a rheology the standard. We do not wish here to say that the VP model is the best model, or that it is the one and only model to be used.

R1#17, Page 2, lines 48-49 vs line 50: "Two classical solutions coexist and set two limit angles for the orientation of fractures: the Coulomb angle (...)". There is something unclear and contradictory between this and the previous sentence. You invoke the Coulomb theory here, in the context of friction or fracturing? I understand it is the later, but please make that clear by answering my previous comment.

We consider the case of fracture, but this applies also a dense pack of ice floes. We do not understand why these two concepts should be separated. The creation of LKFs in sea ice has been referred to as "fracture" in several preceding publications (e.g., Hutchings et al., 2005).

**R1.2#15**, The Coulomb theory (originally for friction) has been adapted and extensively used to describe fracturing in brittle materials, but these are two completely different phenomena (friction and fracture) and so is the interpretation of this theory in terms of angles. This is why these two concepts should be separated. See my response to R1-3: you just need to state more clearly in a short sentence what you are describing: shear bands in a granular media or brittle fracturing, so that the reader follows your line of thought.

The revised introduction now states that we describe sea ice as a granular medium (See R1.2#1 above).

R1#23, Page 3, lines 74-76: You state that uni-axial compression experiments showed that (3) the fracture angle is a function of the confining pressure. How did you determine that without performing bi-axial compression experiments? Is there a typo here?

No, this is no typo. Ringeisen et al. (2019) showed that the fracture angles changes with the confining pressure when a elliptical yield curve is used, the forcing was uniaxial but the ice was confined, hence similar to a bi-axial loading.

We modify the text to now read: "In Ringeisen et al. (2019), the confinement was achieved by adding thinner ice on either side of an ice slab subjected to uni-axial loading." on L105

**R1.2#16**, I see. It would be clearer and shorter if you wrote "compression experiments with uni-axial loading and laterial confinement added via the addition of thinner ice (Ringeisen et al., 2019)" because uni-axial compression experiments with confinement are in fact bi-axial compression experiments.

We rewrote this part.

This paragraph now reads on L109 of the revised manuscript "In addition, uni-axial loading compression experiments with lateral confinement (achieved via the addition of thinner ice surrounding the ice slab, Ringeisen et al., 2019) showed that:..."

R1#28, Page 4, line 90: "In these different classes of models, various rheologies can be defined". This is not true and/or not clear: these are rheological models and therefore they do not include different rheologies. I think that you mean that these different models require the definition of different components: a constitutive relation (all models), a yield/damage curve/criterion (all models including a threshold mechanism, i.e., a change in mechanical behavior) and a flow rule (only plastic flow models). I therefore suggest to rephrase and clarify this passage and the next sentence, that is "in a VP rheology, a yield curve and plastic potential (flow rule) must be defined". In the same line of idea, I do not really see the point of the last sentence of this paragraph. Maybe it can be cut if some rephrasing is made at the beginning of the paragraph?

A VP model with a different yield curve and/or a different flow rule can describe a different physics in the modeled material. A VP rheology with a Mohr-Coulomb yield curve (e.g. Tremblay and Mysak, 1997) will create different results than the one with an elliptical yield curve. The last statement is important for this paper, because it stresses the fact that changing the flow rule changes the system dynamics.

**R1.2#17,** Again this is not clear: a rheological model has its own rheology, that determines if it is elastic, plastic, viscous, etc. A model with a different yield curve will lead to different results with the same constitutive equation indeed but does not change the relationship stress-deformation. The flow rule problem concern plastic models only. The sentence should therefore read "In these different classes of models, various mechanical components can be defined" or "in the VP sea ice model, various yield curves and flow rules can be defined".

We disagree on this point, although we agree that semantics for sea ice models are globally unclear.:

- The reviewer is right, Viscous-Plastic is the *Rheological Model*. It can be seen as the physical behavior and can be described with basic mechanical elements like Dashpots Elements and Frictional Elements.
- Rheology is wrongly used in most of current sea ice literature, ours included. We could argue that rheology should probably only describe the science of flow and deformation of material. However, the sense of "VP rheologies" as a differentiation between different yield curves in the VP rheological model became dominant, e.g. König Beatty and Holland (2010); Zhang and Rothrock (2005); Ip et al. (1991).
- In some range, this distinction makes sense because changing the yield curve will change the stress-strain relationship. For example, with a Mohr–Coulomb yield curve the ice shear strength always increases as the compression stress increased, this is not the case with the elliptical yield curve.
- Changing the yield curve and/or the plastic potential modifies the formulation of the viscosities  $\eta$  and  $\zeta$ , hence the constitutive equations. In other words, the stress-strain-rates relationship is changed, because the viscosities depend on the strain-rates as well. The behavior is still visco-plastic, but the constitutive equations are changed.

**We rewrite this paragraph starting on L43 of the revised manuscript:**

"Different rheological models assume different material behavior before and after fracture. Common sea ice rheological models are, for example, Viscous-Plastic (VP, Hibler, 1977), Elastic-Plastic (EP, Coon et al., 1974), Elastic-Anisotropic-Plastic (EAP, Tsamados et al., 2013), or Maxwell-Elasto-Brittle (MEB, Dansereau et al., 2016), . In these different rheological models, various stress-strain(-rate) relationships, or constitutive equations, can be defined. In the following, we refer to models with different constitutive equations as different rheologies. We focus on the VP rheological model. A specific VP rheology is defined by a yield curve and plastic potential. The yield curve defines the stress criteria for the transition from small viscous deformations (creep) to the large plastic deformations (friction). The plastic potential determines the ensuing post-fracture deformation, called the flow rule. The flow rule is normal to the plastic potential (Drucker and Prager, 1952). The plastic potential can be independent of, or equal to the yield curve. In the latter case, the flow rule is also normal to the yield curve and is called a normal-flow rule or associated flow rule. Several yield curves have been used in sea ice VP models, some with a normal flow rule (Hibler, 1979; Zhang and Rothrock, 2005) and some with a non-normal flow rule (Ip et al., 1991; Tremblay and Mysak, 1997; Hibler and Schulson, 2000; Wang, 2007)."

R1#33, Page 4, line 108: "We consider sea ice as a 2D viscous-plastic material". See my previous major comment: please explain the physical link between this viscous-plastic assumption and that of a granular material.

See our answer to the general comment R1#2

**R1.2#18**, This comment is not clearly answered in R1-2 and should be included somewhere in the introduction (see my major comment above).

In the revised introduction, we clarify why the sea ice VP rheology is suitable for modeling sea ice as granular material (See R1.2#1 above).

R1#43, Section 4 and figures 6 and 7: How are the angles of the features observed on fields such as shown on figure 6 measured, i.e., estimated? It would be important to mention what method is used.

This is described in Section 3 Experimental setup and numerical scheme, Line 245 to Line 250.

**R1.2#19**, Please add a short description on **how** the GIMP Measure Tool estimates (automatically or not?) the angle from the simulated fields (method, errors?).

The angles are measured manually.

We modify the L301 of the manuscript: "The intersection angles between the LKFs are measured manually with the Measure Tool from the GNU Image Manipulation Program (GIMP, version 2.8.16, gimp. org). We estimated the accuracy as  $\pm 1^{\circ}$  (Ringeisen et al., 2019)."

**R1#44**, Result section, figure 7 and page 15, lines 292 and 306-308: "the theory predicts the fracture angles accurately" and "The results illustrate clearly how the yield curve defines the stress for which the ice will deform, that is, the transition between viscous and plastic deformation, and how the relative shape of the plastic potential with respect to the yield curve defines both the type of deformation (convergence or shear) along the fracture line and the fracture angle. The resulting fracture angles are in excellent agreement with the Roscoe angle predictions (Roscoe, 1970)." There is my major comment about the results. In section 2.3, you describe how the yield curve, flow rule and angles are related in your model. By prescribing the yield curve and plastic potential ellipse ratios, you prescribe locally the angle (Roscoe) of "fractures". Figure 7 shows that at the macro-scale, i.e., the scale of the ice floe you indeed retrieve that angle. What is prescribed at the local scale is what you get at the macro-scale in your model, as expected in a model of plastic flow. Therefore my understanding is that these tests serve to verify that your numerical scheme is OK. Is that the case? To better illustrate that point, it would be relevant to show the (deformation?) fields at different stages of the compression experiment, to illustrate how the features arise in your model.

We show the fracture after 5 seconds of simulation, in order to get the initial fracture and avoid more complex interactions that might create more fractures (see Fig. 6 in (Ringeisen et al., 2019)). Please see our answer to the general comment R1#4

R1.2#20, The sense of my question was : what happens within the first 5 seconds of the simulation? (see my major comment above).

The fracture is created instantly, i.e., at the first timestep. Because the viscosity for the viscous behavior is so large (deformation timescale of 35 years), we do not see the fracture progression.

We add a statement on L303 of the revised manuscript "Although the forced deformation is very slow, the stresses reach the yield curve already in the first timestep (0.1 s). The fracture is created immediately, but because of the large viscosity of the viscous states with a deformation timescale of approximately 35 years the fracture progression is not visible immediately (Ringeisen et al., 2019). Therefore we show the deformation after 5 s. During these 5 s there is no fundamental change other than the initial deformation becoming clearer."

R1#46, Page 15, lines 306-308: "The results illustrate clearly how the yield curve defines the stress for which the ice will deform, that is, the transition between viscous and plastic deformation, and how the relative shape of the plastic potential with respect to the yield curve defines both the type of deformation (convergence or shear) along the fracture line and the fracture angle. The resulting fracture angles are in excellent agreement with the Roscoe angle predictions (Roscoe, 1970)." But you prescribe the yield and plastic potential in your model: why would you not expect what you get to indeed be what you prescribe? In other words, you do not make any distinction between what you prescribe at the micro-scale (scale of your discretization) in your model and your macroscale results and you do not discuss why you expect these behavior to be identical or not : that is missing from your work and interpretation of your continuum model.

See our answer to general comment R1#5

R1.2#21, Please see my major comment above.

We compared two different concept of the angles of fracture because it is not clear which one determines the angles of fracture in a sea ice VP model with non-normal flow rule. We modified the introduction to clarify this point (See R1.2#9).

**R1#48**, Page 17, line 382: "sea ice mechanical strength properties (yield curve) and deformation (flow rule)". Again, you write this with the perspective of a VP model, but mechanical strength properties and deformation are not only determined by the yield criterion and flow rules in other rheological models for sea ice. Please be specific and make this distinction clear. Also, I do not understand why Dansereau et al. (2016) is cited in this context.

We refer to Dansereau et al. (2016) in this context because the way the damage parameters act as the history of the model deformation is very interesting, and could be a representation of the state of the local ice (broken/unbroken), i.e. "sea ice mechanical strength properties (yield curve)" cited before.

We reformulate the sentence on L450 of the revised manuscript "...; the sea ice mechanical strength properties (i.e., yield curve) and deformation (i.e., flow rule for VP rheologies) should vary in time and space depending on, for example, the time-varying distribution of the contact normals, floe size distributions, or a damage parameter, as per observations and laboratory or numerical experiments (Overland et al., 1998; Hutter et al., 2019; Horvat and Tziperman, 2017; Roach et al., 2018; Balendran and Nemat-Nasser, 1993; Dansereau et al., 2016; Plante et al., 2020)"

**R1.2#22,** I see your point, but the numerical experiments in Dansereau et al., 2016 does not show that mechanical properties involved in the damage criteria should depend on the damage itself. Instead this dependency was not added in the model because of lack of agreement in (e.g., experimental) supports, hence my reaction to this sentence. Maybe this sentence is just not clear and should be rephrased?

I see your point, we rephrase this sentence. We do not point here to the idea of a mechanical properties being involved in the damage criteria, but only to the concept of damage for sea ice modelling.

We reorganize the mentioned sentence on L463 of the revised manuscript "...; the sea ice mechanical strength properties (i.e., yield curve) and deformation (i.e., flow rule for VP rheologies) should vary in time and space depending on additional variables or parameterizations, for example, the time-varying distribution of the contact normals (Balendran and Nemat-Nasser, 1993), floe size distributions (Horvat and Tziperman, 2017; Roach et al., 2018), or a damage parameter (Dansereau et al., 2016; Plante et al., 2020), as per observations and laboratory or numerical experiments (Overland et al., 1998; Hutter et al., 2019)."

**Other minor comments**

**R1.2#23,** Page 1, line 32: "In granular media like sea ice"... then "Note that in this study, we consider sea ice to be of granular nature". See my response to your answer to my major comment above. You should first state that you make the assumption that sea ice is mostly of granular nature and give some references supporting this assumption. Hence reverse the two first sentences here. And then next sentence : "For this reason, we can consider here..."

We agree that this could be clearer and its arguments better connected. We rewrite this paragraph to describe the granular properties and the brittle behavior of sea ice and state that we model sea ice as a granular material (See R1.2#1 above).

**R1.2#24,** Page 2, lines 40-42: To avoid repetition: "Other models represent sea ice (...)" and then skip VP as an example.

The whole paragraph was rewriten and now avoid repetitions (See R1.2#17 above).

**R1.2#25**, Page 2, line 43: "In these different classes of models, various rheologies can be specified". ? This sentence still does not make sense. I think it could be just removed without impacting the text.

The whole paragraph was rewritten to clarify the terms *rheological models* and *rheology*, and the fact that (See R1.2#17 above).

**R1.2#26,** Page 2, line 44: "The yield curve defines the stress criteria for the transition from small viscous deformations to large plastic deformations". The deformations are not necessary small or large so I would remove these adjectives. Also, I would add at the beginning "In the VP sea ice model" so that the reader understand that this is inverted compared to standard visco-plastic rheologies (see my previous comment about this). Another solution is to move the sentence on page 2, lines 53 to 55 here to make this distinction clear.

The deformations are small from viscous behaviour (because  $\dot{\epsilon}$  is small, resulting in small deformations), and deformations are large with the plastic behaviour ( $\dot{\epsilon}$  is large, resulting in large deformation). Concerning the second point, this paragraph was rewritten.

**R1.2#27**, Page 2, lines 49-50: "It is important to note that two PLASTIC models..." Corrected as suggested

**R1.2#28**, Page 3, lines 90-91: Again, and in agreement to your response to my main comment, I would not focus too much on sand and would remove this sentence, which is I

believe just an example of the previous one.

We do not agree, this an important information to specify that  $\delta$  and  $\phi$  can be different, and have been measured to be different.

**R1.2#29**, Page 4, line 93: "The theory" change to "the concept"? Corrected as suggested

**R1.2#30,** Page 4, lines 112-113: "in contrast to observing stress which requires in-situ measurements". This comparison is not really relevant here or it would need a longer description. I think it could simply be cut to make the text shorter.

We think that this relevant and keep this short statement.

**R1.2#31**, Page 4, line 122: "for comparision" add with previous simulations. Corrected as suggested.

**R1.2#32**, Page 4, line 123: "sea ice as a granular material like sea ice". Corrected as suggested.

**R1.2#33**, Page 4, line 125: "uni-axial compression experiments VP simulations". Corrected as suggested.

**R1.2#34**, Page 5, line 141: "In an ideal plastic model, the stresses are independent of the strain rates". This part of the sentence could be cut (VP is by definition not an ideal plastic but a viscous-plastic model so this is implicit).

Corrected as suggested

**R1.2#35,** Page 5, line 152: "Some other state variables are a function of P; for instance, the tensile strength T is usually defined as  $T = k t \cdot P$ , where the tensile factor  $k t \downarrow 0$  (König Beatty and Holland, 2010). Others are not, such as the ellipse aspect ratio (Hibler, 1979) or the internal angle of friction (Ip et al., 1991)" These sentences is not directly relevant to what you do in your paper and could be cut.

Corrected as suggested

**R1.2#36**, Page 5, line 154: "For two-dimensional sea ice, stress is a rank two tensor; thus, it has four components." This is very generic and can be cut: if you have mentioned that you consider sea ice to be 2D in the intro it is already implied.

Corrected as suggested

**R1.2#37**, Page 6, line 158: "The yield curve can be represented in principal stress ( $\sigma_1$  and  $\sigma_2$ ) or stress invariants space ( $\sigma_I$  and  $\sigma_{II}$ )." Again, this can be cut.

This is necessary, this is the first time in the paper that the principal stress ( $\sigma_1$  and  $\sigma_2$ ) and the stress invariants space ( $\sigma_I$  and  $\sigma_{II}$ ) are defined. In order for the paper to be self-sufficient, this sentence is kept.

**R1.2#38,** Page 7, line 190-191: "and VP rheologies can be considered as ideal plastic". Please make the distinction! only the (converged) VP model for sea ice could be considered as ideal plastic, on the time scales relevant for sea ice modelling, not standard visco-plastic rheologies (like I said, the viscous vs plastic behavior is inverted with respect to sea ice VP).

Corrected as suggested

**R1.2#39**, Page 12, line 294: "The angle of each fracture lines". Corrected as suggested

---

## Author Response (AR3)

**Author's Answer to editor's comments**

Dear Editor,

We are glad of your decision to accept our manuscript for publication.

Concerning your comments:

- We meant no disrespect. This point was already covered in our earlier review and we should perhaps have repeated it here again for the sake of completeness. What we meant is that we are missing bi-axial test of granular material. The studies cited below are all for tri-axial tests but given that sea ice can "escape" in the third dimension, it is not clear that they are directly applicable to a 2D granular material like sea ice. We clarify this point in the manuscript and add a reference to Schulson (2006) per your suggestion:
    - **We add "Note that some of these last experiments are tri-axial tests, and that bi-axial tests of 2D granular sea ice might yield different results as sea ice can "*escape*" in the vertical direction. Bi-axial tests on sea ice samples show that small confinements lead to coulombic shear faults fractures with a similar internal friction coefficient and similar fracture angle. However, larger confinements lead to a spalling raft-like behavior with a broader range of fracture angles Schulson et al. (2006a)." on L114**
- The other 3 comments were corrected as suggested.
- There exists two main types of fracture, brittle and ductile fracture. Both are observed in granular media and in sea ice. Brittle behavior and granular nature are not excluding each other. Brittle fracture is typically accompanied by crack propagation. In a VP model, we have (brittle) fracture with crack propagating at infinite speed (a VP model is almost an ideal plastic material). One could also argue that "instantaneous crack propagation" means "no crack propagation" and that the VP model really displays ductile fracture (i.e. fracture after large plastic deformation). We interpret the fracture in VP model with option #1 above, but this is still be unclear. However, we have clarified the terms used in the manuscript:
    - **We add "We use the term *fracture* as the failure of a compact assemblage of floes and define the *fracture angle* as half of the angle between intersecting LKFs." on L42**

Best wishes,
Damien Ringeisen
On behalf of the three authors